# A clade of receptor-like cytoplasmic kinases and 14-3-3 proteins coordinate inositol hexaphosphate accumulation

Li Lin Xu[1,2], Meng Qi Cui [1,2], Chen Xu [1,2], Miao Jing Zhang[1], Gui Xin Li[3], Ji Ming Xu[1], Xiao Dan Wu[4], Chuan Zao Mao [1], Wo Na Ding[5], Moussa Benhamed [6], Zhong Jie Ding [1] ✉ & Shao Jian Zheng [1,2] ✉

Inositol hexaphosphate (InsP$_6$) is the major storage form of phosphorus in seeds. Reducing seed InsP$_6$ content is a breeding objective in agriculture, as InsP$_6$ negatively impacts animal nutrition and the environment. Nevertheless, how InsP$_6$ accumulation is regulated remains largely unknown. Here, we identify a clade of receptor-like cytoplasmic kinases (RLCKs), named Inositol Polyphosphate-related Cytoplasmic Kinases 1-6 (IPCK1-IPCK6), deeply involved in InsP$_6$ accumulation. The InsP$_6$ concentration is dramatically reduced in seeds of *ipck* quadruple (*T-4m/C-4m*) and quintuple (*C-5m*) mutants, accompanied with the obviously increase of phosphate (Pi) concentration. The plasma membrane-localized IPCKs recruit IPK1 involved in InsP$_6$ synthesis, and facilitate its binding and activity via phosphorylation of GRF 14-3-3 proteins. IPCKs also recruit IPK2s and PI-PLCs required for InsP$_4$/InsP$_5$ and InsP$_3$ biosynthesis respectively, to form a potential IPCK-GRF-PLC-IPK2-IPK1 complex. Our findings therefore uncover a regulatory mechanism of InsP$_6$ accumulation governed by IPCKs, shedding light on the mechanisms of InsP biosynthesis in eukaryotes.

Inositol hexaphosphate (InsP$_6$), also known as phytic acid, is ubiquitous in eukaryotes and regulates a plenty of cellular functions, including stress responses[1], development[2] and phosphate (Pi) homeostasis[3]. As the overall high amount of P in plant seeds (e.g. cereal and legume seeds), InsP$_6$ nevertheless cannot be efficiently digested by humans and nonruminants[4]. The undigested InsP$_6$ not only reduces the bioavailability of essential mineral elements (e.g. Fe, Zn, and Ca) and amino acids in the digestive tract, but also is considered a leading source of phosphorus pollution from agriculture when excreted in animal waste[5,6]. Therefore, reducing InsP$_6$ content in seeds is one of the important breeding objectives in agriculture. Although efforts have

been made in some species, such as maize and rice[2,3], the very limited understanding on the regulatory mechanisms of InsP$_6$ biosynthetic pathway has hampered this breeding activity.

In the mature seeds, InsP$_6$ is stored and organized in globoids[7]. During the seed germination stage, InsP$_6$ is hydrolyzed to release Pi and mineral elements, which provide nutrients and energy for the early growth of seedlings[4,6]. InsP$_6$ also plays an important role in the triggering of Ca$^{2+}$ signals, the auxin storage and transport, phosphatidylinositol signaling, cell wall synthesis and the production of secondary metabolites[1,3,4,8]. InsP$_6$ can be formed by two different pathways: a lipid-dependent pathway, where phospholipase C (PI-PLC) catalyzes

[1]State Key Laboratory of Plant Physiology and Biochemistry, College of Life Sciences, Zhejiang University, 310058 Hangzhou, China. [2]Guangdong Laboratory for Lingnan Modern Agriculture, College of Natural Resources and Environment, South China Agricultural University, 5100642 Guangzhou, China. [3]College of Agronomy and Biotechnology, Zhejiang University, 310058 Hangzhou, China. [4]Analysis Center of Agrobiology and Environmental Sciences, Zhejiang University, 310058 Hangzhou, China. [5]Ningbo Key Laboratory of Agricultural Germplasm Resources Mining and Environmental Regulation, College of Science and Technology, Ningbo University, 315300 Ningbo, China. [6]Université Paris-Saclay, CNRS, INRAE, Univ Evry, Institute of Plant Sciences Paris-Saclay (IPS2), 10 91405 Orsay, France. ✉e-mail: zjding@zju.edu.cn; sjzheng@zju.edu.cn

the hydrolysis of phosphatidylinositol 4,5-bisphosphate (PtdIns(4,5)$P_2$) into *myo*-inositol P3 (Ins(1,4,5)$P_3$ or InsP$_3$), which is then progressively phosphorylated to InsP$_6$[9–13]; and a lipid-independent pathway, where *myo*-inositol (3) P1 (Ins(3)$P_1$) undergoes a series of inositol kinases, including *myo*-inositol kinase (MIK), ITPKs and IPK1[14–17].

*IPK1* (an inositol polyphosphate kinase) has been cloned in rice, soybean, maize, and *Arabidopsis*[18–21], which is mainly responsible for catalyzing the conversion of inositol pentaphosphate (InsP$_5$) to InsP$_6$. Not only was the InsP$_6$ concentration decreased by ~80% in *atipk1-1* loss-of-function mutant versus WT, the concentration of InsP$_7$ and InsP$_8$ were also reduced, with the Pi concentration obviously increased[21]. In addition, IPK1 presumably cooperates with IPK2 (including IPK2$\alpha$ and IPK2$\beta$ that potentially harbor 6-/3-kinase activity and may sequentially phosphorylate Ins(1,4,5)$P_3$ to generate InsP$_5$ via an Ins(1,3,4,6)$P_4$ intermediate) and or ITPKs to convert the PI-PLC-generated or the glycolysis-derived InsP$_3$ into InsP$_6$ by phosphorylation[12,14,22,23]. It was found that more strongly phosphorylated species exist in the InsP$_6$-derived inositol pyrophosphates InsP$_7$ and InsP$_8$ through di-phosphoinositol-pentakisphosphate kinases VIH1/2 (named after VIP1 homologs), lack of which leads to constitutive Pi starvation response and impaired plant growth and development[24–28]. Furthermore, studies in rice and *Arabidopsis* showed that AtSPX1/OsSPX4 may be a cellular sensor for InsP$_8$, and that InsP$_8$ acts as a 'signaling translator' to reflect cellular Pi levels. These findings also reveal that Pi homeostasis in plants is regulated by kinases involved in InsP synthesis, most likely due to their indirect contribution to the synthesis of InsP$_8$[23–25,29–34]. Despite the importance of InsPs and their biosynthesis in plant, how InsP (particularly InsP$_6$) biosynthetic pathway is regulated, and how the enzymes or kinases (e.g. IPK1, IPK2, PI-PLC) involved are coordinated, remain elusive.

Receptor-like cytoplasmic kinases (RLCKs), which lack extracellular ligand-binding domains, have emerged as a major class of signaling proteins that regulate plant cellular activities in transmembrane signaling in response to biotic and abiotic stresses[35]. The pathways activated by receptor-like kinases (RLKs) and receptor-like proteins (RLPs) lead to the phosphorylation of RLCKs, which relay specific intracellular outputs through phosphorylation and activation of signaling components, including MAPK cascades, ROS production, cytosolic calcium (Ca$^{2+}$) influx, and so on[36–38]. In addition, some RLCKs are induced to express differentially under cold, salt, and dehydration conditions, such as *Arabidopsis* RLCK *CALMODULIN-BINDING RECEPTOR-LIKE CYTOPLASMIC KINASE1* (*CRLK1*)[39], *OsRLCK253*[40], rice RLCK *GROWTH UNDER DROUGHT KINASE* (*GUDK*)[41], *COLD-RESPONSIVE PROTEIN KINASE1* (*CRPK1*)[42]. Although the mechanisms for RLKs, RLPs and RLCKs to sense abiotic/biotic stress signals have been established, there are still many gaps in our understanding how RLCKs regulate plant growth, development and stress response. Identifying the substrates of RLCKs is a prerequisite for the better understanding of RLCKs functions on membrane processes in plants.

In this study, we performed a quantitative phosphoproteomics to identify potential regulators in Pi homeostasis. We found that the phosphorylation levels of two kinases belonging to RLCK V subfamily were inhibited by Pi deficiency challenge. Till now, the RLCK V subfamily has not been documented in terms of functions. Here we showed that these two kinases together with other related RLCK V subfamily members redundantly functioned in inositol polyphosphates accumulation. We therefore named them Inositol Polyphosphate-related Cytoplasmic Kinases 1-6 (IPCK1-IPCK6). Our findings show that IPCKs recruit IPK1, IPK2s and PI-PLCs via phosphorylation of GRF 14-3-3 proteins to modulate InsP$_6$ synthesis, revealing a previously unknown mechanism in the regulation of InsP accumulation in eukaryotes.

## Results

### A clade of RLCK V subfamily kinases involved in InsP$_6$ accumulation

We initially employed quantitative phosphoproteomics using wild-type (WT) plants transferred from control (1.25 mM Pi) to Pi deficiency (10 μM) condition for 1 h, to identify potential regulators in sensing extracellular Pi levels or in Pi homeostasis. We found that the phosphorylation of two kinases (here named IPCK1 and IPCK2) belonging RLCK V subfamily, were inhibited by Pi deficiency treatment (Supplementary Fig. 1a). There are 11 members in the RLCK V subfamily, six of which are closely related including IPCK1 and IPCK2 (Fig. 1a). The six RLCKs all have a transmembrane domain (Fig. 1b), making them expressed on the plasma membrane (Fig. 1c). We further showed that they were expressed in seedlings, leaves, inflorescences, siliques and embryos (Fig. 1d). These similar expression patterns indicate probable function redundancy of IPCKs. In line with this, we found that the single T-DNA insertion mutants (*ipck1*, *ipck2*, *ipck3* and *ipck4*) had no visible difference with WT in Pi deficiency responses, Pi homeostasis and Pi-related gene expression (Supplementary Fig. 1b–g), nor the double (*ipck1 ipck2*) and triple (*ipck1 ipck2 ipck3*) mutants (Supplementary Fig. 1e–g).

We next generated the *ipck1 ipck2 ipck3 ipck4* quadruple mutants either by crossing (named *T-4 m*) or by CRISPR (named *C-4m*). As the expression of *IPCK5*, but not of *IPCK6*, was substantially up-regulated in *T-4m* (Supplementary Fig. 1h), we further generated the quintuple mutant (*ipck1-ipck5*, named *C-5m*) by knocking out *IPCK5* in the *T-4m* genetic background. Similarly, the expression of *IPCK6* was substantially up-regulated in *C-5m* (Supplementary Fig. 1h), but we actually failed to obtain the sextuple mutant, which is lethal during embryo development (Supplementary Fig. 1i), implying the indispensable role of IPCKs in plant growth. We found that both the quadruple (*T-4m/C-4m*) and quintuple (*C-5m*) mutants showed WT-like responses to Pi deficiency (Supplementary Fig. 1j). Nevertheless, the Pi concentration, particularly in seeds, was substantially increased in *T-4m/C-4m* versus WT, and was more in *C-5m* than in *T-4m/C-4m* (Fig. 2a). We also found that the total P concentration was only mildly increased in seeds of *T-4m/C-4m* and *C-5m*, and was comparable in seedlings of WT and mutants (Fig. 2b), indicating their possible involvement in P homeostasis. Since InsP$_6$ is the major form of phosphorus in seeds, we next found that the InsP$_6$ concentration in seeds and seedlings was obviously reduced in *T-4m/C-4m* versus WT, and that the reduction was enhanced in *C-5m* (Fig. 2c–e). The reduced accumulation of InsP$_6$ in *C-5m* was further confirmed by HPLC-MS/MS assay (Fig. 2f). Both the increase of Pi accumulation and the decrease of InsP$_6$ concentration in *T-4m* could be restored by *IPCK1* or *IPCK2* (Figs. 2a, d). These results suggest that IPCKs may be involved in InsP$_6$ synthesis.

To determine if lack of IPCKs affects the accumulation of other inositol polyphosphates, we used HPLC-MS/MS to measure the concentration of InsPs in seedlings, and found that both the concentration of InsP$_5$/InsP$_6$ were significantly reduced in *C-5m*, with InsP$_5$ concentration much lower than InsP$_6$, while the concentration of InsP$_3$ showed a mild decrease in *C-5m* seedlings (Supplementary Fig. 2). We also observed a decrease of InsP$_8$ concentration in *C-5m* versus WT (Supplementary Fig. 2). As InsP$_8$ acts as a 'signaling translator' of Pi levels to repress the expression of Pi starvation-induced (PSI) genes (e.g. *IPS1*, *SPX1* and *PHT1;1*)[25], we further showed that the expression of these PSI genes was mildly increased in the *T-4m/C-4m* and *C-5m* mutants (Fig. 2g). Taken together, these data indicate that IPCKs function redundantly in InsP$_6$ accumulation.

### IPCKs regulate InsP$_6$ accumulation dependently on IPK1

To determine how IPCKs contribute to InsP$_6$ accumulation, we screened IPCK1 interacting proteins through a split-ubiquitin

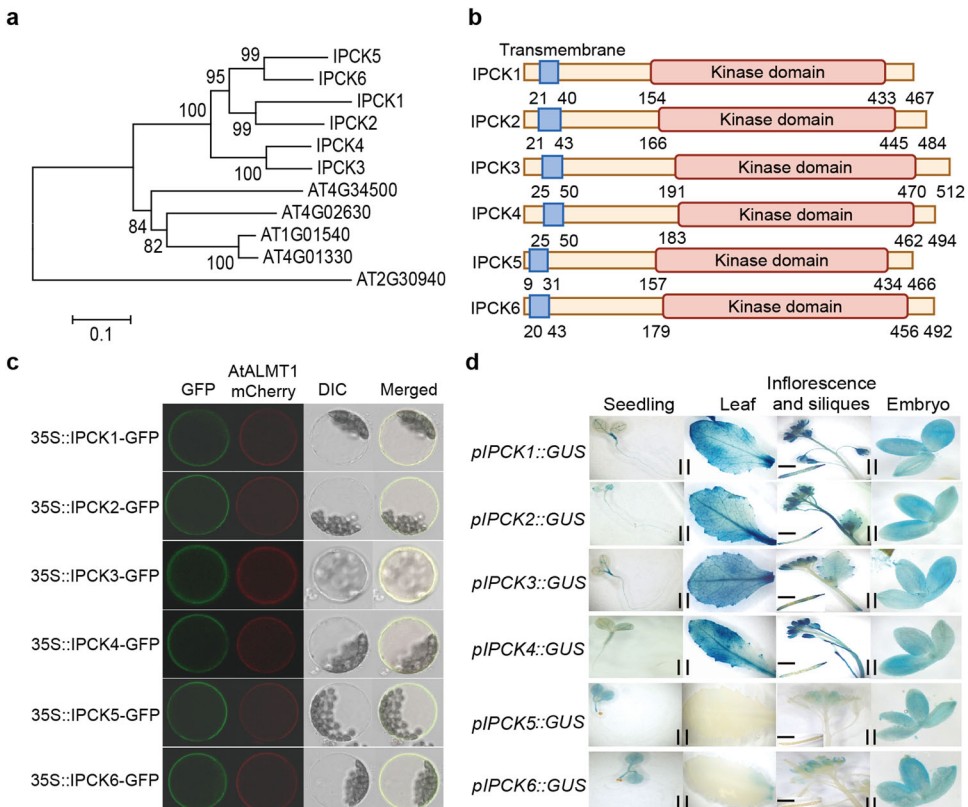

**Fig. 1 | Expression patterns of IPCKs. a** Phylogenetic tree analysis of the RLCK V subfamily. **b** Structural diagram of the RLCK V subfamily protein. The blue box represents transmembrane domain and the red box represents kinase domain. **c** Expression and localization of IPCK(1-6)-GFP fusions in wild-type (WT) mesophyll protoplasts. AtALMT1-mCherry RFP fluorescence was used as a plasma membrane marker. Columns 1–4 indicate GFP signals, bright-field differential interference contrast (DIC), and merged images of GFP and RFP, respectively. Bar = 10 μm. Three independent repeats were done with similar results. **d** *pIPCK(1-6)::GUS* reporter gene expression in different organs. Columns 1–4 indicate seedling (bar = 5 mm), leaf (bar = 5 mm), Inflorescence and siliques (bar = 5 mm), Embryo (bar = 0.5 mm).

membrane yeast two-hybrid system (Supplementary Dataset 1), finding IPK1, the key synthase of InsP$_6$, to be a good candidate. We next confirmed the interaction between IPCK1 and IPK1 using BiFC analysis, and found their interaction mainly occurring on the plasma membrane (Fig. 3a). The subcellular localization of IPK1 was primarily found in the cytosol (Supplementary Fig. 3a). We showed that IPK1 also interacted with other IPCKs (Fig. 3a). Additionally, the in vitro pull-down assay and tobacco Split-LUC imaging experiments further validated that IPCK1 interacts with IPK1 (Figs. 3b, c). To ascertain which part of IPCK1 is critical for the interaction with IPK1, we did BiFC and Split-LUC analysis using truncated IPCK1, and found that the N terminal (1-153 amino acids), rather than the kinase domain, of IPCK1 was required for interaction with IPK1, although the interaction took place in the unknown foci next to the nucleus (Supplementary Fig. 3b–e).

Because IPK1 plays a key role in synthesizing InsP$_6$, we observed that the InsP$_6$ concentration was remarkably reduced in *ipk1* in both seeds and seedlings, compared with that in WT and *T-4m* (Fig. 3d). Conversely, the Pi concentration was significantly increased in *ipk1* (Fig. 3e), and the total P concentration was increased as well, but in a much milder way (Fig. 3f). Furthermore, we showed that over-expression of *IPK1* was sufficient to restore the accumulation of InsP$_6$, Pi and total P of *T-4m* to that of WT, suggesting that *IPK1* is epistatic to *IPCKs* (Figs. 3d–f). Although *ipk1* mutant also showed no phenotype difference with WT under Pi deficient condition, it was revealed that the expression of many Pi starvation responsive (PSR) genes was increased in the *ipk1* mutant versus WT under Pi sufficient condition[21]. Our further transcriptome profiling showed that about 46.2% of the PSR genes differentially expressed in *ipk1* mutants (versus WT) were also affected in *C-5m* (SRA accession: PRJNA941157; Fig. 3g), suggesting

that *IPCKs* and *IPK1* likely act in the same pathway. In accord with this, the retarded growth of the *C-5m* mutants displayed a similarly retarded growth with the plants lacking IPK1 (Fig. 3h). Collectively, these data indicate that IPCKs regulating InsP$_6$ accumulation is IPK1 dependent.

### IPCK1 phosphorylates GRF4 to promote InsP$_6$ accumulation

To investigate how IPCKs regulate IPK1 function, we further showed that IPCK1 could not phosphorylate IPK1 in vitro (Supplementary Fig. 4a), which is consistent with the observation that the kinase domain of IPCK1 did not interact with IPK1 (Supplementary Fig. 3d). We next wondered if there are intermediate molecules or chaperones that help IPCK1 with the regulation of IPK1 activity. To this end, we performed immunoprecipitation-mass spectrometry (IP-MS) assay, and identified a class of GRF (General Regulatory Factors, also known as 14-3-3) proteins as IPCK1 potential interactants (Supplementary Dataset 2). Since GRFs have been shown to be able to directly bind and modulate the activity of target proteins after activation by upstream regulators[43–45], they appear to be the ideal candidate intermediating IPCK1-IPK1 regulation. Among the GRF proteins, GRF4 was found to be most abundant in IPCK1 immunoprecipitants. We then confirmed the interaction between IPCK1 and GRF4 using BiFC, Pull-down and Split-LUC assays (Fig. 4a–c). As expected, we also showed that GRF4 could interact with IPK1 (Figs. 4a–c), favoring its role in connecting IPCK1 and IPK1.

Furthermore, we found that IPCK1 could phosphorylate GRF4 in vitro (Fig. 4d), and that the phosphorylation occurred on Ser242 (S242) of GRF4 as identified by MS. Substitution of S242 to Ala (GRF4$^{S242A}$) blocked this phosphorylation (Fig. 4d). We also demonstrated that the kinase domain of IPCK1 is required for GRF4 phosphorylation, as a truncated IPCK1 (21-467 amino acids, IPCK1$^{21-467}$) including the kinase

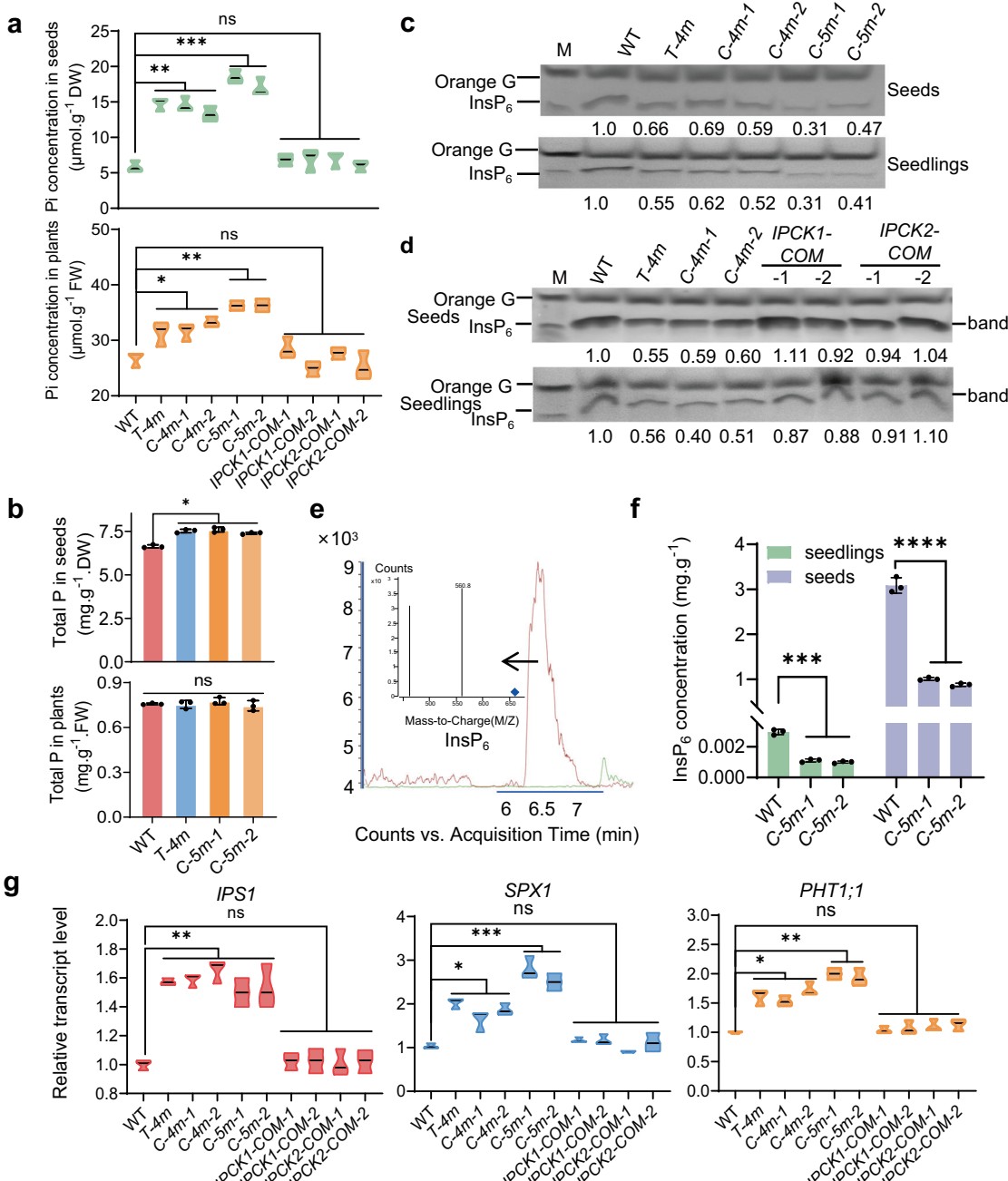

**Fig. 2 | IPCKs function redundantly in InsP6 accumulation. a** Pi concentration in dry seeds or seedlings grown on 1.25 mM Pi medium for 10 d. *IPCK1/2-COM* represent the *IPCK1/2* complementation lines of the *T-4m* mutant driven by the *IPCK1/2* native promoters (2087 bp for IPCK1 promoter, 2000 bp for IPCK2 promoter). Values are mean ± SD from four biological replicates. FW, fresh weight; DW, dry weight. **b** Total P concentration in seeds or 7-day-old seedlings. Values are mean ± SD from three biological replicates. **c, d** Isolation and SDS-PAGE analysis of InsP6. The independent experiment was repeated three times with similar result.

**e** HPLC-MS/MS showing the representative band 1 in (**d**). **f** The concentration of InsP6 in 2.4 g of dry seeds and 10 g of 12-day-old seedlings detected by HPLC-MS/ MS. Values are mean ± SD from three biological replicates. **g** Expression analysis of Phosphate Starvation-Induced (PSI) genes in 10-day-old seedlings grown on 1.25 mM Pi medium. Values are mean ± SD from three biological replicates. All experiments were repeated at least three times with similar results. All data were analyzed by unpaired *t*-test (ns indicates non-significant, *$P < 0.05$, **$P < 0.01$, ***$P < 0.001$, ****$P < 0.0001$).

domain could phosphorylate GRF4 in vitro, while a kinase dead mutation (IPCK1[K182E]) abolished this phosphorylation (Supplementary Fig. 4b). We next generated the antibodies specifically recognize the phosphorylated S242 of GRF4 (pS242), and confirmed their action in vitro (Fig. 4e). In order to verify the in vivo phosphorylation, we generated *35 S::GRF4-flag/WT* and *35 S::GRF4-flag/T-4m* transgenic lines, and found that the phosphorylation level of S242 of GRF4 was indeed reduced in the *T-4m* background compared with that in WT (Fig. 4f). Additionally, we showed that the S242 is highly conserved in

the GRF subfamily with the only except of GRF11, suggesting the importance of this site for GRF function (Supplementary Fig. 4c). Overall, these results indicate that IPCKs phosphorylate the S242 of GRF4.

The next question is whether GRF4 is involved in InsP6 accumulation. We showed that the InsP6 levels in WT and *grf4* loss-of-function mutant were comparable (Supplementary Fig. 4d), which may be attributed to the functional redundancy of GRFs. With GRF3 interacting strongly with IPCKs and being closely related to GRF4

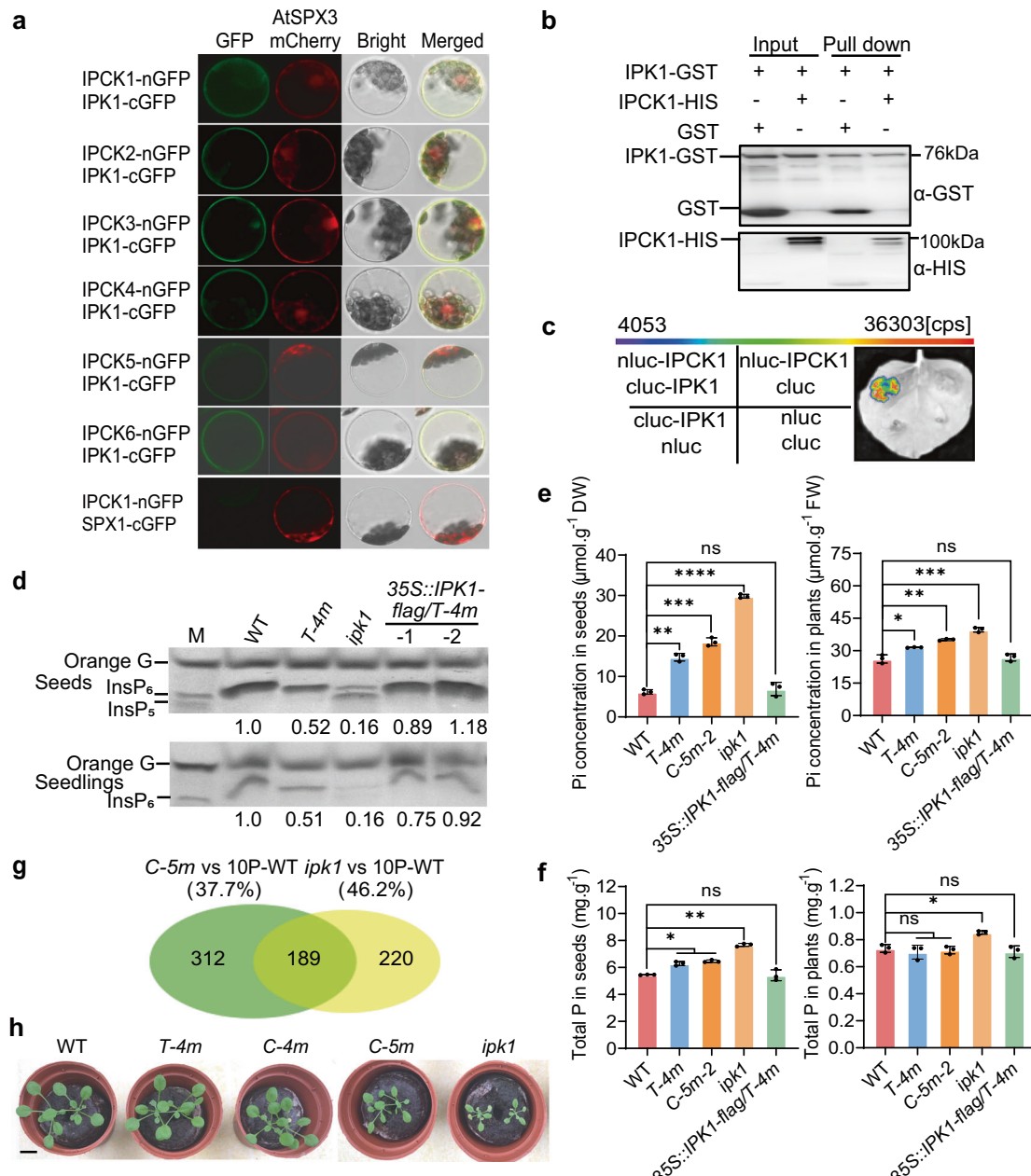

**Fig. 3 | IPCKs regulating InsP6 accumulation dependently of IPK1. a** BiFC assay showing the interaction between IPCK1-6 and IPK1 in protoplast. SPX1 was used as a negative control. AtSPX3-mCherry RFP fluorescence was used as colocalization marker of plasma membrane, cytoplasm and nucleus. Bar = 10 μm. **b** In vitro pull-down assay showing the interaction between IPCK1 and IPK1 **c** Split-LUC assay showing the interaction between IPCK1 and IPK1. Cps means the fluorescence value. **d** Isolation and SDS-PAGE analysis of InsP6 in seeds or seedlings. **e** Pi concentration in seeds or seedlings grown on 1.25 mM Pi medium for 10 days. The data of WT, *T-4m*, and *C-5m-2* are same as shown in Fig. 2a, which were obtained on the same batch of samples. Values are mean ± SD from four biological replicate. FW, fresh weight; DW, dry weight. **f** Total P concentration in seeds or 7-day-old seedlings. Values are mean ± SD from three biological replicates. **g** RNA-seq showing the Phosphate Starvation Responsive (PSR) genes in *C-5m* and *ipk1* mutant. The differentially expressed genes between *C-5m* and WT or *ipk1* and WT under normal condition were compared with that in WT under low Pi (10 μM Pi) condition versus normal condition. P < 0.01; fold change >=1.5. **h** Growth of indicated genotypes on soil for 25 days. Bar = 1 cm. All experiments were repeated at least three times with similar results. All data were analyzed by unpaired *t*-test (ns indicates non-significant, *P < 0.05, **P < 0.01, ***P < 0.001, ****P < 0.0001).

(Supplementary Fig. 4e; Supplementary Dataset 2), we generated the *grf3 grf4* double mutant, and found that the InsP6 concentration was notably reduced in *grf3 grf4* versus WT (Fig. 4g), but not in the *grf3* single mutant (Supplementary Fig. 4d). We also showed that *GRF4*, rather than *GRF4^{S242A}*, was capable to rescue the InsP6 accumulation in the *grf3 grf4* double mutant (Fig. 4g), suggesting that the phosphorylation of S242 is of importance for GRF4 function in InsP6 accumulation. In addition, we found that overexpression of *GRF4*, but not of

*GRF4^{S242A}*, largely restored the InsP6 levels in the *T-4m* mutant (Fig. 4h), indicating that GRF4 is epistatic to IPCKs in InsP6 accumulation. Taken together, these results reveal that IPCK1 promotes InsP6 accumulation via phosphorylation of S242 of GRF4.

### IPCK1 facilitates IPK1 activity by phosphorylating GRF4
Given that GRF4 interacts with IPK1 and is involved in InsP6 accumulation (Figs. 4a–h), to clarify the role of S242 phosphorylation of GRF4

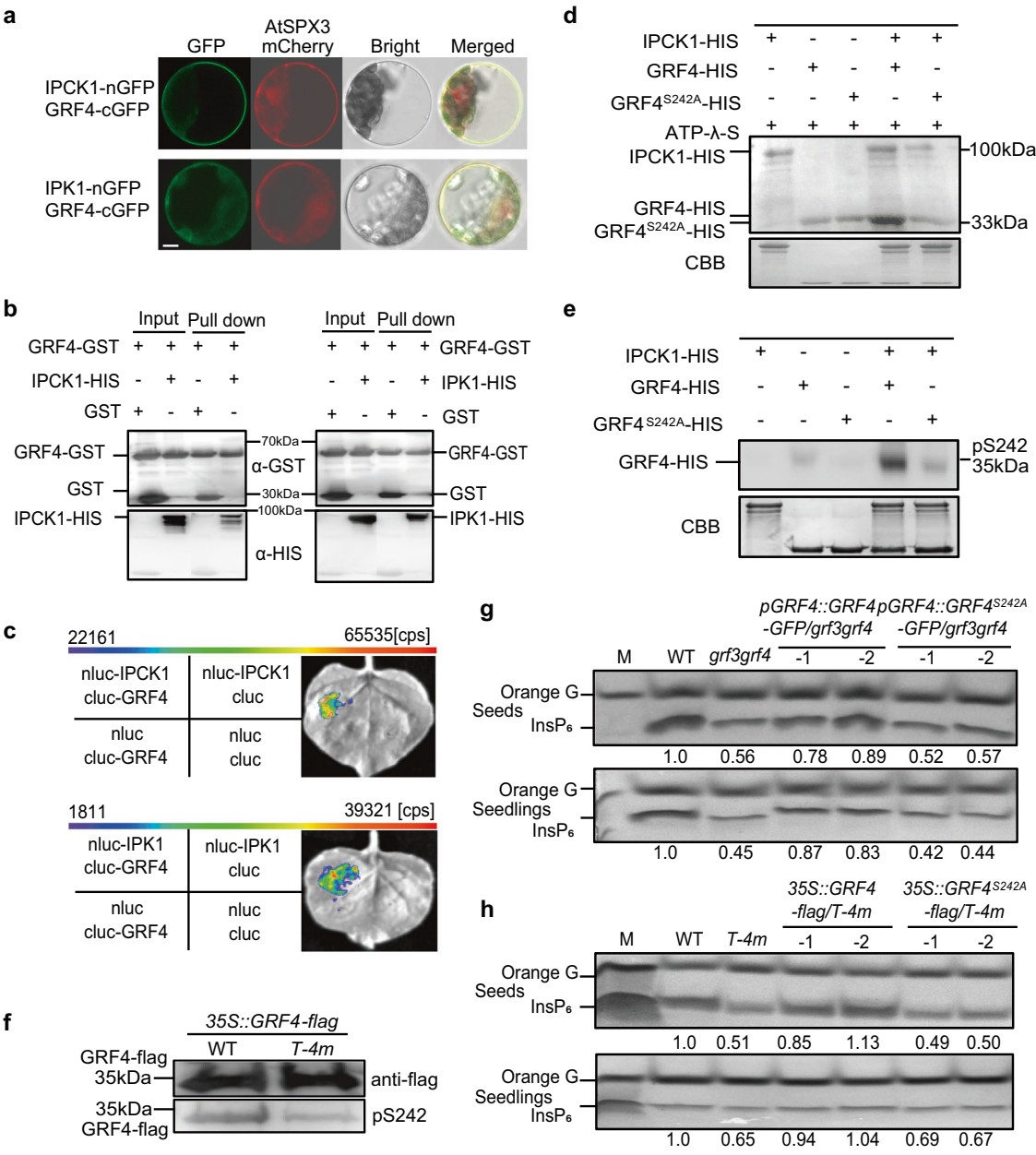

**Fig. 4 | IPCK1 phosphorylates GRF4 to promote InsP$_6$ accumulation. a** BiFC showing the interaction between IPCK1/IPK1 and GRF4 in protoplast. Bar = 10 μm. **b** In vitro pull-down assay showing the interaction between IPCK1/IPK1 and GRF4. **c** Split-LUC assay showing the interaction between IPCK1/IPK1 and GRF4. Cps indicates the fluorescence value. **d** In vitro phosphorylation assay showing that IPCK1 phosphorylates GRF4 but not GRF4$^{S242A}$. CBB, Coomassie brilliant blue. **e** In vitro phosphorylation assay showing that IPCK1 phosphorylates GRF4 but not GRF4$^{S242A}$, anti-pS242 antibodies were used for immunoblot analysis. **f** In vivo phosphorylation level of GRF4 in WT and *T-4m* seedlings. FLAG-GRF4 was extracted from 7-day-old seedlings expressing *35 S::GRF4-flag* in WT and *T-4m* backgrounds, and was analyzed by immunoblotting with anti-flag and anti-pS242 antibodies. **g**, **h** Isolation and SDS-PAGE analysis of InsP$_6$ in indicated genotypes. All experiments were repeated at least three times with similar results.

in IPCK1-IPK1 regulation, we next compared the interaction of WT (GRF4) and mutated (GRF4$^{S242A}$, or phosphomimicking GRF4$^{S242D}$) GRF4 with IPK1 in the Split-LUC assay, finding that the phosphorylation of S242 is required for the interaction between IPK1 and GRF4 (Fig. 5a). To further determine if the binding of IPCK1 or GRF4 affects the IPK1 activity, we performed an in vitro InsP$_6$ synthesis assay using InsP$_5$ (2OH-InsP$_5$) as the substrate (Supplementary Fig. 5). We showed that IPK1 had the highest activity in the presence of both IPCK1 and GRF4, or IPCK1 and GRF4$^{S242D}$, but had a low activity when IPCK1 and GRF4$^{S242A}$ were present (Figs. 5b, c). In addition, GRF4$^{S242D}$ alone was more effective than GRF4 in promoting IPK1 activity (Figs. 5b, c). Our further genetic analysis showed that overexpression of *GRF4* or *GRF4$^{S242A}$* could

not restore the InsP$_6$ levels in the *ipk1* mutant, favoring that *IPK1* is epistatic to *GRF4* in InsP$_6$ accumulation (Fig. 5d). Collectively, these results indicate that IPCK1 may regulate IPK1 activity and InsP$_6$ synthesis via formation of a IPCK1-GRF4-IPK1 complex, and that the complex formation depends on the phosphorylation of GRF4 conferred by IPCK1.

**IPCKs recruit IPK2s and PI-PLCs to regulate InsP$_6$ accumulation**
In addition to IPK1, we also found that IPK2s (including IPK2α and IPK2β), the key rate-limiting enzyme for InsP$_4$/InsP$_5$ synthesis, could interact with IPCK1 as revealed either by yeast-two hybrid screening or by IP-MS (Supplementary Dataset 1, 2). This interaction was confirmed

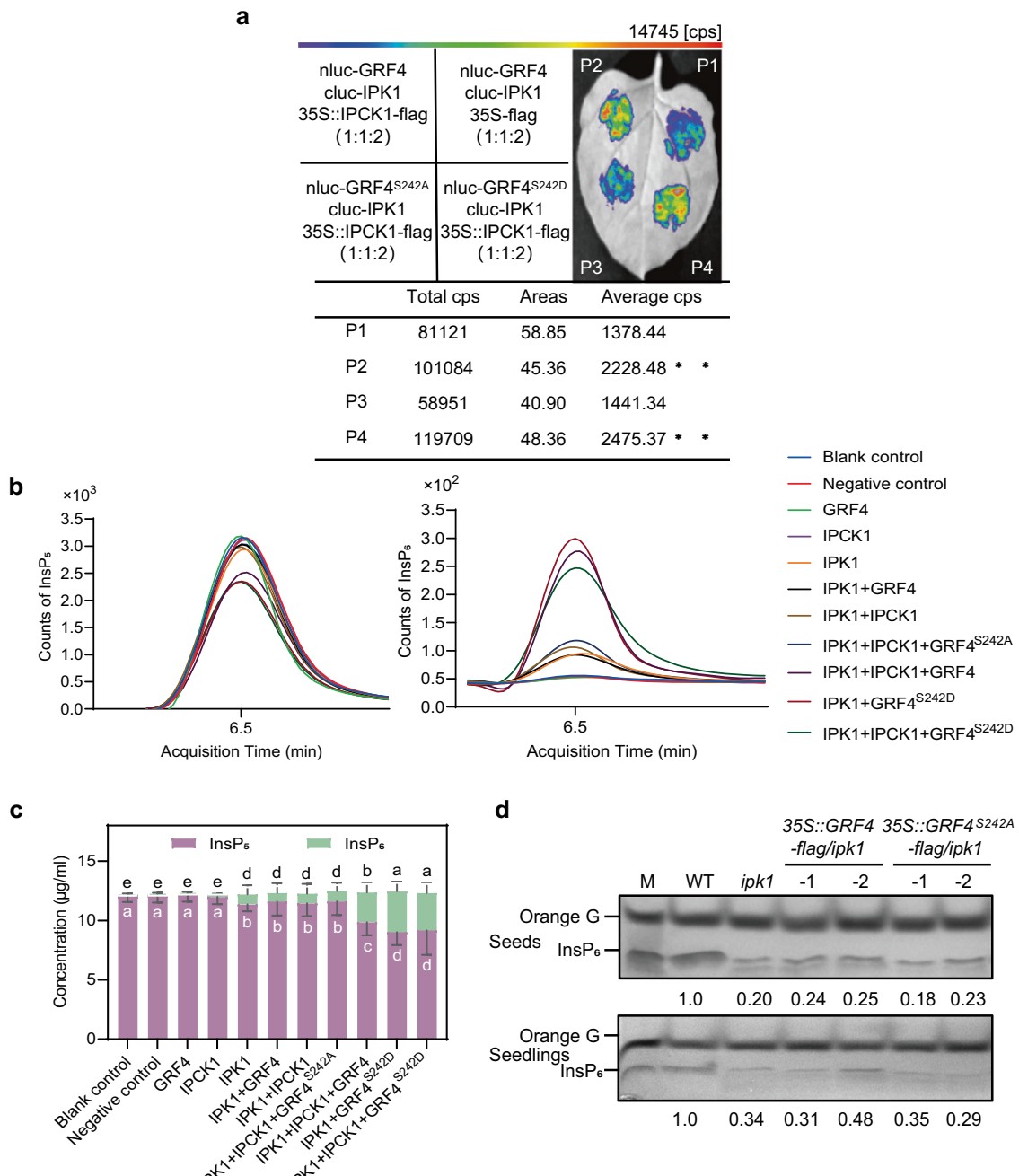

**Fig. 5 | IPCK1 facilitates IPK1 activity by phosphorylating GRF4. a** Split-LUC assay showing the effect of IPCK1 on the interaction between IPK1 and GRF4 or mutated GRF4. Values are given as mean ± SD from three biological replicates. Data were analyzed by unpaired $t$ test. **b, c** in vitro IPK1 activity analysis using InsP$_5$ as the substrate and InsP$_6$ as the product detected with HPLC-MS/MS assay. **b** Chromatogram absorption diagram. The abscissa represents the sample retention time, and the ordinate represents the peak area of the sample chromatogram.

Different color absorption peaks and areas correspond to various protein combinations. ddH$_2$O and 2.5 µg of PSKR1 were used as the blank control and negative control, respectively. **c** The concentration of InsP$_5$/InsP$_6$ converted from absorption peak area in (**b**). Values are mean ± SD from three biological replicates. Data were analyzed by unpaired $t$-test (different letters indicate significant difference, $P < 0.05$). **d** Isolation and SDS-PAGE analysis of InsP$_6$. All experiments were repeated at least three times with similar results.

by BiFC, pull-down and Split-LUC assays (Fig. 6a–c). Similar with IPK1, IPK2s also interact with other IPCKs (Fig. 6a), and the representative IPK2β interacts with the N-terminal (1-153 aa) of IPCK1 (Figs. 6d, e). Furthermore, we showed that the phosphorylation of GRF4 promoted its binding to IPK2β (Fig. 6f). With InsP$_5$ as the synthetic source of InsP$_6$, we found that InsP$_6$ concentration was indeed reduced in *ipk2β* loss-of-function mutant, while the Pi concentration was obviously increased (Figs. 6g, h). Moreover, overexpression of *IPK2β* could largely restore the Pi and InsP$_6$ concentration of *T-4m* to that of WT (Figs. 6g, h),

suggesting that *IPK2β* is epistatic to *IPCKs* in InsP$_6$ accumulation. Together with the observation that lack of IPCKs markedly reduces the InsP$_5$ levels in seedlings (Supplementary Fig. 2b), these results indicate that IPCKs are also involved in InsP$_5$ accumulation probably via regulation of IPK2 activity.

As IPCKs are localized on plasma membrane, and InsP$_5$/InsP$_6$ can be produced by a lipid-dependent pathway in which PI-PLC-generated InsP$_3$ from plasma membrane is used as the substrate for IPK2-mediated InsP$_4$/InsP$_5$ formation, we asked if IPCKs regulate PI-PLC

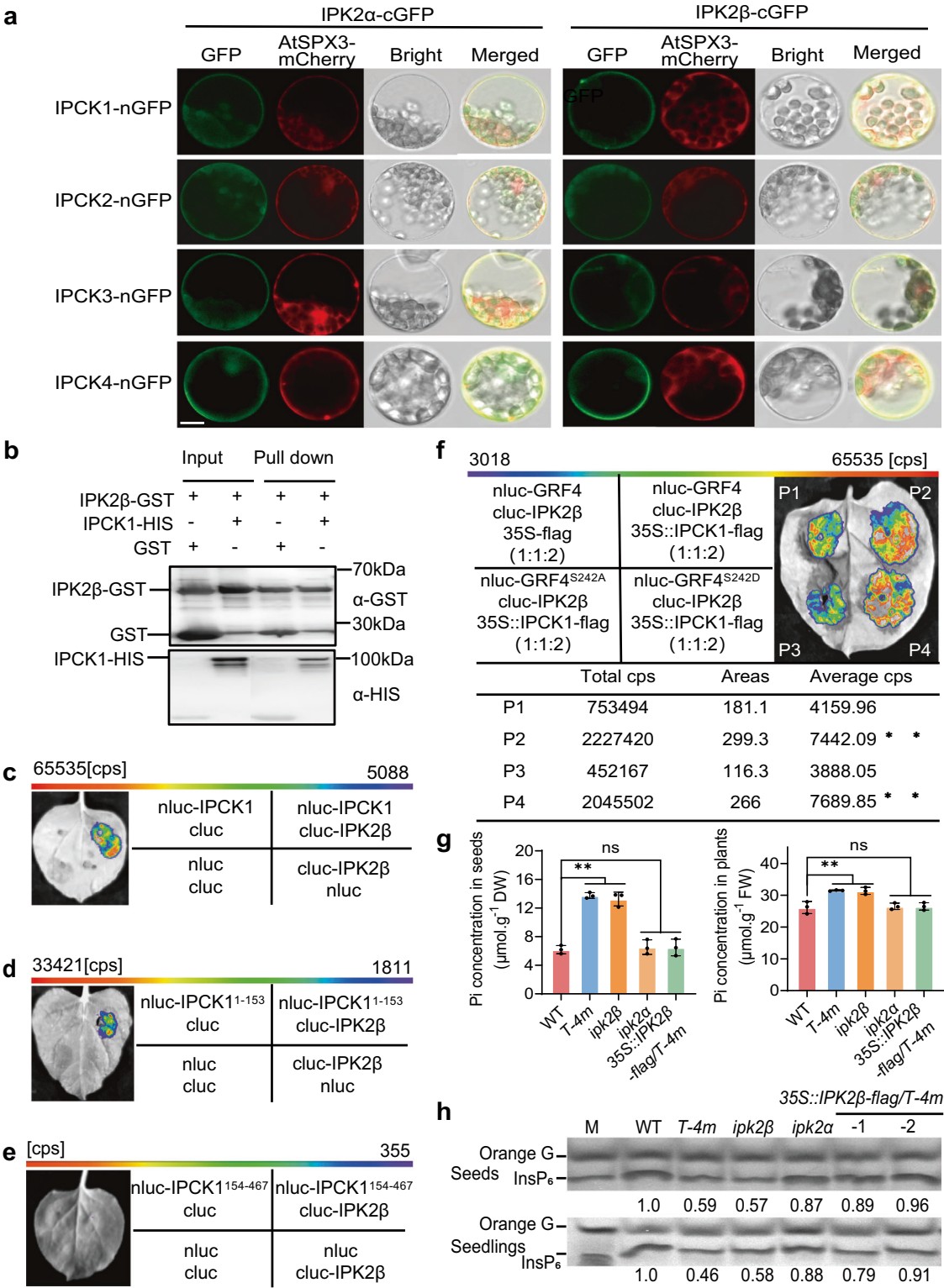

**Fig. 6 | IPCKs recruit IPK2s to regulate InsP$_6$ accumulation. a** BiFC assay showing the interaction between IPCK1-4 and IPK2α/IPK2β in protoplast. Bar = 10 μm. **b** In vitro pull-down assay showing the interaction between IPCK1 and IPK2β. **c, d, e** Split-LUC assay showing the interaction of the intact or truncated IPCK1 with IPK2β. Cps means the fluorescence value. **f** Split-LUC assay showing the effect of IPCK1 on the interaction between IPK2β and GRF4 or mutated GRF4. Values are given as mean ± SD from three biological replicates. **g** Pi concentration in dry seeds or seedlings grown on 1.25 mM Pi medium. Values are mean ± SD from four biological replicates. FW, fresh weight; DW, dry weight. **h** Isolation and SDS-PAGE analysis of InsP$_6$. All experiments were repeated at least three times with similar results. All data were analyzed by unpaired *t*-test (ns indicates non-significant, **$P$ < 0.01).

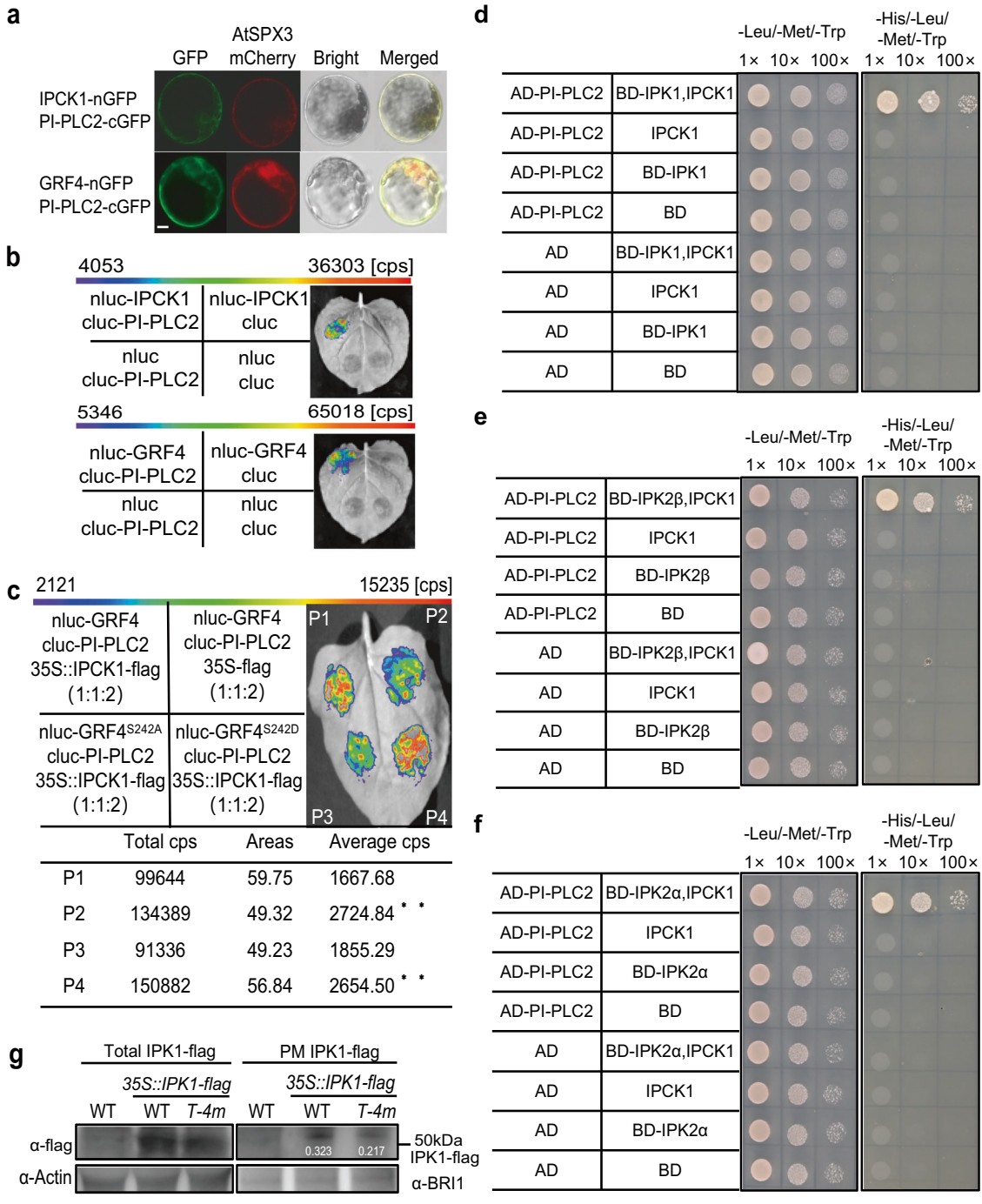

**Fig. 7 | IPCKs recruit IPK1/2s and PI-PLCs to regulate InsP₆ accumulation. a** BiFC assay showing the interaction between IPCK1/GRF4 and PI-PLC2 in protoplast. Bar = 10 μm. **b** Split-LUC assay showing the interaction between PI-PLC2 and IPCK1/GRF4. Cps means the fluorescence value. **c** Split-LUC assay showing the effect of IPCK1 on the interaction between PI-PLC2 and GRF4 or mutated GRF4. Values are given as mean ± SD from three biological replicates. Data were analyzed by unpaired *t* test (**P < 0.01). **d**–**f** Yeast three-hybrid assay showing that IPCK1 recruits PI-PLC2 and IPK1/IPK2s to form a potential complex. **g** Detection of IPK1 protein abundance on the plasma membrane (PM) under WT and *T-4m* backgrounds. Actin and BRI1 serve as internal references for total protein and plasma membrane protein, respectively. WT was used as a negative control. The number in the panel represents the relative proportion of abundance of PM-associated IPK1 proteins to IPK1 total proteins, which are corrected by internal references, respectively. All experiments were repeated at least three times with similar results.

activity as well. Indeed, we found that PI-PLC2 which belongs to PI-PLCs subfamily and potentially functions in InsP₃ synthesis[46,47], was a potential interactant of IPCK1 (Supplementary Dataset 2). The interaction between IPCK1/GRF4 and PI-PLC2 was verified by BiFC and Split-LUC assays (Fig. 7a, b). Additionally, we proved that PI-PLC7, which has the highest homology to PI-PLC2 protein, interacts with IPCK1/2 and GRF4 as well (Supplementary Fig. 6a). Although IPCK1 was unable to phosphorylate PI-PLC2 in vitro (Supplementary Fig. 6b), it could

recruit the binding of GRF4 to PI-PLC2 via phosphorylation of GRF4 S242 residue, to form a IPCK1-GRF4-PI-PLC2 complex as did for IPK1 and IPK2β (Fig. 7c), suggesting that IPCK1 probably regulates PI-PLC2 activity. In line with this, InsP₃ concentration was decreased significantly in plants lacking IPCKs (Supplementary Fig. 2b).

An efficient biosynthetic manner depends to some extent on the spatial distance between synthetases in the cell. We found that IPK1 could interact with IPK2s in vivo (Supplementary Fig. 6c), implying that

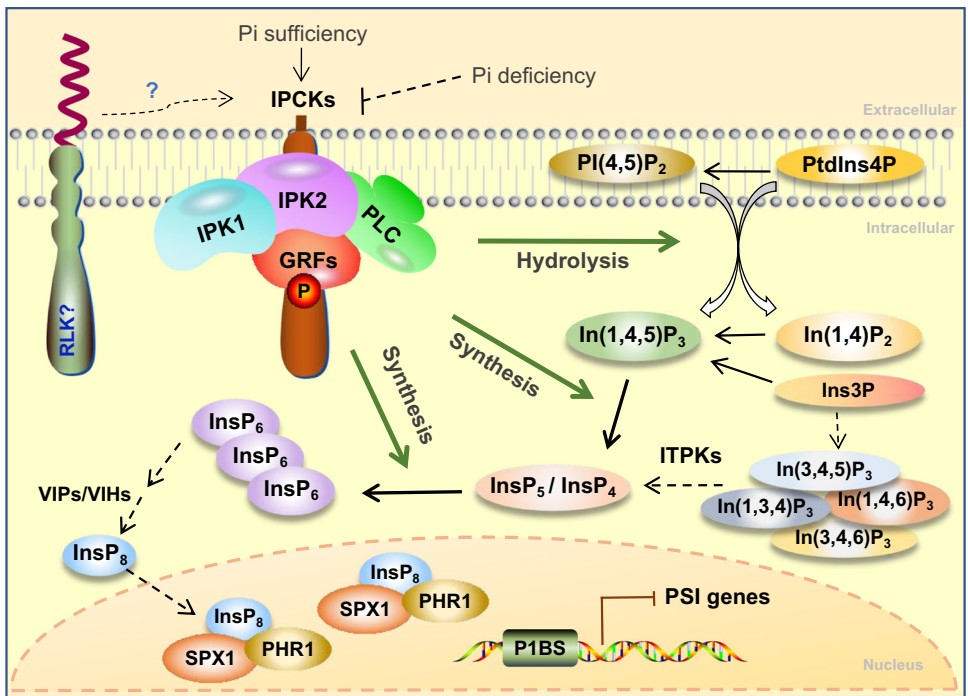

**Fig. 8 | The working model of IPCKs in InsP6 synthesis.** IPCKs interact with part of IPK1, IPK2s, and PI-PLCs pools, facilitating the activity of these enzymes, via phosphorylation of GRF 14-3-3 proteins. The recruited enzymes form a potential plasma membrane-located complex, therefore establishing a spatially efficient InsP6 synthetic manner. The altered concentration of InsP6 may affect the levels of InsP8, in turn affecting the expression of PSR genes. Pi deficiency treatment may inhibit the activity of IPCKs via reducing the phosphorylation levels on specific Ser residues, and hence repress InsP6 biosynthesis. "→" indicates the process that IPCKs participate in, "⇢" indicates the process that may be indirectly affected by IPCKs, "⊥" indicates inhibition.

the synthesis of InsP4/InsP5 and InsP6 are likely to be coupled. However, PI-PLC2 could not directly interact with IPK2s and IPK1. Instead, a yeast three-hybrid assay showed that PI-PLC2 could interact with IPK1/2 in the presence of IPCK1 (Figs. 7d–f). These data suggest that IPCKs may recruit IPK1/2 and PI-PLC2 to form a complex to the plasma membrane. To verify this hypothesis, we further detected the IPK1-flag protein abundance on the plasma membrane in both WT and *T-4m* backgrounds, finding that the relative abundance of plasma membrane-associated IPK1 is reduced (by >30%) in *T-4m* versus WT (Fig. 7g). Additionally, we also revealed that these genes have spatial co-expression in many tissues of plant[23,48], including the developing seeds (Fig. 1d; Supplementary Fig. 7). These results collectively suggest that IPCKs likely recruit IPK1, IPK2s and PI-PLC to form a complex to regulate InsP6 accumulation.

## Discussion

### IPCKs plays an important role in InsP6 accumulation

InsP6 serves as the phosphorus reservoir in seeds[2], and is also considered as a potential signaling molecule during plant development[49]. The synthesis of InsP6 and other InsPs are conferred by evolutionarily conserved enzymes[50]. For instance, the lipid-dependent InsP6 biosynthetic pathway is predominantly mediated by PI-PLCs, IPK2s, and IPK1 in Arabidopsis[51]. Nevertheless, how InsP6 synthesis is regulated, and how the function of the involved enzymes is coordinated, remain poorly understood. In this study, we identified a clade of RLCKs (named IPCKs) that are of importance for InsP6 synthesis and accumulation. On the one hand, IPCKs facilitate the activity of IPK1, IPK2s and PI-PLCs by interaction with them respectively via phosphorylation of GRF4 14-3-3 protein. On the other hand, IPCKs recruit these enzymes to the plasma membrane to form a potential complex, therefore likely establishing a spatially efficient InsP6 synthetic manner (Fig. 8). In this case, IPCKs function as the organizers to modulate and orchestrate the function of the enzymes

involved in InsPs synthesis, hence uncovering a previously unknown mechanism of InsPs biosynthesis in eukaryotes.

The model of IPCKs-recruited complex is supported by in vitro and in vivo evidence (Fig. 7). Importantly, we observed that the abundance of plasma membrane-associated IPK1 was reduced in *ipck* quadruple mutant versus WT (Fig. 7g). This reduction was indeed underestimated, since *IPK1-flag* used in this assay was driven by *35S* promoter that has a much broader expression than *PICKs* in WT. This IPCKs-governed InsP6 synthetic manner is reasonable and necessary for cell function, because the levels of some intermediate InsPs, such as InsP3, InsP4 and InsP5, must be strictly controlled. InsP3 and InsP4 are well-known second messengers that regulate cytosolic $Ca^{2+}$ concentration in animal cells[52,53]. In organisms from yeast to humans, IPMK (Inositol phlyphosphate multikinase) and IP3 3-Kinase (IP3K) are responsible for generating inositol phosphates[53,54], and overexpression of the N-terminal of human IPMK in HEK293 cells selectively interrupts endogenous mTOR-raptor interactions and inhibits mTORC1 signaling[55]. In plants, although InsP3 is not considered as the signaling molecule as in animals[52–55], it can participate in ABA-induced $Ca^{2+}$ release in guard cells, SAL1-mediated $Ca^{2+}$ release in PIN protein polarity distribution, and auxin transport[56–58]. During seed germination, InsP3 is essential for auxin-regulated embryogenesis, and can also favorably regulate ABA signal transduction[59,60]. Whereas over-accumulation of InsP3/InsP4/InsP5 may activate pathways that are not supposed to be used under normal conditions, and thus impair cell homeostasis and function. The IPCKs-regulated InsP6 synthesis enables the PI-PLC-generated InsP3 to be rapidly transformed into InsP6, therefore minimizing the lifetime of InsP3/InsP4/InsP5 and safeguarding the cell function during InsP6 biosynthesis. This is important, particularly in the developing seeds where InsP6 is synthesized in large quantities. Additionally, we found that the expression of *IPCK6* was enhanced in *C-5m* quintuple mutant (Supplementary Fig. 1h), and that the sextuple mutant (knockout of *IPCK6* in *C-5m*) was lethal during

embryo development (Supplementary Fig. 1i), showing a similar phenotype to the *ipk2α ipk2β* double mutant[22]. These observations suggest that IPCKs are essential for InsP biosynthesis and thus successful embryo development.

In addition to acting as a key rate-limiting enzyme in InsP$_6$ synthesis, IPK1 together with IPK2 or PI-PLC also participate in other signaling pathways with different ways. Both mammalian and plant IPK1 have an important role in growth and development, as null mutation in mouse IPK1 leads to embryonic lethal and plant *atipk1* loss-of-function mutant has severe development defects, however, the underlying regulatory mechanism remains unknown[21,61]. Mammalian IP3 3-kinase/IP3K (IPK2 orthologs) can physically interact with and physiologically regulate Target of Rapamycin (TOR) and AMP-activated protein kinase (AMPK)[62]. Yeast Ipk2 (IP3K/Arg82) is involved in cellular mRNA export from the nucleus with Ipk1 and controls arginine-dependent gene expression[63]. Plant *AtIPK2* and *OsIPK2* play a role in gibberellin and auxin signaling pathway, respectively[64,65]. In addition, PI-PLCs were also reported to function in signal transduction. In mammalian cells, PI-PLCs interact with a range of regulators including G-proteins, tyrosine kinases and others, to provide basic mechanisms of PI-PLC activity regulation and coordination with other cell effectors[66,67]. In plants, it has been reported that the expression of most of the *PI-PLC* genes are induced in response to diverse environmental stimuli such as *OsPLC1/3*, *AtPLC1/3/9*, *TaPLC1*, *GmPLC1* and so on[68,69]. Despite this progress, it is currently unknown how these enzymes (PI-PLC, IPK2, IPK1) are functionally coordinated in animals, plants, or microorganisms. The PI-PLC2-IPK2-IPK1 machinery governed by IPCKs may therefore provide insights into the mechanism of InsP biosynthesis in eukaryotes.

Although we propose that IPCKs recruit IPK1 and IPK2s to InsP$_6$ accumulation, not all of them are recruited to the plasma membrane, since they still have high expression in the cytosol[22] (Supplementary Fig. 3a). The free IPK1 and IPK2s possibly with lower activity may also to some extents contribute to the synthesis of InsPs. Together with the evidence that the *ipck* quadruple mutant (*T-4m*) is a mild mutant in InsP$_6$ reduction, in which IPCK5 and IPCK6 are still active and expression of *IPCK5* is even promoted (Supplementary Fig. 1h), these may be the reasons why overexpression of *IPK1/2β* could fully restore the decrease of InsP$_6$ in *T-4m* (Figs. 3d, 6h). Moreover, we cannot completely rule out the possibility that IPCKs may affect other pathway of InsP$_6$ biosynthesis, although we have verified that IPCKs do not interact with ITPK1 (Supplementary Fig. 6a). The inability to describe stereoisomers or enantiomers in this study does not allow us to discount the probable effect of IPCKs on InsP6 synthesis arising from ITPKs[14].

RLCKs often functionally and physically associate with receptor-like kinases (RLKs) to relay extracellular signals to the cytosol via phosphorylation events in regulating plant growth, development and immunity[35]. Our findings present a previously unknown role of RLCK as a scaffold-like protein, in addition to as a kinase, to regulate the activity of target enzymes. This may expand the understanding of how RLCK works in plant.

## IPCKs potentially mediate Pi-regulated InsP synthesis

IPCK1 and IPCK2 were initially identified because of their reduced phosphorylation levels under Pi deficiency challenge (Supplementary Fig. 1a). The altered phosphorylation occurred on the different Ser residues (S59 in IPCK1 and S105 in IPCK2). Substitution of these Ser to Ala abolished the in vitro phosphorylation of GRF4 conferred by IPCK1 and IPCK2 (Supplementary Fig. 8a), suggesting that the inhibition of phosphorylation of S59 in IPCK1 and S105 in IPCK2 may suppress their kinase activity. Since the phosphorylation of S242 promotes the interaction of GRF4 with IPK1/2/PLC2 (Figs. 5a, 6f, 7d), we hence speculate that the extracellular Pi status may influence the activity of IPCKs on GRF phosphorylation, thereby modulating InsP$_6$ accumulation and Pi homeostasis in the cytosol by either regulation of the

activity of IPK1/2/PLC2 or the organization of this complex. However, both S59 and S105 are not conserved among IPCKs (Supplementary Fig. 8b), implying a possibility that Pi deficiency may inhibit the phosphorylation of IPCKs on different sites, which needs to be determined in the future study. Overall, we propose that IPCKs may mediate InsPs accumulation in response to Pi levels. When Pi is sufficient, IPCKs organize an efficient InsP$_6$ synthesis manner to utilize or store Pi; under Pi starvation, the phosphorylation of IPCKs is inhibited, leading to a compromised level of InsP$_6$ synthesis (Fig. 8). This regulation is necessary, since plant needs to restrain InsP$_6$ synthesis to make the limited Pi available for more essential processes (e.g. photosynthesis) when exposed to Pi deficiency. Inhibiting the activity of IPCKs, the organizers of InsP$_6$ synthesis, is definitely an efficient strategy to rapidly brake InsPs synthesis. In addition, the phosphorylation levels of IPCKs are likely regulated by kinases and phosphatases. RLK may be a good candidate for sensing extracellular Pi levels and phosphorylating IPCKs to regulate cytosolic Pi homeostasis, which requires future investigation.

InsP$_8$ is a signaling translator that reflects cellular Pi levels, and is perceived by SPX proteins to regulate the Pi-starvation response (PSR)[24]. The expression of PSR genes was known to be affected in *ipk1* loss-of-function mutant under Pi sufficient condition, due to the reduced level of InsP$_6$ in the mutant[21]. We observed that the mRNA levels of many of these PSR genes were likewise altered in plants lacking IPCKs (Fig. 3g). Since IPCK1 does not interact with VIH1 (Supplementary Fig. 6a), the rate-limiting enzyme responsible for InsP$_8$ synthesis[24,25], we speculate that IPCKs may not directly participate in InsP$_8$ synthesis, but their loss of function reduces InsP$_6$ and InsP$_8$ accumulation (Fig. 2f, Supplementary Fig. 2b), thereby affecting the expression of PSR genes. Interestingly, we found that lack of IPCKs did not affect the concentration of InsP$_7$. We hypothesize that InsP7 may function as a transitional intermediate in InsP$_8$ synthesis in plants. Once synthesized, the InsP$_7$ might be rapidly converted to InsP$_8$ to maintain the phosphorus homeostasis. In line with this, the concentration of InsP$_7$ is much lower than that of InsP$_8$ in WT plants (Supplementary Fig. 2b). Alternatively, we cannot completely rule out the possibility that IPCKs may indirectly affect VIH1 expression or activity.

In conclusion, our study presents a previously uncharacterized role for the RLCK V subfamily as a critical organizer of the InsP$_6$ synthesis, providing insights into the mechanism of InsP synthesis in eukaryotes. Our findings may also provide clues for future molecular breeding of crops with low phytic acid in seed.

## Methods

### Material acquisition and plant growth conditions

Ecotype Columbia of *Arabidopsis thaliana* was used as WT in this study. T-DNA insertion mutants *ipck1* (SALK_047485C), *ipck2* (SAIL_916_B10), *ipck3* (SALK_026210C), *ipck4* (SAIL_913_F05), *ipk1* (SALK_065337C), *ipk2α* (SALK_206456C) and *ipk2β* (SALK_104995C), *grf3* (SALK_205814C) and *grf4* (SALK_088321C) were obtained from the Arabidopsis Biological Resource Center (http://abrc.osu.edu). The *ipck1 ipck2*, *ipck1 ipck2 ipck3*, *ipck1 ipck3 ipck4*, *T-4m* and *grf3 grf4* multiple mutants were generated by crossing. The homozygosity of these mutants was verified with PCR. *IPCK2* were knocked out with CRISPR method in the *ipck1 ipck3 ipck4* background to construct *C-4m*. *C-5m* was generated by knockout of *IPCK5* on the basis of *T-4m*. *pIPCK(1-6)::GUS* were constructed by cloning their promoter fragment (-2 k bp) into pCAMBIA1300-GUS vector, respectively. *IPCK1-COM*, *IPCK2-COM*, *pGRF4::GRF4-GFP* were generated by constructing their native promoter-driven cDNA in a modified binary expression vector pCAMBIA1300-GFP. *35 S::IPK1-flag*, *35 S::IPK2β-flag*, *35 S::IPCK1-flag*, *35 S::GRF4-flag*, *35 S::GRF4-flag* and *35 S::GRF4-flag* were generated by cloning their open reading frame into the vector of pcDNA3.1-3xFlag. The site-directed mutagenesis vectors of *pGRF4::GRF4$^{S242A}$-GFP*, *35 S::GRF4$^{S242A}$-flag* and *35 S::GRF4$^{S242A}$-flag* were mutated following the

introductions of Q5 site-directed mutation kit (NEB). All vectors were then transformed into the plants using agrobacterium tumefaciens strain GV3101. The primers used are listed in Supplementary Dataset 3. The medium was prepared with 1/2 Murashige and Skoog medium powder (Phytotech), supplemented with 0.5 % (w/v) sucrose, and 0.8% (w/v) agar (Sigma). The seeds were vernalized at 4 °C for 3 d, and then germinated on medium. Seedlings were grown in the growth chamber at 22–23 °C under a photoperiod of 16/8 h (light/dark).

## Subcellular localization analysis

CDSs of *IPCK1-6* were cloned into HBT95::GFP vector. The GFP fusion constructs were transformed into the protoplasts and imaged by a confocal microscopy (LSM710, Carl Zeiss). Arabidopsis protoplast preparation and transformation were as described previously[70]. In brief, 4-week-old young rosette-stage leaves were used for protoplast preparation. Leaves were first cut in 1–2 mm of strips, and incubated in the digesting solution at room temperature for 2–3 h. The protoplasts were released by gently swirling the solution, pelleted by centrifugation at low speed, and then resuspended in W5 buffer (154 mM NaCl, 125 mM $CaCl_2$, 5 mM KCl, 2 mM MES, pH 5.7). The centrifuge-wash process was done for another time, and the protoplasts were incubated in MMG buffer (15 mM $MgCl_2$, 0.1% MES, 0.4 M mannitol, pH 5.7) on ice for 1 h to allow them to recover. For protoplast transformation, 200 μl of protoplast suspension were incubated with 1–2 μg of plasmid DNA and 40% PEG for 5–7 min, and then were diluted with W5 buffer to stop the transformation process. The transformed protoplasts were kept at dime light for 16–18 h, and then imaged under confocal microscopy.

## Yeast experiments

A DUAL membrane yeast two-hybrid system (Dualsystems Biotech) was used to screen for the proteins interacting with IPCK1. The full CDS of *IPCK1* was cloned into the pBT3-STE vector, and then co-transformed together with X-pPR3-N vector constructed plasmid membrane library into the yeast strain NMY51. Yeast growth was shown on SC/-Trp/-Leu and SC/-Trp/-Leu/-His/-Ade medium (Takara). IPCK1-STE/pPR3-N were used as negative control. Yeast three-hybrid assays were performed using the AD and pBridge vectors. Plasmid-containing cells were acquired, characterized and further converted into AH109 competent cell. The interaction between the AD and pBridge probes was monitored on SC/-Leu/-Met/-Trp and SC/-His/-Leu/-Met/-Trp medium (Takara). The primers used are listed in Supplementary Dataset 3.

## Immunoblotting analysis

Total proteins of *35 S::IPCK1-flag/WT* transgenic lines grown on agar medium for 10 d were extracted, the detailed operating methods of immunoprecipitation and western blotting are described as previously[71]. In short, seedlings were ground in fine powder and lysed with RIPA buffer. The supernatant containing total proteins was collected after centrifugation at 12,000 *g* for 15 min, and then incubated with flag magnetic beads (MedChemExpress) for 2 h on ice to capture the antibody-protein complex. The beads were next washed several times with PBS buffer, and the proteins were eluted by adding loading buffer and heating at 98°C for 10 min. α-flag and α-actin (ABclonal) antibodies were used for Western Blot analysis.

## Bimolecular fluorescence complementation (BiFC) assay

BiFC assays were performed using Arabidopsis protoplasts as described above. Various BiFC constructs were transiently expressed in protoplast. Primers used are listed in Supplementary Dataset 3.

## Pull-down assay

The pull-down assay was performed as previously described[72]. Briefly, the CDS of *IPCK1*, *GRF4*, *IPK1* and *IPK2β* were cloned into

pCOLD-TF to generate HIS-tagged recombinant plasmids. The CDS of *IPCK1*, *GRF4* were cloned into pGEX-4T-1 to generate GST-tagged recombinant plasmids. Both the GST- and HIS-tagged proteins were expressed in *E. coli* strain BL21 (DE3) cells. The GST/Ni-NTA beads were used to obtain the bait proteins, and then incubated with the target proteins with different tag on ice for 2 h. The beads were next washed for several times with PBS buffer, and the proteins were eluted with loading buffer for Western Blot assay. The combination of IPCK1-HIS/IPCK1-GST and GRF4-HIS/GRF4-GST were used as positive control. IPCK1-HIS/GST and GRF4-HIS/GST were used as a negative control. α-HIS and α-GST (ABclonal) antibodies were used to detect the target proteins.

## Split-luciferase complementation (Split-LUC) assay

The cDNAs of genes were separately fused with the N- or C-terminal parts of the luciferase reporter gene *LUC* (*nLUC* and *cLUC*) to generate the *X-nLUC* or *X-cLUC* vectors. The site-directed mutagenesis vectors were mutated following the manual. The constructs were co-infiltrated into tobacco leaves, and the LUC activities were analyzed after infiltration with D-Luciferin (Thermo) for 48 h. The primers used are listed in Supplementary Dataset 3.

## Pi assay

Pi concentration was measured with the phosphomolybdate colorimetric assay as described by Jain et al.[73]. In brief, 0.02 g of seedlings or 0.01 g of seeds were grinded into fine powder with liquid nitrogen, and were immediately added into the inorganic phosphorus extraction solution (0.01 g per 100 ul volume), followed by the addition of 1% glacial acetic acid in a 1:9 ratio. The samples were then incubated in a 42 °C water bath for 30 min, followed by centrifugation at 12,000 *g* for 5 min. 150 μl of supernatant were next added into 350 μl of coloration, and were incubated in the 42 °C water bath for another 30 min. After the reaction is completed, the samples were kept at room temperature for 5 min. 200 μl of each sample was taken for absorbance measurement at 820 nm, and the same method was used to generate the calibration curve. The Pi concentration in the sample was calculated based on the calibration curve.

## Inductively coupled plasma mass spectrometry (ICP-MS) assay

0.02 g of dry seeds or 0.05 g of seedlings were collected for each sample and digested with 0.5 ml of concentrated nitric acid at 200 °C. After all samples were dissolved into a colorless and uniform liquid, they were further diluted to 5 ml with $ddH_2O$ and filtered to remove impurities. ICP-MS (Agilent 7500ce Santa Clara CA USA) was used for total P detection as described by Sun et al.[71].

## Isolation and SDS-PAGE/HPLC-MS/MS analysis of InsPs

The enrichment of InsPs with $TiO_2$ beads and SDS-PAGE assay were performed as described by Wilson et al.[74]. In brief, 1 g total seedlings grown on agar medium for 15 d or 0.1 g dry seeds were ground in liquid nitrogen, suspended in 5 ml of 1 M cold perchloric acid, and kept rotating for 15 min at 4 °C. After centrifugation at 12,000 g for 10 min at 4 °C, the supernatant was transferred into a new tube containing 30 mg of $TiO_2$ beads (5 mm Titansphere; GL Sciences, Japan) and kept rotating at 4 °C for 20 – 30 min. After centrifugation at 5000 g for 10 min at 4 °C, the beads were transferred into a new 1.5 ml tube and washed with pre-cold PA for 3–5 times, then eluted with 500 μl of 10% ammonia solution. The eluate was freeze-dried at −50 °C and resuspended with 50 μl of 10% ammonia solution, and the enriched InsPs was resolved in a 33% polyacrylamide/Tris−borate−EDTA (TBE) gel and stained with toluidine blue. 10 μM synthetic $InsP_5$ (*myo*-Inositol-1,3,4,5,6-pentaphosphate ammonium salt, Cayman) and $InsP_6$ (Sichem) were used as the markers and standards. In order to verify the specific components in the eluate, the total eluate was firstly freeze-dried, then dissolved with 100 μl 80% acetonitrile.

InsPs were detected using Hydrophilic Interaction High Performance Liquid Chromatography-Tandem Mass Spectrometry on an Agilent 1290 infinity HPLC system coupled to an Agilent 6460 triple Quad LC-MS/MS using InfinityLab Poroshell 120 HILIC-Z (2.1 × 100) column (Agilent Technologies, USA). Nitrogen was used as the sheath gas and drying gas. The nebulizer pressure was set to 45 psi and the flow rate of drying gas was 5 L/min. The flow rate and temperature of the sheath gas were 11 L/min and 350°C, respectively. Chromatographic separation was carried out on a HPLC column (100 × 2.1 mm, 2.7 μm). The column temperature was 35°C. The mobile phases consisted of (A) ammonium acetate in distilled water (pH 10.0) and (B) Acetonitrile. The gradient program was as follows: 0−10 min, 90% → 55% of B; 10−12 min, 55% → 90% of B; 12−20 min, 90% of B. The flow rate was set at 0.3 ml/min, and the injection volume was 10 μl. Mass spectrometric detection was completed by use of an electrospray ionization (ESI) source in negative ion multiple-reaction monitoring (MRM) mode. InsPs were identified based on comparison to known InsPs species. The mass spectrometry parameters corresponding to different InsPs show as below: $InsP_3$ (MRM: 419 - > 321, 419 - > 337, Acquisition time is 6−7 min); $InsP_4$ (MRM: 499 - > 401, 499 - > 417, Acquisition time is 6−7 min); $InsP_5$ (MRM: 579 - > 480.9, 579 - > 382.8, Acquisition time is 6−7 min); $InsP_6$ (MRM: 659 - > 560.8, 659 - > 577, Acquisition time is 6−7 min); $InsP_7$ (MRM: 739 - > 575, 739 - > 657, Acquisition time is 5−6 min); $InsP_8$ (MRM: 819 - > 737, 819 - > 655, Acquisition time is 4−5 min). According to the regression equation calculated from the standard sample, substitute the response value of the sample into the equation to convert the corresponding concentration.

For $InsP_3$/$InsP_4$/$InsP_5$/$InsP_6$/$InsP_7$/$InsP_8$ detection in seeds and seedlings, 10 g of 12-day-old seedlings or 2.4 g of dry seeds were used for InsPs enrichment with 300 mg of $TiO_2$ beads for each sample. The enriched substances were analyzed by HPLC-MS/MS. The purchased $InsP_3$ (1,4,5-$InsP_3$, MedChemExpress), $InsP_4$ (1,3,4,5-$InsP_4$, MedChemExpress), $InsP_5$ and $InsP_6$, $InsP_7$ (5-$InsP_7$) and $InsP_8$ (1,5-$InsP_8$) from Lei lab[25] were used as standard samples for generating the calibration curves.

### Protein purification and phosphorylation analysis in vitro
pCold TF vector containing various CDSs referred in this study were transformed into DE3, and the recombinant proteins were expressed and purified. The purified fusion proteins were added into the kinase buffer (100 mM Hepes, 50 mM KCl, 50 mM $MgCl_2$, 0.05% Triton X − 100, PH 7.5) with 1 mM ATP-gamma-s (Abcam) in a total volume of 20 μl. Mixtures were incubated at 37 °C for 30 min, and then incubated with 2.5 mM p-Nitrobenzyl mesylate Alkylation (Abcam) for 2 h. For western blotting, anti-Phosphoserine/phospho-Ser antibody (Abcam, 1:10000) and anti-Rabbit-IgG-HRP (Sigma, 1:20000) were used to detect the phosphorylated band.

### Phosphorylation analysis in vivo
GRF4-flag protein was extracted from 7-day-old seedlings expressing 35 S::GRF4-flag in WT and T-4m backgrounds. Anti-flag and anti-pS242 antibodies were used to detect the phosphorylation degree of GRF4 in immunoblotting.

### In vitro IPK1 activity assay
Briefly, the purified proteins of IPK1-HIS (2.5 μg), IPCK1-HIS (3.75 μg), GRF4-HIS (2.5 μg), GRF4^242A-HIS (2.5 μg) and GRF4^242D-HIS (2.5 μg), which were enriched with elution buffer (50 mM Tris, 250 mM imidazole, pH 8.0), were used for IPK1 activity assay. The different combinations of proteins were added into 10 mM Tris buffer (pH 8.0) with 60 μM ATP and 39 μM $InsP_5$ (myo-Inositol-1,3,4,5,6-pentaphosphate ammonium salt) as the substrate in a total volume of 50 μl. The reactions were done at 37 °C for up to 6 h to produce $InsP_6$, and then were terminated by adding 50 μl of 100% acetonitrile. The concentrations of $InsP_5$/$InsP_6$ were detected with HPLC-MS/MS. PSKR1, which is a kinase

protein[75], was used as a negative control, and $ddH_2O$ was used as the blank control.

In order to generate the calibration curves for $InsP_5$ and $InsP_6$, different amounts of $InsP_5$ and $InsP_6$ (1, 0.5, 0.2, and 0.1 ppm) were respectively added to the reaction buffer (including 10 mM Tris, pH 8.0, 60 μM ATP, with the final volume of 50 μl), which was further subjected to a 37°C water bath for 6 h. The reaction was stopped by the addition of 50 μl of 100% acetonitrile. HPLC-MS/MS was used to detect the $InsP_5$ and $InsP_6$ content as described above. The calibration curve equation, associated coefficient values and recovery rate were calculated based on the $InsP_5$ and $InsP_6$ chromatographic absorbance values that were measured.

For ATP $K_m$ calculation, the kinetic analysis of IPK1 kinase activity with different concentrations of ATP (0.01, 0.015, 0.03, 0.06, 0.09, 0.12, 0.15, 0.25 and 0.5 mM) was performed in a reaction buffer (10 mM Tris, pH 8.0) containing 39 μM $InsP_5$ and 2.5 μg IPK1-HIS protein in a total volume of 50 μl. The reaction mixtures were incubated at 37 °C for 6 h. The concentrations of $InsP_5$ and $InsP_6$ were detected with HPLC-MS/MS, and rate constants was fitted and analyzed using GraphPad Prism 8.4.

### Plant plasma membrane protein extraction
0.03 g of 7-day-old 35 S::IPK1-flag/WT and 35 S::IPK1-flag/T-4m fresh seedlings were collected, and the membrane proteins were extracted according to the instructions of Plant Membrane Protein Extraction Kit (Invent Biotechnologies, Inc.v4). For western blotting, the anti-flag antibody (Abcam, 1:10000) was used to detect IPK1 protein, and the anti-actin (1:5000) and BRI1 antibody (1:500) were used as internal reference for total proteins and membrane proteins, respectively.

### qRT-PCR and RNA-Seq
Total RNAs were extracted from seedlings using an RNAprep Pure Plant Kit (Tiangen Biotech), and cDNA synthesis was conducted with ReverTra Ace qPCR RT Master Mix with gDNA Remover (Toyobo). Real-time qPCR analysis was performed using SYBR Green Realtime PCR Master Mix (Toyobo) on a Roche LightCycler 480 real-time qPCR system. Data were normalized with ACTIN2 and mRNA abundance was calculated using the delta Ct method[76]. The primers are listed in Supplementary Dataset 3. For RNA-Seq, 7-day-old seedlings of ipk1/C-5m treated with 1.25 mM Pi for 3 d, and 7-day-old WT seedlings were treated with 1.25 mM and 10 μM Pi respectively for 3 d (the differential transcript of WT induced by 10 μM Pi is used as a positive control named 10P-WT), samples were collected and total RNAs were extracted using an RNAprep Pure Plant Kit. RNA-Seq was performed by Novegene Co., Ltd (Beijing).

### Mass spectrometry of phosphorylation sites
To prepare samples for identifying putative phosphorylation site of GRF4 by IPCK1 with mass spectrometry, puried IPCK1-HIS and GRF4-HIS proteins were added to the kinase buffer with ATP and incubated at 37 °C for 30 min. The proteins were then separated in 10% SDS gel and dyed with the silver staining kit according to manufacturer's protocol (Pierce Silver Stain kit, Thermo Fisher). The phosphorylation bands were collected and sent to APPLIED PROTEIN TECHNOLOGY (Shanghai) for mass spectrometry to analyze the phosphorylation sites.

### Phylogenetic analysis and protein sequence alignment
The phylogenetic analysis of the RLCK V subfamily from Arabidopsis thaliana were done using MEGA 6.0. Multiple sequence alignment was performed with DNAMAN 8.0.

### Reporting summary
Further information on research design is available in the Nature Portfolio Reporting Summary linked to this article.

## Data availability

All data generated and analyzed in this study are available in the article and its supplementary information file. The RNA-seq data were deposited to NCBI SRA with accession number PRJNA941157. Metabolomics data has been deposited to the Metabolomics Workbench database with Project ID PR001986 [https://doi.org/10.21228/M81142]. Source data are provided with this paper.

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

## Acknowledgements

The authors thank Dr. Jing Ying Yan (Agricultural Experimental Station, Zhejiang University) for assistance of growing plant materials. This study was supported by the National Key Research and Development Program of China (2021YFF1000402), the Guangdong Laboratory for Lingnan Modern Agriculture (NT2021010), the Ministry of Education and Bureau of Foreign Experts of China (B14027), the ZJU Tang Scholar Foundation, and the Fundamental Research Funds for the Central Universities.

## Author contributions

Experimental design: L.L.X., Z.J.D., S.J.Z., C.Z.M., and M.B.; vector construction and transgenic line generation: L.L.X. and G.X.L.; biochemical experiments: L.L.X., M.Q.C., M.J.Z., and C.X.; ICP-MS, qRT-PCR, Pi analysis: L.L.X., X.D.W., and J.M.X.; data collection and analysis: L.L.X., Z.J.D., and W.N.D.; manuscript writing and revision: L.L.X., Z.J.D., and S.J.Z.

## Competing interests

The authors declare no competing interests.
