## [Peer Review File · Nature Communications]

A clade of receptor-like cytoplasmic kinases and 14-3-3 proteins coordinate the inositol hexaphosphate accumulationReviewer #1 (Remarks to the Author):

The manuscript claims that 'PICKs recruit IPK1, IPK2s and PI-PLCs via phosphorylation of GRF 14-3-3 proteins to establish an efficient IP6 biosynthetic pathway, revealing a previously unknown mechanism in regulation of IP biosynthesis in eukaryotes'.

I will restrict my review to aspects of inositol phosphate analysis, enzymology and biochemistry.

The data of Figure 5b,c are not easily understood. The experimental detail provided does not allow easy interpretation or an interpretation that is rooted in current understanding of IPK1 catalysis.

InsP7 (IP7) is not a known product of IPK of yeast, plants, animals, apico-complexans or any other organism, yet the manuscript claims to have measured only InsP7 as product. To convert Ins(1,3,4,5,6)P5 to InsP7 (isomer, undefined), the enzyme would need to convert Ins(1,3,4,5,6)P5 to Ins(1,2,3,4,5,6)P6 to InsP7 (a PP-InsP5, eg. 5PP-Ins(1,2,3,4,6)P5) OR phosphophorylate Ins(1,3,4,5,6)P5 on two separate phosphates to make a bis-PP-InsP3 (eg. bis-1,5-PP-Ins(3,4,6)P3). Which of these 'possibilities' is claimed?

IPK1 has only ever been shown to phosphorylate the (single) axial hydroxyl on myo-inositol's carbon position 2. Thus IPK1 is widely reported to convert Ins(1,3,4,5,6)P5 to Ins(1,2,3,4,5,6)P6, Ins(1,4,5,6)P4 to Ins(1,2,4,5,6)P5 or Ins(3,4,5,6)P4 to Ins(2,3,4,5,6)P5.

Figure 5b shows product 'InsP7' only. It does so only for assays including IPK1. It should show InsP5, InsP6 and InsP7 in the same chromatogram, so that the extent of conversion of substrate to product can be assessed. As presented the data, is very 'selective'.

The authors should determine rate constants for reactions performed under initial rate conditions (ie. in which it can be shown that less than ca. 10% of substrate is converted to product) ie. they must show that disappearance of substrate correlates with increase in product.

None of the experimental data by which product is identified as InsP7 is given, the manuscript makes no mention of which InsP7 isomer is produced nor does it mention the use of a standard by which the 'identification' can be validated.

More worryingly, the negative control reports levels of substrate (InsP5) 1000-fold less than the amount of 'product' formed.

Calibration curves by which detector response of undefined LC-MS,MS parent/daughter transitions can be calibrated are not described. Indeed, no details of the transitions: they will be different for IP7, IP6 and IP5, and possibly different for different isomers, are provided.

Beyond these analytical details, there are issues in the set up of the enzyme assays. These phosphotransfer assays with ATP and inositol phosphate substrate at sub millimolar levels is performed in 100 mM phosphate. This is an unprecedented choice of buffer for an inositol phosphate hydroxy-kinase assay. The products of these assays are simply freeze-dried and reconstituted in 80% acetonitrile. All of this is performed against a background of 100 mM phosphate.

Have the authors undertaken any experiments to assess the recovery (solubilization) of substrates/products?

For an assay performed for 6h, the assay would have been better performed with an ATP-regenerating assay. It is very likely that the multiple purified proteins added to the assay (no indication of purity is provided) might carry phosphatase activities that could degrade added ATP. Assays that allow monitoring of substrates (inositol phosphate and ATP) and products are particularly useful to assess for interfering activities.

In summary, this reviewer has no confidence in any of the enzymology on which the defining claim of the manuscript, 'Our findings therefore establish an efficient lipid-dependent IP6 biosynthesis

pathway governed by PICKs, shedding new light on the mechanisms of IPs biosynthesis in eukaryote' are made.

The manuscript lacks test that PLC, IPK2 and IPK1 constitute a pathway, a metabolic concept. The review cited (Gillaspy et al 2020) could be a starting point for construction of incisive experiments that attempt to discriminate between alternative lipid-dependent and lipid-independent pathways of InsP6 synthesis. Indeed, if IPK1 is epistatic to IPK2 and PLC2, mutation of PLC2 alone should alter InsP6 levels.

To analyse a pathway, the authors should use methods of analysis that measure the level of pathway intermediates, not just an end product.

The manuscript provides no explanation or hypothesis how the activity of IPK1 is modulated, neither at transcriptional or post-transcriptional level.

The manuscript appears to reuse data of Figure 2a (Pi content of seeds and separately Pi content of plants, for WT, T-4m, C-5m-2) in Figure 3e.

The manuscript provides no methodological details for the data of Figure 7e

The analysis of inositol phosphates by PAGE after pre-concentration on TiO₂ in Figures 2c,d; 3d; 4g,h; 5e appear to be single determinations that could be prone to inconsistencies in recovery of inositol phosphates on TiO₂ and the loading of gels. The method is semi-quantitative at best.

'Values are given as mean \pm SD, n = 3', is a suffix to many of the figure legends including eg., Fig S3. Not a single panel of Fig. S3 provides values of mean and sd.

Separately, one may assume that the numerical values on panel D (and on other gels analyzing inositol phosphates) reflect densitometry of the InsP6 band in different genotypes relative to WT. For single lanes of the different genotypes – this could merely reflect differences in recovery/loading of sample. There is no internal calibration to an 'invariant' reference: indeed, only a small part of the gel is shown.

Reviewer #2 (Remarks to the Author):

The manuscript "A clade of receptor like cytoplasmic kinases and 14 3 3 proteins coordinate the 1 lipid originated inositol hexaphosphate biosynthesis" shows a well-designed approximation to decipher the regulatory mechanisms that underlie InsPs biosynthesis. This manuscript shows the interaction of one clade of RLCKs, named Phosphoinositide related 23 Cytoplasmic Kinases 1 with IPK1 and 2 regulating InsP6 biosynthesis and compromising inositol pyrophosphates synthesis downstream.

Overall, it is a well-written and organized work, that demonstrates the importance of forming this complex together with GRF4 in order to correctly sense Pi levels and regulate InsP6 biosynthesis. Saying this, I have also some improvements to suggest to clarify some aspects of the proposed work.

Minor aspects:

Overall the manuscript please correct the nomenclature for InsPs. For instance, InsP6 (6 in lower case) instead of IP6, and please be consistent.

Line 74 is missing references on *Chlamydomonas reinhardtii* vip1-1 mutant

Line 103 "...redundantly functioned in inositol phosphate biosynthesis" Please name them as inositol polyphosphates

Line 153 "IP8 was might be slightly affected " Please reword this sentence

Figure 1 regarding panel C AtSPX3-mCherry RFP fluorescence, please indicate in the text or legend that it is a colocalization control as SPX3 is also related to phosphate regulation in plants.

Major aspects:

1. In order to further explore the redundancy of the PICK genes and their interaction, I wonder whether the authors had also tried C-5m with PICK6 instead of PICK5. If not, they should consider this line as an additional control.
2. I have some concerns regarding figure 2 panels C and D. InsP6 levels are highly downregulated in the mutants which are clear, especially for C5-m1 and C5-m2 (in C panel) but compared with the lower gel (seedlings) at panel D C-4m-1 and C-4m-2 should have approximately the same drop but (to my eyes) there is not such a drop in the bands of InsP6 please authors comment on this. I would recommend including the HPLC-MS/MS quantitation of seedlings as well.
3. I highly recommend authors mention the results of complemented lines to support their hypothesis on the redundancy of PICK genes.
4. I highly recommend clarifying the content of the supplemental tables regarding IP data for example including the protein ID and if possible the sequence of the peptide detected. Please name the tables as they are named in the manuscript.
5. I would recommend including a negative control maybe with PICK5 or 6 in the BiFC analysis (Figure 3)

Taking together all these aspects, I would recommend the publication of this manuscript after these major changes have been done.

Reviewer #3 (Remarks to the Author):

Overall this is a very strong paper that in my opinion will move the field of inositol polyphosphate homeostasis and in particular InsP6 synthesis forward in a very significant manner. The findings will also obtain attention from non-plant researchers and people interested in inositol pyrophosphate signaling.

The authors walked a long way, encountered redundancy at several levels that they had to overcome by generating higher order mutants (PICKs and 14-3-3 proteins/GRFs) that were then complemented. The main message that PICKs function as organizers to modulate the function of the enzymes involved in inositol phosphate synthesis is convincing.

I see mostly issues with respect to the introduction and how the authors put their findings in context to what has been published in the field. I think the name "Phosphoinositide related Cytoplasmic Kinases (PICKS) is unfortunate and I hope the authors will reconsider this name (explanation below). I do also have some minor issues with respect to the experimental part. I will list the issues as they appeared to me when reading the MS, i.e. mostly in order:

1. Line 37: First sentence of the Intro (on the roles of phytic acid in stress responses, development, phosphate homeostasis, DNA repair and membrane trafficking). The authors provide a single experimental paper on oxidative stress responses in a particular maize low phytic acid mutant by Doria et al. to reference all of the roles of phytic acid. I find this inappropriate and would like to suggest to either cite the original papers or cite one or several dedicated reviews on phytic acid.

2. Line 57: "phosphorylated to IP6; and a lipid independent pathway, where myo inositol (3) P1 (Ins(3)P1) from glycolysis undergoes myo inositol 1 phosphate synthase (MIPS) and a series of phosphorylations until IP6 is formed". This sentence has a strange wording. To my understanding, MIPS proteins catalyze the conversion of D-glucose-6-phosphate to Ins(3)P

3. Line 62: "IPK1 can also specifically phosphorylate the D-2 hydroxyl on IP6 to produce IP7 (an inositol pyrophosphate)". This has never been shown nor has it been claimed by anyone to my knowledge. There is also no immediate reference provided by the authors. Maybe the authors refer

to the Kuo et al. (2014) Plant J paper that is cited two sentences further down (citation 14)? However, Kuo et al. make no claim that IPK1 is directly involved in the phosphorylation of InsP6 to InsP7 to my knowledge.

4.Line 65. The following sentence is strangely worded and I believe incorrect "In addition, IPK1 cooperates with IPK2 (including IPK2 α and IPK2 β that harbor 6/3 kinase activity and sequentially phosphorylate Ins(1,4,5)P3 to generate IP5 via an Ins(1,3,4,6)P4 intermediate) and ITPKs to convert the PI PLC generated or the glycolysis derived IP3 into IP6 by transferring phosphorylation of the phosphate group on ATP".

To my knowledge, with the exception of budding yeast, it is not known how IP6 is produced in eukaryotes. Only the role of IPK1 in catalyzing the conversion of IP5[2-OH] to IP6 seems to be established in various organism. In contrast, the role of IPK2 in IP6 synthesis and the first steps until IP5 are poorly understood. The statements made by the authors appear to come from the important Stevenson -Paulik et al. (2005) PNAS paper which is later cited (reference 63, but should be cited here). The statement "ITPKs to convert the PI PLC generated or the glycolysis derived IP3 into IP6 by transferring phosphorylation of the phosphate group on ATP" is incorrect. As shown by Desfougères et al. (2019) PNAS <https://www.pnas.org/doi/epdf/10.1073/pnas.1911431116>: Plant (and human) ITPK1 can rescue the InsP_x profile of yeast plc KO because it can use Ins(1)P1 produced by ISC1 that acts on Inositol Phosphoryl Ceramide (IPC). ITPK1 cannot restore IP6 in plc1 isc1 dKO strain and ITPK1 does not recognize Ins(1,4,5)P3, a PLC product in vitro.

5.I wonder if the authors mixed up IPK1 and ITPK1 at some point with respect to inositol pyrophosphate synthesis. ITPK1 was the first plant enzyme shown to catalyze IP6 to 5-IP7 conversion in vitro and in yeast (Laha et al. (2019) ACS Chem Biol (<https://pubs.acs.org/doi/full/10.1021/acscchembio.9b00423>) and W Whitfield et al (2020) Biochem J (<https://portlandpress.com/biochemj/article/477/14/2621/225707/An-ATP-responsive-metabolic-cassette-comprised-of>) First evidence that ITPK1 is indeed involved in 5-InsP7 synthesis in planta comes from Riemer et al. (2021) Mol Plant (<https://pubmed.ncbi.nlm.nih.gov/34274522/>)

6.Line 70 ". Recent studies have found that more strongly phosphorylated species exist in the IP6 derived inositol pyrophosphates IP7 and IP8 through di phosphoinositol pentakisphosphate kinases VIH1 / 2 (named after VIP1 homologs), lack of which leads to constitutive Pi starvation response and impaired plant growth and 73 development 15,16 .

I think something got mixed up with the references here.

The first evidence of inositol pyrophosphates in plants comes to my knowledge from Brearley and Hanke (1996) Biochem. J. (<https://pubmed.ncbi.nlm.nih.gov/8761483/>) and Lemtiri-Chlieh et al. (2000) PNAS (<https://www.pnas.org/doi/abs/10.1073/pnas.140217497?doi=10.1073/pnas.140217497>). First description of plant enzymes involved in their synthesis from Desai et al (2014) Plant J (in vitro and yeast) and Laha et al (2015) Plant Cell (in vitro, yeast and in planta). First description of enzyme involved in plant 5-InsP7 synthesis by Riemer et al (2021). The papers reporting that vih1 vih2 double mutants result in PSR are Dong et al (2019) Mol Plant (now citation 22) and Zhu et al (2019) elife (now citation 19).

7.Line 76 "... These findings also reveal that Pi homeostasis in plants is regulated by kinases involved in IPs synthesis, most likely due to their indirect contribution to the synthesis of IP8" Here two more papers should be cited: the Stevenson-Paulik (2005) PNAS paper 2005 (see above) which was the first paper reporting a PSR response of a mutant defective in InsP6 synthesis and Riemer et al. (2021) Mol Plant paper that showed ITPK1 controls phosphate starvation responses by catalyzing the first step in inositol pyrophosphate synthesis, i.e. 5-IP7, the precursor of IP8 (see above).

8.Line 104: "RLCK V subfamily has not been documented in terms of functions. Here we showed that these two kinases together with other related RLCK V subfamily members redundantly functioned in inositol phosphate biosynthesis. We therefore named them Phosphoinositide related Cytoplasmic Kinases 1 6 (PICK1 PICK6)."

I find this name very unfortunate and I strongly suggest to reconsider it!

Phosphoinositides are (membrane) lipids (!) generated by phosphorylation of the inositol headgroup of phosphatidylinositol. They have a typical phospholipid structure (headgroup, glycerol backbone, acyl chain(s) and should not be confused with inositol phosphates which lack the glycerol backbone and acyl chains.

9. Line 110 "In the preliminary study, we employed a quantitative phosphoproteomics using wild type (WT) plants under control (1.25 mM Pi) and Pi deficiency (10 μ M Pi) treatments for 1 h, to identify potential regulators in Pi homeostasis". I wonder how this experiment was done. Where plants transferred from a hydroponic + P culture to a hydroponic minus P culture? Or where they transferred from +P to minus P solidified (MS based) media? I wonder if the authors agree that Pi deficiency cannot be induced in one hour. Maybe the authors should reword this: 'Challenge of P sufficient plants with low P (10 μ M) growth medium', something along these lines. If seedlings were transferred between solidified media (e.g. MS-plates), how can they say that changes in the phospho-proteome is related to the low P content of the media and not just a consequence of injury (damage of root hairs etc.) due to transfer? Was there a control in which plants were transferred from plus to plus P medium to account for plant responses to damage or handling? Along the same line of thought: The sentence in line 366 "PICK1 and PICK2 were initially identified because of their reduced phosphorylation levels under Pi deficiency (Supplementary Fig. 1a)" is probably misleading. If the treatment was really only done for one hour, I think one cannot call this treatment P deficiency.

10. Line 115 and figure 1 (and respective figure legend). The authors should not use red and green as many people are colorblind. They could combine red with blue or cyan or combine green with magenta or orange (something that should be easily done with the images they have).

11. For clarity, authors should indicate in the figure legend that the green box represents a transmembrane domain. They should also explain why they used AtSPX3 (as a cytosolic marker?).

12. Lines 135: the authors should mention how they measured P – I cannot find this in the method section.

13. Line 142: e). "Both the increase of Pi accumulation and 142 the decrease of IP6 content in T 4m could be restored by PICK1 or PICK2 (Fig. 2a, 2d)." The authors should mention under control of which promoter PICK1 and PICK2 were expressed for this experiment. I could not find the information in the method part (it should be mentioned there but also here).

14. Line 159 "... finding IPK1, the key synthase of IP6 / IP7, to be a good candidate". See also comment above: No evidence that IPK1 is directly mediating IP7 synthesis by any previous work to my knowledge.

15. Line 208: "...the kinase domain could phosphorylate GRF4 in vitro , while a kinase dead mutation 208 (PICK 1 K182E) abolished this phosphorylation (Supplementary Fig. 3b I find this not so convincing as there is much less protein of the catalytic dead mutant version as compared to wt

16. Line 242/Figure 5. In Figure 5b on the very top "...GRF4S242" does not indicate whether it refers to S242A or the phosphomimic S242D protein variant.

17. Line 278: PLC activity as well. "...Indeed, we found that PI PLC2 which belongs to PI PLCs subfamily and functions in IP3 synthesis..." To my knowledge its under debate whether it is IP2 or IP3 synthesis in plants. Both might play a role as suggested by work of Teun Munnik (https://link.springer.com/chapter/10.1007/978-3-642-42011-5_2; <https://academic.oup.com/pcp/article/59/3/469/4772709?login=true>

18. Line 285: "...PLC2 via phosphorylation of GRF4 S242 residue, to form a 285 PICK1 GRF4 PI PLC2 complex as did for IPK1 and IPK2 β (Fig. 7d)." If I am not mistaken, several negative controls are missing in this figure.

19. Line 523: "IPK1 activity assay was performed as previous described with some modifications" I

wonder whether Dong et al really carried out IPK1 activity assays, or do the authors refer to activity assay of the InsP7 kinase VIH?

20. Line 321. I would like to urge the authors to not repeat unfounded claims about "IP3 signaling" in plants. I suggest reading a small section written in the paper Zhang et al (<https://academic.oup.com/pcp/article/59/3/469/4772709?login=true>) by the lab of Teun Munnik which boils this down better than any other reference I am aware of and provides many relevant citations. I am citing from this paper: "Much less is clear about the PLC signaling system in plants (Ischebeck et al. 2010, Munnik 2014, Heilmann 2016, Heilmann and Ischebeck 2016, Gerth et al. 2017). Initially, it was thought to be equivalent to the animal paradigm since most of the components driving the pathway were thought to be present (Munnik et al. 1998a, Stevenson et al. 2000, Meijer and Munnik 2003), especially when microinjected IP3 was shown to release Ca²⁺ from an intracellular store and to induce stomatal closure (Blatt et al. 1990, Gilroy et al. 1990, Allen and Sanders 1994, Hunt and Gray 2001). However, 20 years later, Brearley's lab provided evidence that it was not IP3, but its subsequent conversion into IP6, that caused these effects (Lemtiri-Chlieh et al. 2000, Lemtiri-Chlieh et al. 2003). Similarly, not DAG but its phosphorylated product, phosphatidic acid (PA), has been emerging as the plant lipid second messenger (Munnik 2001, Laxalt and Munnik 2002, Testerink and Munnik 2005, Arisz et al. 2009, Testerink and Munnik 2011, McLoughlin and Testerink 2013, Pokotylo et al. 2014, Munnik 2014, Hou et al. 2016, Vermeer et al. 2017). Moreover, genome sequencing has meanwhile confirmed that flowering plants lack homologs of both the IP3 receptor and PKC (Wheeler and Brownlee 2008, Munnik and Testerink 2009, Munnik and Vermeer 2010, Munnik and Nielsen 2011, Munnik 2014, Heilmann 2016, Gerth et al. 2017)."

21. Authors should also check for typos and grammar (overall it is well written though).

Reviewer #4 (Remarks to the Author):

In this manuscript, Xu and colleagues report the identification of a group of receptor-like cytoplasmic kinases (RLCKs) that is proposed to recruit multi-protein complex that can modulate the synthesis of different inositol polyphosphates (IPs), including inositol hexakisphosphate (phytic acid or IP6).

The study started by the finding that phosphorus (P) deficiency inhibits the phosphorylation of two members of the subfamily V of RLCKs. Closely related members of this subfamily show comparable expression at the tissue level. Disruption of four or five of these genes (named as PICK1 to 5) resulted in increased phosphate concentrations in seeds and in whole seedlings, while a sextuple mutant is likely embryo lethal. This phenotype was accompanied by decreased levels of IP6 and inositol pentaphosphate (IP5), and upregulation of P deficiency-induced genes. Using BiFC, in vitro pull-down and split-LUC assays, the authors found that PICK1-4 can physically interact with IPK1, the main kinase responsible for IP6 synthesis, and with IPK2 α and β , which form IP4 and IP5. While PICKs cannot phosphorylate IPK1, IP-MS identified GRFs (14-3-3 proteins) as a putative interacting proteins with PICK1, which was confirmed with BiFC and split-LUC. The authors then show that GRF4 can be phosphorylated at Ser242 by PICK1, and interact with IPK1. According to semi-quantitative PAGE, *grf3grf4* double mutant has decreased IP6 levels, which can be rescued with the native GRF4 protein but not with a Ser242Ala variant. Next, the authors attempted to demonstrate in vitro IP6 biosynthesis activity of a PICK1- GRF4-IPK1 recombinant proteins. Although IP5 was supplied as substrate, surprisingly only IP7 but not IP6 was detected, an activity that was increased when IPK1, PICK1 and GRF4 (native or as a phosphomimicking variant) were present together. Finally, the authors show that PICKs and GRF4 can physically interact with PI-PLC2 and 7, which are involved in lipid-dependent IP3 synthesis. In conclusion, the authors suggest that a large protein complex formed by PICKs, GRF4, IPK1, IPK2 and IP-PLCs modulates lipid-dependent IP6 biosynthesis.

The reported findings are novel and of potential relevance as they reveal an unknown mechanism regulating inositol polyphosphate biosynthesis in plant cells. The manuscript presents an impressive amount of data and identifies several interactions among known and previously uncharacterized proteins. While the protein-protein interactions assays are validated with independent approaches, they heavily rely on transient expression with constitutive promoters. Furthermore, the biological context in which the proposed interactions occur, and how and when

they control the synthesis of specific IPs are less clear. Points that need further attention by the authors are detailed below.

Major comments:

1) The study contains a wealth of data regarding the interaction of several proteins and present some elegant evidence of how these interactions can regulate the function of IP kinases. However, one weak point is that these interactions are presented without a clear biological context. As a consequence, the model derived from the study (Fig. 8) shows the identified protein complex loosely connected to known biosynthesis or signaling pathways. For instance, while the manuscript starts with the discovery that the phosphorylation of PICK1 and 2 is inhibited in response to P deficiency, the P status-dependent context is not further worked out during the study. Is PICK1 phosphorylation relevant at all and required for its interaction with GRF4? Does it affect the ability of PICKs to recruit other proteins? If yes, then is the formation of the proposed complex altered according to the cell's P status?

2) Another weak point is that the shown protein-protein interactions derived almost exclusively from transient expression (with constitutive promoters) or in vitro assays. This is insufficient. At least the key interactions should be also demonstrated in planta (e.g. by co-IP) using the native promoters to drive the expression of the candidate genes. A similar weakness concerns the existence of the proposed protein complex. To demonstrate that a PICK-recruited complex is indeed formed at the plasma membrane, it is necessary to show that e.g., IPK1 or IPK2s are associated to the plasma-membrane and that this localization is disturbed in pick quadruple or quintuple mutants.

3) I am also not satisfied with the analysis of IPs. For the most part, the manuscript mainly report IP levels as relative changes based on quantification of band intensities from SDS-PAGE gels. A major shortcoming of this approach and even of the HPLC-MS/MS method used in the study, is that only IP5 and IP6 but no other IP could be detected. It is not even clear whether the IP5 species that serves as substrate for IP6 synthesis (i.e., 2-OH IP5) was the IP5 species disturbed by the loss of PICKs. Furthermore, none of the IP concentrations presented, irrespective of the method used, are normalized to seed or plant mass nor compared with a statistical method. A more thorough characterization of the IP profile, at least of the pick mutants, should be additionally carried out by SAX-HPLC or CE-MS as e.g., Kuo et al., 2018 doi.org/10.1111/tpj.13974 and Riemer et al., 2021 doi.org/10.1016/j.molp.2021.07.011. This would significantly strengthen the manuscript by more directly determining the steps in IP6 synthesis that are modulated by PICKs, and by providing much needed direct support to clarify whether PICKs do indeed control the lipid-dependent route of IP6 synthesis, as proposed in the model and in the manuscript title.

4) For the most part the manuscript is poorly written and provide insufficient methodological information to understand the results or to follow the rationale behind the selection of candidate proteins. i) To improve the grammar and logic of the text, the authors should seek assistance of professional language editing or from a colleague with good command of English. ii) To facilitate understanding and interpretation of the results, more details are necessary in Material and Methods, including the name of all plasmids used in the different experiments/assays, gRNA sequences and types of mutations obtained, detailed description of protocols instead of referring to published studies, and instrumentation (e.g., brand and model of ICP-MS). iii) To make the figures self-explanatory, more experimental information must be included in the legends, as is done in articles published in Nat Commun.

5) When attempting to investigate IPK1 activity when supplying IP5 as a substrate, the authors only detected IP7 but not the expected direct product IP6. They simply concluded that IP6 was immediately converted to IP7 (lines 242-243). This is not acceptable. Does it mean that IPK1 has also IPK6 kinase activity, and that this activity is so high that all generated IP6 is consumed to generate IP7? That would go against published work. Without a clear characterization of the reason behind this unexpected result, the results shown in Fig. 5b-d are meaningless and definitely cannot be used to conclude about IPK1 activity. Furthermore, what IP5 species was provided as substrate in the in vitro assay? IPK1 can phosphorylate 2OH-IP5 but not other IP5 forms.

6) IP6 is a major P-storage compound in seeds and the lipid-independent pathway has been

proposed to dominate IP6 biosynthesis in seeds. However, the results from the present manuscript suggest that a lipid-dependent route also makes a large contribution. How can membrane-derived phosphatidylinositol supply enough substrate to support the synthesis of large amounts of IP6 stored in seeds? Is there a possibility that PICKs also (or even mainly) affect the lipid-independent pathway?

7) Following the last comment and considering the importance of these questions to contextualize the identified mechanism, it is necessary to determine whether the lipid-independent pathway is also modulated by PICKs. The absence of physical interaction with proteins involved with the lipid-independent pathway does not automatically preclude that a direct or even indirect control can occur. At least the expression of MIPs and the analyses of intermediates formed by the lipid-independent route (see also comment 2) should be compared in WT and pick multiple mutants.

8) Protein phosphorylation usually means control of activity. The authors show that PICK1 can phosphorylate GRF4 and that the phosphorylation level of GRF4's S242 is decreased in the pick quadruple mutant. However, is GRF4 phosphorylation at all modulated? If yes, and if it is important to modulate IP6 synthesis, suggested by the authors (e.g., model in Fig. 8), then how does it respond to the P nutritional status? Or is it developmentally controlled being e.g., activated during seed development when P reserves in the form of IP6 are generated?

9) The altered expression of some Pi deficiency-induced genes cannot be used to imply that IP8 accumulation is reduced in the pick quadruple and quintuple mutants, as written in lines 152-153. To conclude about the level of any inositol pyrophosphate, such as IP8, their concentrations must be determined. Without these data, even the connection of the proposed protein complex with IP8-mediated Pi signaling, as shown in the model (Fig. 8), is too speculative.

10) The subcellular localization data shown in Fig. 1c are insufficient to prove that PICKs are present in the plasma-membrane (PM). A PM-specific dye or fusion protein should be used for the co-localization. Some green signal is also overlapping with SPX3:mCherry, even in what seems to be the nucleus (e.g., PICK5). Please check if the selected GFP filter does completely block mCherry-derived signal from passing through.

11) Although the authors provide some evidence that the many interactions that they identified may indeed occur in planta, apart from PICK1-6, no tissue-specific localization is shown for GRF4, IPK1, IPK2s, and PI-PLC2,7 nor reference to other studies is provided. Thus, it remains unclear whether the different proteins that can be recruited by PICKs are even present in the same cells at the same time. This is key to demonstrate not only whether the interactions occur, but also where and under what conditions.

12) According to the proposed model, many IP kinases would assemble as a large complex at the cytosol-facing side of the plasma-membrane to generate different IPs. If assembled in such a way, then how do the authors envisage that intermediates, such as IP4 and IP5 can still accumulate instead of the reactions always ending up in IP6? This aspect should at least be discussed.

13) The criteria used for the selection of candidates in split-ubiquitin membrane Y2H screens or immunoprecipitation MS are not clearly described in the text. For instance, in lines 198-199, the authors write that "...GRF4 was found to be most abundant in PICK1 immunoprecipitants.". In the Excel sheet, GRF4 appears with a PepCount value of 20, below many other proteins. Were any of the other top proteins tested as well? Furthermore, it is not even clear why PICK1 and PICK2 were selected in the first place, as they were not the only proteins, whose phosphorylation was inhibited in response to P deficiency. Please provide more clear explanations for the selection of these candidates.

14) Considering that PICKs are required to recruit IPK1 via GRF4 phosphorylation, then how to interpret that overexpression of IPK1 or GRF4 can restore IP6 concentrations and Pi accumulation in the pick quadruple mutant?

Minor points:

15) The files containing the reported "Supplementary Tables" should have been named as such. It was difficult to find out to what specific Table each Excel sheet was referring to.

- 16) Apart from showing that a PICK-GRF4-IPK1-IPK2-PLC multi protein complex can be formed, its connection to known pathways shown in the model of Fig. 8 are less clear. For example, the authors represent IP6 mainly in the context of Pi signaling. However, their results indicate that IP6 levels are decreased also in seeds, where IP6 may have mainly a role as a P-storage form.
- 17) Lines 41-44: it is not the presence of phytate per se but rather the overall high amount of P in seeds that is a main source of P pollution. Even ruminants release a lot of P in their waste. Please rephrase accordingly.
- 18) Line 52: the term "excitation of Ca²⁺ signals" is inappropriate. Replace with "triggering".
- 19) Line 58: something is missing in the phrase "...undergoes myo-inositol-1-phosphate-synthase". Please rephrase accordingly.
- 20) Lines 98 and 110: writing "preliminary study" is incorrect if referring to data generated in the present study.
- 21) Fig. 1d: embryos are labeled as "endosperm". Please correct. Furthermore, all embryos shown in the figure are completely damaged. GUS staining should be shown with intact embryos.
- 22) Line 129 and throughout the manuscript: instead of "pentamutant" please write "quintuple mutant".
- 23) Line 135 and throughout the manuscript: in plant nutrition, "content" is used to express amount per tissue, organ or whole plant, while amount per unit mass (as $\mu\text{mol g}^{-1}$ DW) refers to "concentration". Please correct in the text and figures.
- 24) Line 162: previously IPK1 was shown to localize in the nucleus and cytosol (Kuo et al., 2018 doi.org/10.1111/tpj.13974). What could be the reason that IPK1 was not localized in the nucleus in the present study?
- 25) In Fig. 1f, IP concentrations in seeds are expressed as ng μL^{-1} . This suggest that the values are expressed per unit extract. How are these values normalized to the mass of seeds used for extraction?
- 26) Please provide a full picture of all SDS/PAGE gels shown in the figures.
- 27) Fig. 1d and elsewhere: two gels are show but without any explanation. Do they show two independent preparations? Furthermore, I cannot find the " \pm SD" from the grey values. Are these values expressed relative to WT? This seems to be the case but the procedure is not clearly described neither in the Material and Methods nor in the legends.
- 28) Fig. 2: clarify in the figure that "COM" refers to complementation of the T-4m quadruple mutant.
- 29) Lines 173-174 and Fig. 3e: Please clarify whether total P concentration was significantly altered or only slightly changed as it was the case for the pick quadruple and quintuple mutants.
- 30) Lines 188-191: The authors claim that PICK1 cannot phosphorylate IPK1 in vitro. However, can IPK1 be phosphorylated at all? If yes, please include a reference.
- 31) Fig.3f: What does "10P-WT" mean? Was the transcriptome of C-5m and ipk1 compared to wild-type plants grown on low-P conditions? If yes, clarify in the text and add details about this specific growth condition in the legend and Material and Methods.

The following outlines our point-by-point responses (in blue) to reviewers' comments:

Reviewer #1 (Remarks to the Author):

The manuscript claims that 'PICKs recruit IPK1, IPK2s and PI-PLCs via phosphorylation of GRF 14-3-3 proteins to establish an efficient IP6 biosynthetic pathway, revealing a previously unknown mechanism in regulation of IP biosynthesis in eukaryotes'. I will restrict my review to aspects of inositol phosphate analysis, enzymology and biochemistry.

Response: We thank the reviewer for the insightful comments on the aspects of InsP analysis, enzymology and biochemistry. We have carefully considered all these comments, conducted all the suggested experiments, clarified the misleading information regarding the measurement of InsP₇ and revised our manuscript accordingly.

Concern1: The data of Fig 5b,c are not easily understood. The experimental detail provided does not allow easy interpretation or an interpretation that is rooted in current understanding of IPK1 catalysis. InsP₇ (IP₇) is not a known product of IPK of yeast, plants, animals, apico-complexans or any other organism, yet the manuscript claims to have measured only InsP₇ as product. To convert Ins(1,3,4,5,6)P₅ to InsP₇ (isomer, undefined), the enzyme would need to convert Ins(1,3,4,5,6)P₅ to Ins(1,2,3,4,5,6)P₆ to InsP₇ (a PP-InsP₅, eg. 5PP-Ins(1,2,3,4,6)P₅) OR phosphophorylate Ins(1,3,4,5,6)P₅ on two separate phosphates to make a bis-PP-InsP₃ (eg. bis-1,5-PP-Ins(3,4,6)P₃). Which of these 'possibilities' is claimed? IPK1 has only ever been shown to phosphorylate the (single) axial hydroxyl on myo-inositol's carbon position 2. Thus IPK1 is widely reported to convert Ins(1,3,4,5,6)P₅ to Ins(1,2,3,4,5,6)P₆, Ins(1,4,5,6)P₄ to Ins(1,2,4,5,6)P₅ or Ins(3,4,5,6)P₄ to Ins(2,3,4,5,6)P₅.

Response: We greatly thank the reviewer for pointing out this. We agree that IPK1 is unlikely to convert InsP₆ to InsP₇. The key point is that with the previous protocol, we really could not detect InsP₆. By trying many different combinations of proteins and buffer compositions in the measurement, we indeed found that the neutral pH and high concentration of Na⁺ and phosphate in the buffer interfered the detection of InsP₆ although we still do not know why and how InsP₇ is synthesized in the *in vitro* IPK1 activity assay. We therefore have optimized the components of the reaction buffer, where Tris (pH 8.0) is used and sodium and phosphate are excluded, and have successfully detected the InsP₆ products, but not InsP₇, synthesized by IPK1 when InsP₅ (2OH-InsP₅) is used as the substrate. We have presented these new data of *in vitro* IPK1 activity assay (Fig 5b-c) and corrected the related descriptions in the revised manuscript.

Fig. 5 IPCK1 facilitates IPK1 activity by phosphorylating GRF4

b Chromatogram absorption diagram. The abscissa represents the sample retention time, and the ordinate represents the peak area of the sample chromatogram. Different color absorption peaks and areas correspond to various protein combinations. ddH₂O and 2.5 μg of PSKR1 were used as the blank control and negative control, respectively. **c** The concentration of InsP₅ / InsP₆ converted from absorption peak area in **b**. Values are mean \pm SD, n = 3.

Concern2: Fig 5b shows product ‘InsP7’ only. It does so only for assays including IPK1. It should show InsP₅, InsP₆ and InsP₇ in the same chromatogram, so that the extent of conversion of substrate to product can be assessed. As presented the data, is very ‘selective’.

Response: We thank the reviewer for this good suggestion. As stated in the last point, the optimization of the reaction buffer enables the detection of InsP₆ and InsP₅, but not InsP₇ in the *in vitro* assay. As suggested, both InsP₅ and InsP₆ contents are now presented in the new data. We show that the amount of synthesized InsP₆ are in accord with that of reduced InsP₅. The results are shown in Fig 5b-c.

Concern3: The authors should determine rate constants for reactions performed under initial rate conditions (ie. in which it can be shown that less than ca. 10% of substrate is converted to product) ie. they must show that disappearance of substrate correlates with increase in product.

Response: Those comments are valuable and very helpful. We have determined the rate constants for the reactions, and shown that the disappearance of InsP₅ correlates with the increase of InsP₆. The data are shown in Supplementary Fig 5b-c, and the reaction rate constant K_m is 0.04328 ± 0.00923 mM.

Supplementary Fig. 5 *in vitro* IPK1 activity assay

b The dynamic concentrations of InsP₅ and InsP₆ in the IPK1 activity assay. Each concentration was calculated from the integrated HPLC peaks. Values are mean \pm SD, n = 3. **c** The ATP-dependent kinetic analysis of IPK1 activity. Data were fitted to the Michaelis-Menten equation. Each point is a single measurement from a single experiment. the experiment was repeated three times. K_m and V_{max} (mean \pm SD, n = 3) are given in the text. Assays were performed in 50 μ l of reaction buffers (10 mM Tris, pH 8.0, 39 μ M InsP₅ and 2.5 μ g IPK1-HIS protein) with corresponding ATP concentration (0.01, 0.015, 0.03, 0.06, 0.09, 0.12, 0.15, 0.25, 0.5 mM).

Concern4: None of the experimental data by which product is identified as InsP₇ is given, the manuscript makes no mention of which InsP₇ isomer is produced nor does it mention the use of a standard by which the ‘identification’ can be validated.

Response: The detection of InsP₇ is indeed an unexpected result due to unknown reason as we mentioned in the first point. InsP₆ is really unstable when the solution system is unproper. For example, a relative low pH may cause InsP₆ binding to certain amino acid residues of proteins, and several metal ions (e.g. Ca, Mg, Zn, Na, Fe, etc) have high affinity to InsP₆, leading to InsP₆ precipitation. The optimization of the reaction buffer now enables the detection of InsP₆ instead of InsP₇. We have presented the new results of the IPK1 activity assay (Fig. 5b-c) in the revised manuscript.

Concern5: More worryingly, the negative control reports levels of substrate (InsP₅) 1000-fold less than the amount of ‘product’ formed.

Response: We guess the reviewer had mixed the ordinate units of Fig 5b and 5d in the original manuscript. Fig 5b and 5d actually have different ordinate units, and Fig 5d should be compared with Fig 5c. The levels of InsP₅ ($\sim 1.5 \mu\text{g } \mu\text{l}^{-1}$) in the negative control are largely comparable to the amount of the ‘product’ formed (0-1.5 $\mu\text{g } \mu\text{l}^{-1}$). But anyway, those results are unexpected due to the unproper reaction condition. we have presented the new data of InsP₅ and InsP₆ (Fig. 5b-c) in the revised manuscript.

Concern6: Calibration curves by which detector response of undefined LC-MS,MS parent/daughter transitions can be calibrated are not described. Indeed, no details of the transitions: they will be different for IP₇, IP₆ and IP₅, and possibly different for different isomers, are provided.

Response: We thank the reviewer for this comment. The calibration curves for InsP₅ and InsP₆ have been presented in Supplementary Fig. 5a (with corresponding recovery rate).

Supplementary Fig. 5 *in vitro* IPK1 activity assay

a Calibrations of InsP₅ and InsP₆. 0.1, 0.2, 0.4 and 1 µg/mg of InsP₅ and InsP₆ were added to the reaction buffer (containing 10 mM Tris, pH 8.0, 60 µM ATP) and were detected with HPLC-MS/MS after 6 h of reaction. The percentage represents the recovery rate corresponding to each point.

Concern7: Beyond these analytical details, there are issues in the set up of the enzyme assays. These phosphotransfer assays with ATP and inositol phosphate substrate at sub millimolar levels is performed in 100 mM phosphate. This is an unprecedented choice of buffer for an inositol phosphate hydroxy-kinase assay. The products of these assays are simply freeze-dried and reconstituted in 80% acetonitrile. All of this is performed against a background of 100 mM phosphate.

Response: We thank the reviewer for this important suggestion. The high concentrations of phosphate and sodium truly affect the detection of InsP₆. We therefore have changed the PBS buffer with 10 mM Tris (pH 8.0), which makes the synthesized InsP₆ detectable *in vitro*.

Concern8: Have the authors undertaken any experiments to assess the recovery (solubilization) of substrates/products?

Response: We thank the reviewer for this suggestion. We have determined the recovery rates of InsP₅ and InsP₆, which are more than 77% (Supplementary Fig. 5a). The calibration curves indicate that the recovery rates are not affected by reaction time.

Concern9: For an assay performed for 6h, the assay would have been better performed with an ATP-regenerating assay. It is very likely that the multiple purified proteins added to the assay (no indication of purity is provided) might carry phosphatase activities that could degrade added ATP. Assays that allow monitoring of substrates (inositol phosphate and ATP) and products are particularly useful to assess for interfering activities. In summary, this reviewer has no confidence in any of the enzymology on which the defining claim of the manuscript, ‘Our findings therefore establish an efficient lipid-dependent IP6 biosynthesis pathway governed by PICKs, shedding new light on the mechanisms of IPs biosynthesis in eukaryote’ are made.

Response: We are grateful for the suggestion. We attempted to add the ATP-regenerating system (phosphate creatine and creatine kinase) in the reaction buffer, but found this would severely interfere the detection of InsP₆. Therefore, we gave up the option of the ATP-regenerating assay. Nevertheless, we detected the ATP content in the reaction buffer, and found it was still abundant even after 6 hours of enzymatic reaction

(Supplementary Fig. 5e), suggesting that the amount of ATP used is far in excess of that required for the reaction. In this case, even though the purified proteins might carry phosphatase activities, they would not interfere the assay for InsP₆ synthesis.

Supplementary Fig. 5 *in vitro* IPK1 activity assay

e Detection of the final ATP abundance in the reaction buffer after 6 h of reaction. Blank control presents no protein was added in reaction system.

Concern10: The manuscript lacks test that PLC, IPK2 and IPK1 constitute a pathway, a metabolic concept. The review cited (Gillaspy et al 2020) could be a starting point for construction of incisive experiments that attempt to discriminate between alternative lipid-dependent and lipid-independent pathways of InsP₆ synthesis. Indeed, if IPK1 is epistatic to IPK2 and PLC2, mutation of PLC2 alone should alter InsP₆ levels. To analyse a pathway, the authors should use methods of analysis that measure the level of pathway intermediates, not just an end product.

Response: We greatly appreciate these comments. Because a number of previous studies have proposed or demonstrated that IPK1, IPK2 and PLC are involved in the same lipid-dependent pathway of InsP₆ biosynthesis (Stevenson-Paulik et al, 2005, PNAS; Kuo et al, 2014, 2018, Plant J; Zhang et al, 2018, PCP; also shown as below Table, Freed et al, 2020, Plants), we therefore focus mainly on how these enzymes are organized and regulated in this study, instead of demonstration that they constitute a same pathway.

We appreciate the review paper (Gillaspy et al, 2020) recommended by the referee. However, it's difficult to discriminate the lipid-dependent and lipid-independent pathways by measuring the levels of intermediates, since the two pathways have similar intermediates that have identical molecular weights. Instead, we have detected the expression of key genes (e.g. *MIK* and *ITPK1*) involved in lipid-independent pathway, and found that they are comparable in WT and *C-5m* (please seed the figure below). This may in some ways favors the conclusion that PICKs regulate the lipid-dependent InsP₆ biosynthesis pathway. Furthermore, lack of IPK2 indeed results in notable reduction in InsP₆ synthesis (Zhan et al, 2015, Plant J; Fig 6h in this study). However, due to the functional redundancy of *PLC* genes, mutation of *PLC2* alone does not lead to obviously altered InsP₆ accumulation. Since the generation of high-order *plc* mutants takes quite a long time, we would prefer to reserve it for future study.

Pathway	Mutant	Impact on InsP ₆	Impact on PP-InsPs
Lipid-Dependent Pathway	plc	Nine genes; characterized single mutants have no change in InsP ₆ [32]	No Change [32]
	ipk2α *	Lethal Knock-Out [30]	Unknown
	ipk2β *	35% reduction in mass seed InsP ₆ ; no change in seedling tissue as measured by radiolabeling [30]	Unknown
	ipk1 *	83% reduction in mass seed InsP ₆ ; 93% reduction in seedlings as measured by radiolabeling [30]	InsP ₇ and InsP ₈ are reduced [33,34]
Lipid-Independent Pathway	mik	62–66% reduction in mass seed InsP ₆ [35]	Unknown
	lpa1	47–57% reduction in mass seed InsP ₆ [35]	Unknown
	itpk1–4	46% reduction in mass seed InsP ₆ in Atitpk1 mutants; no changes in Atitpk2 or Atitpk3 ; 40–51% reduction in Atitpk4 [35]	Atitpk1 and Atitpk4 have reduced InsP ₇ and InsP ₈ [33,36]
Phytate Storage	mrp5	73–80% reduction in mass seed InsP ₆ [35]; decreased InsP ₆ in seeds and vegetative tissue as measured by radiolabeling [4]	Elevated InsP ₇ and InsP ₈ [5]
PP-InsP Synthetic Pathway	vip1	No change reported [34,37]	Increased InsP ₇ and Decreased InsP ₈ [34,37]
	vip1/vip2	No change reported [37]	Increased InsP ₇ and Decreased InsP ₈ [37,38]

Table. (cited from Freed et al, 2020)

Fig. Expression analysis of *MIK* and *ITPK1*

Concern11: The manuscript provides no explanation or hypothesis how the activity of IPK1 is modulated, neither at transcriptional or post-transcriptional level.

Response: We have shown that the expression of IPK1 is comparable in WT and *C-5m* mutants (Please see the figure below), suggesting that the activity of IPK1 is not modulated at transcriptional level. Although PICKs can interact with IPK1, they cannot phosphorylate it (Fig. S4a). Nevertheless, we have demonstrated that PICKs can interact with and phosphorylate a subfamily of GRF proteins at conserved Ser residue (Fig. 4). These GRF proteins are well-known to regulate the translocation, structure, enzyme activity, or stability of target proteins via protein-protein interaction (Zhao et

al, 2021; Huang et al, 2022). We show that the phosphorylation of the GRF proteins promote their interaction with IPK1, which may alter the conformation of IPK1, in turn enhancing its activity in InsP₆ biosynthesis (Fig. 5). Therefore, we propose that PICKs regulate IPK1 mainly at post-translational level.

Fig. Expression analysis of *IPK1*

Concern12: The manuscript appears to reuse data of Fig 2a (Pi content of seeds and separately Pi content of plants, for WT, T-4m, C-5m-2) in Fig 3e.

Response: We thank the reviewer for pointing out this. Since the data related to Pi content in Fig2a and 3e are obtained on the same batch of samples, they actually share the same Pi content data of WT, T-4m, and C-5m-2. To avoid misunderstanding, we have added corresponding explanations to the legend in Fig 3e.

Concern13: The manuscript provides no methodological details for the data of Fig 7e.

Response: We thank the reviewer for pointing out this. We have provided more methodological details for data of Fig 7e in the revised manuscript.

Concern14: The analysis of inositol phosphates by PAGE after pre-concentration on TiO₂ in Figs 2c,d; 3d; 4g,h; 5e appear to be single determinations that could be prone to inconsistencies in recovery of inositol phosphates on TiO₂ and the loading of gels. The method is semi-quantitative at best.

Response: We agree that this PAGE method is semi-quantitative. Nevertheless, we independently repeated each of these assays at least three times, and obtained similar results. Furthermore, the results of Figs 2c, d have been quantitatively confirmed by HPLC-MS/MS (Fig 2f), indicating that the PAGE method is reliable.

Concern15: ‘Values are given as mean \pm SD, n = 3’, is a suffix to many of the Fig legends including eg., Fig S3. Not a single panel of Fig. S3 provides values of mean and sd.

Response: We appreciate this feedback and agree that mean \pm SD should be removed from these PAGE Fig legends. We have corrected these in the revised manuscript.

Concern16: Separately, one may assume that the numerical values on panel D (and on other gels analyzing inositol phosphates) reflect densitometry of the InsP₆ band in different genotypes relative to WT. For single lanes of the different genotypes – this could merely reflect differences in recovery/loading of sample. There is no internal calibration to an ‘invariant’ reference: indeed, only a small part of the gel is shown.

Response: We appreciate this valuable suggestion. Unfortunately, the proper internal reference has not yet been developed in this assay (Dong et al, Mol Plant, 2019, 12: 1463; Wilson et al, Open Biol, 2015, 5: 150014; Losito et al, PLoS One, 2009, doi:

10.1371/journal.pone.0005580). Nevertheless, our experiments have all been validated at least three times, so the reliability of the difference among samples still stays strongly. Moreover, the uncropped photos of gels have been provided in the supplemental data.

Reviewer #2 (Remarks to the Author):

The manuscript "A clade of receptor like cytoplasmic kinases and 14 3 3 proteins coordinate the 1 lipid originated inositol hexaphosphate biosynthesis" shows a well-designed approximation to decipher the regulatory mechanisms that underlie InsPs biosynthesis. This manuscript shows the interaction of one clade of RLCKs, named Phosphoinositide related 23 Cytoplasmic Kinases 1 with IPK1 and 2 regulating InsP6 biosynthesis and compromising inositol pyrophosphates synthesis downstream. Overall, it is a well-written and organized work, that demonstrates the importance of forming this complex together with GRF4 in order to correctly sense Pi levels and regulate InsP6 biosynthesis. Saying this, I have also some improvements to suggest to clarify some aspects of the proposed work.

Response: We thank the reviewer for the precious time and insightful comments. We have carefully considered all these comments and revised the manuscript accordingly.

Minor aspects:

Comment1: Overall the manuscript please correct the nomenclature for InsPs. For instance, InsP6 (6 in lower case) instead of IP6, and please be consistent.

Response: We thank the reviewer for this suggestion, and have corrected the nomenclature for all InsPs in the revised manuscript.

Comment2: Line 74 is missing references on *Chlamydomonas reinhardtii* vip1-1 mutant

Response: We thank the reviewer for pointing out this, and have added the corresponding literature (Line 74, Ref 24).

Comment3: Line 103 "...redundantly functioned in inositol phosphate biosynthesis" Please name them as inositol polyphosphates

Response: We have corrected this description.

Comment4: Line 153 "IP8 was might be slightly affected " Please reword this sentence

Response: This sentence has been reworded in the revised manuscript.

Comment5: Fig 1 regarding panel C AtSPX3-mCherry RFP fluorescence, please indicate in the text or legend that it is a colocalization control as SPX3 is also related to phosphate regulation in plants.

Response: We thank the reviewer for this suggestion. Because AtSPX3-mCherry is not a proper plasma membrane (PM) marker, we have used AtALMT1-mCherry instead to show the colocalization of PICKs to the PM. We have included this information in Fig 1 legend.

Major aspects:

Comment6: In order to further explore the redundancy of the PICK genes and their interaction, I wonder whether the authors had also tried C-5m with PICK6 instead of PICK5. If not, they should consider this line as an additional control.

Response: We thank the reviewer for this suggestion. In fact, we attempted to generate the *pick* sextuple mutant first instead of the quintuple mutant, by simultaneous knockout of *PICK5* and *PICK6* in *T-4m* quadruple mutant background using CRISPR/CAS9, but found the sextuple mutant is embryonic lethal. In this case, we next tried to obtain the

C-5m either containing *pick5* or *pick6* mutation from the heterozygotes of the sextuple mutant. Unfortunately, after identifying all the mutant lines (> 100), we only obtained the *C-5m* with *pick5* mutation, but failed to get the one with single *pick6* mutation, suggesting that the *PICK6* gene is much more difficult to be edited than *PICK5*. Since *PICK5* and *PICK6* have very similar tissue expression patterns (Fig. 1d), we could speculate that the *C-5m* with *pick5* mutation may have similar InsP₆ levels with that containing *pick6* mutation.

Comment7: I have some concerns regarding Fig 2 panels C and D. InsP₆ levels are highly downregulated in the mutants which are clear, especially for C5-m1 and C5-m2 (in C panel) but compared with the lower gel (seedlings) at panel D C-4m-1 and C-4m-2 should have approximately the same drop but (to my eyes) there is not such a drop in the bands of InsP₆ please authors comment on this. I would recommend including the HPLC-MS/MS quantitation of seedlings as well.

Response: We thank the reviewer for the comment. The *C-4m-1* and *C-4m-2* are *pick* quadruple mutants, which show less reduction of InsP₆ levels than *C-5m-1/2* quintuple mutants. But when compared with WT, the *C-4m-1* and *C-4m-2* mutants show a clear decrease (by ~50%) of InsP₆ accumulation in seedlings in both Fig 2C and 2D. Similar results were obtained from at least three independent experiments. Moreover, we have performed the additional HPLC-MS/MS analysis of InsP₆ content in seedlings, showing that the InsP₆ levels are significantly reduced in *C-5m* (Fig2f).

Fig. 2 IPCKs function redundantly in InsP₆ biosynthesis

f The concentration of InsP₆ in 2.4g of dry seeds and 10 g of 12-day-old seedlings detected by HPLC-MS/MS (n = 3).

Comment8: I highly recommend authors mention the results of complemented lines to support their hypothesis on the redundancy of PICK genes.

Response: We thank the reviewer for this suggestion, and have added the description of complemented lines to support our conclusion on the redundancy of PICK genes. For example, 'Both the increase of Pi accumulation and the decrease of InsP₆ levels in *T-4m* could be restored by *IPCK1* or *IPCK2*, indicating that PICK genes function redundantly in InsP₆ and Pi homeostasis'.

Comment9: I highly recommend clarifying the content of the supplemental tables regarding IP data for example including the protein ID and if possible the sequence of the peptide detected. Please name the tables as they are named in the manuscript.

Response: We appreciate this suggestion, and have provided the corresponding protein ID and peptide information in Supplementary Table 2.

Comment10: I would recommend including a negative control maybe with PICK5 or 6 in the BiFC analysis (Fig 3). Taking together all these aspects, I would recommend the publication of this manuscript after these major changes have been done.

Response: As suggested, we have included the additional BiFC analysis of PICK5/6 and IPK1 in the revised manuscript (Fig. 3a). Because PICK5 and PICK6 can also interact with IPK1, we therefore have added another negative control (SPX1) in the assay.

Fig. 3 IPCKs regulating InsP_6 biosynthesis dependently of IPK1

a BiFC assay showing the interaction between IPCK1-6 and IPK1 in protoplast. SPX1 was used as a negative control. AtSPX3-mCherry RFP fluorescence was used as colocalization marker of plasma membrane, cytoplasm and nucleus. Bar = 10 μm .

Reviewer #3 (Remarks to the Author):

Overall this is a very strong paper that in my opinion will move the field of inositol polyphosphate homeostasis and in particular InsP6 synthesis forward in a very significant manner. The findings will also obtain attention from non-plant researchers and people interested in inositol pyrophosphate signaling. The authors walked a long way, encountered redundancy at several levels that they had to overcome by generating higher order mutants (PICKs and 14-3-3 proteins/GRFs) that were then complemented. The main message that PICKs function as organizers to modulate the function of the enzymes involved in inositol phosphate synthesis is convincing. I see mostly issues with respect to the introduction and how the authors put their findings in context to what has been published in the field. I think the name “Phosphoinositide related Cytoplasmic Kinases (PICKs) is unfortunate and I hope the authors will reconsider this name (explanation below). I do also have some minor issues with respect to the experimental part. I will list the issues as they appeared to me when reading the MS, i.e. mostly in order:

Response: We thank the reviewer for the positive comments. As suggested, we have made extensive corrections to our manuscript, which are listed below. Moreover, the manuscript has also been double-checked, and the typos and grammar errors we found have been corrected.

Concern1: Line 37: First sentence of the Intro (on the roles of phytic acid in stress responses, development, phosphate homeostasis, DNA repair and membrane trafficking). The authors provide a single experimental paper on oxidative stress responses in a particular maize low phytic acid mutant by Doria et al. to reference all of the roles of phytic acid. I find this inappropriate and would like to suggest to either cite the original papers or cite one or several dedicated reviews on phytic acid.

Response: We thank the reviewer for pointing out this. As suggested, we have checked the literature carefully and provided more references (Line 39-42).

Concern2: Line 57: “phosphorylated to IP6; and a lipid independent pathway, where myo inositol (3) P1 (Ins(3)P1) from glycolysis undergoes myo inositol 1 phosphate synthase (MIPS) and a series of phosphorylations until IP6 is formed”. This sentence has a strange wording. To my understanding, MIPS proteins catalyze the conversion of D-glucose-6-phosphate to Ins(3)P.

Response: We thank the reviewer for pointing out it. The myo inositol 1 phosphate synthase (MIPS) should be replaced with myo-inositol kinase (MIK). We have made corrections in the revised manuscript (Line 60).

Concern3: Line 62: “IPK1 can also specifically phosphorylate the D-2 hydroxyl on IP6 to produce IP7 (an inositol pyrophosphate)”. This has never been shown nor has it been claimed by anyone to my knowledge. There is also no immediate reference provided by the authors. Maybe the authors refer to the Kuo et al. (2014) Plant J paper that is cited two sentences further down (citation 14)? However, Kuo et al. make no claim that IPK1 is directly involved in the phosphorylation of InsP6 to InsP7 to my knowledge.

Response: We apologize for providing this incorrect description. There is indeed no evidence indicating that IPK1 has kinase activity for synthesizing InsP7. Therefore, this incorrect description has been deleted in the revised manuscript.

Concern4: Line 65. The following sentence is strangely worded and I believe incorrect

"In addition, IPK1 cooperates with IPK2 (including IPK2 α and IPK2 β that harbor 6/3 kinase activity and sequentially phosphorylate Ins(1,4,5)P₃ to generate IP₅ via an Ins(1,3,4,6)P₄ intermediate) and ITPKs to convert the PI PLC generated or the glycolysis derived IP₃ into IP₆ by transferring phosphorylation of the phosphate group on ATP".

To my knowledge, with the exception of budding yeast, it is not known how IP₆ is produced in eukaryotes. Only the role of IPK1 in catalyzing the conversion of IP₅[2-OH] to IP₆ seems to be established in various organism. In contrast, the role of IPK2 in IP₆ synthesis and the first steps until IP₅ are poorly understood. The statements made by the authors appear to come from the important Stevenson -Paulik et al. (2005) PNAS paper which is later cited (reference 63, but should be cited here). The statement "ITPKs to convert the PI PLC generated or the glycolysis derived IP₃ into IP₆ by transferring phosphorylation of the phosphate group on ATP" is incorrect. As shown by Desfougères et al. (2019) PNAS (<https://www.pnas.org/doi/epdf/10.1073/pnas.1911431116>): Plant (and human) ITPK1 can rescue the InsP_x profile of yeast plc KO because it can use Ins(1)P₁ produced by ISC1 that acts on Inositol Phosphoryl Ceramide (IPC). ITPK1 cannot restore IP₆ in plc1 isc1 dKO strain and ITPK1 does not recognize Ins(1,4,5)P₃, a PLC product in vitro.

Response: We thank the reviewer for these insightful comments. We have corrected the statements and put them in a more conciliatory and speculative tone in the revised manuscript (Line 66-70). We have also corrected the citations according to the suggestions.

Concern5: I wonder if the authors mixed up IPK1 and ITPK1 at some point with respect to inositol pyrophosphate synthesis. ITPK1 was the first plant enzyme shown to catalyze IP₆ to 5-IP₇ conversion in vitro and in yeast (Laha et al. (2019) ACS Chem Biol (<https://pubs.acs.org/doi/full/10.1021/acscchembio.9b00423>) and W Whitfield et al (2020) Biochem J (<https://portlandpress.com/biochemj/article/477/14/2621/225707/An-ATP-responsive-metabolic-cassette-comprised-of>). First evidence that ITPK1 is indeed involved in 5-InsP₇ synthesis in planta comes from Riemer et al. (2021) Mol Plant (<https://pubmed.ncbi.nlm.nih.gov/34274522/>)

Response: We apologize for providing the incorrect description that IPK1 has kinase activity for synthesizing InsP₇. The relevant description has been deleted in the revised manuscript.

Concern6: Line 70 ". Recent studies have found that more strongly phosphorylated species exist in the IP₆ derived inositol pyrophosphates IP₇ and IP₈ through di phosphoinositol pentakisphosphate kinases VIH1 / 2 (named after VIP1 homologs), lack of which leads to constitutive Pi starvation response and impaired plant growth and 73 development 15,16 .

I think something got mixed up with the references here.

The first evidence of inositol pyrophosphates in plants comes to my knowledge from Brearley and Hanke (1996) Biochem. J. (<https://pubmed.ncbi.nlm.nih.gov/8761483/>) and Lemtiri-Chlieh et al. (2000) PNAS (<https://www.pnas.org/doi/abs/10.1073/pnas.140217497?doi=10.1073/pnas.140217497>). First description of plant enzymes involved in their synthesis from Desai et al (2014) Plant J (in vitro and yeast) and Laha et al (2015) Plant Cell (in vitro, yeast and in planta). First description of enzyme involved in plant 5-InsP₇ synthesis by Riemer et al (2021). The papers reporting that vih1 vih2 double mutants result in PSR are Dong et al (2019) Mol Plant (now citation

22) and Zhu et al (2019) elife (now citation 19).

Response: We appreciate this feedback and agree that the citation of references is a bit confusing. We have corrected these information in the revised manuscript.

Concern7: Line 76 "... These findings also reveal that Pi homeostasis in plants is regulated by kinases involved in IPs synthesis, most likely due to their indirect contribution to the synthesis of IP₈" Here two more papers should be cited: the Stevenson-Paulik (2005) PNAS paper 2005 (see above) which was the first paper reporting a PSR response of a mutant defective in InsP₆ synthesis and Riemer et al. (2021) Mol Plant paper that showed ITPK1 controls phosphate starvation responses by catalyzing the first step in inositol pyrophosphate synthesis, i.e. 5-IP₇, the precursor of IP₈ (see above).

Response: We agree with this suggestion, and have cited these papers in the revised manuscript.

Concern8: Line 104: "RLCK V subfamily has not been documented in terms of functions. Here we showed that these two kinases together with other related RLCK V subfamily members redundantly functioned in inositol phosphate biosynthesis. We therefore named them Phosphoinositide related Cytoplasmic Kinases 1 6 (PICK1 PICK6)." I find this name very unfortunate and I strongly suggest to reconsider it! Phosphoinositides are (membrane) lipids (!) generated by phosphorylation of the inositol headgroup of phosphatidylinositol. They have a typical phospholipid structure (headgroup, glycerol backbone, acyl chain(s) and should not be confused with inositol phosphates which lack the glycerol backbone and acyl chains.

Response: We thank the reviewer for pointing out this, and have changed the name 'PICK' to 'IPCK' (Inositol Polyphosphate related Cytoplasmic Kinases) throughout the manuscript.

Concern9: Line 110 "In the preliminary study, we employed a quantitative phosphoproteomics using wild type (WT) plants under control (1.25 mM Pi) and Pi deficiency (10 μ M Pi) treatments for 1 h, to identify potential regulators in Pi homeostasis". I wonder how this experiment was done. Where plants transferred from a hydroponic + P culture to a hydroponic minus P culture? Or where they transferred from +P to minus P solidified (MS based) media? I wonder if the authors agree that Pi deficiency cannot be induced in one hour. Maybe the authors should reword this: 'Challenge of P sufficient plants with low P (10 μ M) growth medium', something along these lines. If seedlings were transferred between solidified media (e.g. MS-plates), how can they say that changes in the phospho-proteome is related to the low P content of the media and not just a consequence of injury (damage of root hairs etc.) due to transfer? Was there a control in which plants were transferred from plus to plus P medium to account for plant responses to damage or handling? Along the same line of thought: The sentence in line 366 "PICK1 and PICK2 were initially identified because of their reduced phosphorylation levels under Pi deficiency (Supplementary Fig. 1a)" is probably misleading. If the treatment was really only done for one hour, I think one cannot call this treatment P deficiency.

Response: We thank the reviewer for this concern. We would like to clarify the original purpose of this quantitative phosphoproteomics that was not specified in the manuscript. Indeed, it is now well-known that plants can sense InsP₈ to modulate Pi starvation responses and signaling, but whether and how they directly sense Pi are still unknown. We speculate that there could be a plasma membrane-localized machinery that can

sense the extracellular Pi levels. Aiming to identify the potential proteins involved in sensing extracellular Pi deficiency, we performed the phosphoproteomics by using hydroponically cultured 6-week-old plants treated with +Pi or -Pi condition for 1 hr (the control plants were transferred from +Pi to -Pi condition). We do think that 1 hr of -Pi treatment may not induce the *in vivo* Pi starvation in plants, which can help us identify the potential proteins that we are interested in. Since we focused mainly on the plasma membrane proteins, we therefore identified the RLCKs in this study. We agree with the reviewer that use of “Pi deficiency” is not proper here, and have revised the corresponding description in the manuscript.

Concern10: Line 115 and Fig 1 (and respective Fig legend). The authors should not use red and green as many people are colorblind. They could combine red with blue or cyan or combine green with magenta or orange (something that should be easily done with the images they have).

Response: We thank the reviewer for this good suggestion, and have modified the figures with proper color combination in the revised manuscript.

Concern11: For clarity, authors should indicate in the Fig legend that the green box represents a transmembrane domain. They should also explain why they used AtSPX3 (as a cytosolic marker?).

Response: We have provided explanations for the green box and AtSPX3-mCherry in the Fig legend in the revised manuscript.

Concern12: Lines 135: the authors should mention how they measured P – I cannot find this in the method section.

Response: We used the ICP-MS to measure the P content of plants, as shown in Line 501.

Concern13: Line 142: e). “Both the increase of Pi accumulation and 142 the decrease of IP6 content in T-4m could be restored by PICK1 or PICK2 (Fig. 2a, 2d).” The authors should mention under control of which promoter PICK1 and PICK2 were expressed for this experiment. I could not find the information in the method part (it should be mentioned there but also here).

Response: We thank the reviewer for pointing out it, and have added the promoter information in the legend of Fig 2.

Concern14: Line 159 “... finding IPK1, the key synthase of IP6 / IP7, to be a good candidate”. See also comment above: No evidence that IPK1 is directly mediating IP7 synthesis by any previous work to my knowledge.

Response: As mentioned above, we have corrected these descriptions in the revised manuscript.

Concern15: Line 208: “...the kinase domain could phosphorylate GRF4 in vitro , while a kinase dead mutation 208 (PICK 1 K182E) abolished this phosphorylation (Supplementary Fig. 3b I find this not so convincing as there is much less protein of the catalytic dead mutant version as compared to wt.

Response: We thank the reviewer for this concern. Actually, the amount of WT and mutated PICK1 are comparable in the assay (please see the Coomassie Brilliant Blue (CBB) staining in the bottom panel). We guess the reviewer could be misguided by the auto-phosphorylation band in the upper panel. Because the PICK1 K182E mutation causes its loss of kinase activity, we can observe that the autophosphorylation of

mutated variant is remarkably reduced compared with that of WT.

Concern16: Line 242/Fig 5. In Fig 5b on the very top "...GRF4S242" does not indicate whether it refers to S242A or the phosphomimic S242D protein variant.

Response: We thank the reviewer for pointing out this. It refers to S242A. The data of Fig 5 has been updated in the revised manuscript.

Concern17: Line 278: PLC activity as well. "...Indeed, we found that PI PLC2 which belongs to PI PLCs subfamily and functions in IP₃ synthesis..." To my knowledge its under debate whether it is IP₂ or IP₃ synthesis in plants. Both might play a role as suggested by work of Teun Munnik (https://link.springer.com/chapter/10.1007/978-3-642-42011-5_2; <https://academic.oup.com/pcp/article/59/3/469/4772709?login=true>

Response: We thank the reviewer for the comment. Although there is still controversy over whether PI-PLC hydrolyzes PIP₂ to produce InsP₂ or InsP₃ in plants, Dowd et al (2006, *Plant Cell*, 18: 1438) demonstrated that the *Petunia* PI-PLC1 can hydrolyze PI(4,5)P₂ into IP₃ and DAG during pollen tube growth, which may in some ways support the potential role of PI-PLCs in InsP₃ biosynthesis. Moreover, Lee et al (1996, *Plant Physiol*, doi: 10.1104/pp.110.3.987) found that ABA-induced plant PI-PLCs kinase activity can result in a large drop in PIP₂ and an increase in InsP₃. In addition, as shown by Singh et al (2015, *Cell Calcium*, 58: 139), the plant PLC class has been subdivided into a well-studied phosphatidylinositol-specific PLCs (PI-PLCs) and a recently identified phosphatidylcholine-PLC (PC-PLC) groups. PI-PLC acts upon a specific substrate, PI (4,5)P₂ at glycerophosphate ester linkage of membrane phospholipids and leads to generation of secondary messengers such as diacylglycerol (DAG) and inositol 1,4,5-trisphosphate (InsP₃).

Nevertheless, to make it less confusing, we have revised this sentence to "...Indeed, we found that PI-PLC2 which belongs to PI-PLCs subfamily and potentially functions in IP₃ synthesis...".

Concern18: Line 285: "...PLC2 via phosphorylation of GRF4 S242 residue, to form a 285 PICK1 GRF4 PI PLC2 complex as did for IPK1 and IPK2 β (Fig. 7d)..". If I am not mistaken, several negative controls are missing in this Fig.

Response: We thank the reviewer for this concern. Actually, this experiment aims to determine whether GRF4 phosphorylation conferred by PICK1 promotes its interaction with IPK1. Therefore, the combination containing 35S-flag empty vector can be set as a negative control.

Concern19: Line 523: "IPK1 activity assay was performed as previous described with some modifications" I wonder whether Dong et al really carried out IPK1 activity assays, or do the authors refer to activity assay of the InsP₇ kinase VIH?

Response: Dong et al did the activity assay of VIH, but not IPK1. We now find that the reaction system used for VIH kinase assay is inapplicable for IPK1 activity assay, and have modified the reaction buffer as described in the methods of revised manuscript.

Concern20: Line 321. I would like to urge the authors to not repeat unfounded claims about "IP₃ signaling" in plants. I suggest reading a small section written in the paper Zhang et al (<https://academic.oup.com/pcp/article/59/3/469/4772709?login=true>) by the lab of Teun Munnik which boils this down better than any other reference I am aware of and provides many relevant citations. I am citing from this paper: "Much less is clear about the PLC signaling system in plants (Ischebeck et al. 2010, Munnik 2014, Heilmann 2016, Heilmann and Ischebeck 2016, Gerth et al. 2017). Initially, it was

thought to be equivalent to the animal paradigm since most of the components driving the pathway were thought to be present (Munnik et al. 1998a, Stevenson et al. 2000, Meijer and Munnik 2003), especially when microinjected IP₃ was shown to release Ca²⁺ from an intracellular store and to induce stomatal closure (Blatt et al. 1990, Gilroy et al. 1990, Allen and Sanders 1994, Hunt and Gray 2001). However, 20 years later, Brearley's lab provided evidence that it was not IP₃, but its subsequent conversion into IP₆, that caused these effects (Lemtiri-Chlieh et al. 2000, Lemtiri-Chlieh et al. 2003). Similarly, not DAG but its phosphorylated product, phosphatidic acid (PA), has been emerging as the plant lipid second messenger (Munnik 2001, Laxalt and Munnik 2002, Testerink and Munnik 2005, Arisz et al. 2009, Testerink and Munnik 2011, McLoughlin and Testerink 2013, Pokotylo et al. 2014, Munnik 2014, Hou et al. 2016, Vermeer et al. 2017). Moreover, genome sequencing has meanwhile confirmed that flowering plants lack homologs of both the IP₃ receptor and PKC (Wheeler and Brownlee 2008, Munnik and Testerink 2009, Munnik and Vermeer 2010, Munnik and Nielsen 2011, Munnik 2014, Heilmann 2016, Gerth et al. 2017)."

Response: We greatly appreciate these comments, and agree that there is no evidence to imply that InsP₃ participates in the signaling pathway as a signal in plant. We have checked the whole paper carefully and made revisions accordingly.

Concern21: Authors should also check for typos and grammar (overall it is well written though).

Response: We have polished the English extensively for the revised manuscript.

Reviewer #4 (Remarks to the Author):

In this manuscript, Xu and colleagues report the identification of a group of receptor-like cytoplasmic kinases (RLCKs) that is proposed to recruit multi-protein complex that can modulate the synthesis of different inositol polyphosphates (IPs), including inositol hexakisphosphate (phytic acid or IP6). The study started by the finding that phosphorus (P) deficiency inhibits the phosphorylation of two members of the subfamily V of RLCKs. Closely related members of this subfamily show comparable expression at the tissue level. Disruption of four or five of these genes (named as PICK1 to 5) resulted in increased phosphate concentrations in seeds and in whole seedlings, while a sextuple mutant is likely embryo lethal. This phenotype was accompanied by decreased levels of IP6 and inositol pentaphosphate (IP5), and upregulation of P deficiency-induced genes. Using BiFC, in vitro pull-down and split-LUC assays, the authors found that PICK1-4 can physically interact with IPK1, the main kinase responsible for IP6 synthesis, and with IPK2 α and β , which form IP4 and IP5. While PICKs cannot phosphorylate IPK1, IP-MS identified GRFs (14-3-3 proteins) as a putative interacting proteins with PICK1, which was confirmed with BiFC and split-LUC. The authors then show that GRF4 can be phosphorylated at Ser242 by PICK1, and interact with IPK1. According to semi-quantitative PAGE, *grf3grf4* double mutant has decreased IP6 levels, which can be rescued with the native GRF4 protein but not with a Ser242Ala variant. Next, the authors attempted to demonstrate in vitro IP6 biosynthesis activity of a PICK1- GRF4-IPK1 recombinant proteins. Although IP5 was supplied as substrate, surprisingly only IP7 but not IP6 was detected, an activity that was increased when IPK1, PICK1 and GRF4 (native or as a phosphomimicking variant) were present together. Finally, the authors show that PICKs and GRF4 can physically interact with PI-PLC2 and 7, which are involved in lipid-dependent IP3 synthesis. In conclusion, the authors suggest that a large protein complex formed by PICKs, GRF4, IPK1, IPK2 and IP-PLCs modulates lipid-dependent IP6 biosynthesis. The reported findings are novel and of potential relevance as they reveal an unknown mechanism regulating inositol polyphosphate biosynthesis in plant cells. The manuscript presents an impressive amount of data and identifies several interactions among known and previously uncharacterized proteins. While the protein-protein interactions assays are validated with independent approaches, they heavily rely on transient expression with constitutive promoters. Furthermore, the biological context in which the proposed interactions occur, and how and when they control the synthesis of specific IPs are less clear. Points that need further attention by the authors are detailed below.

Response: We thank the reviewer for these constructive comments that are very helpful for improving this manuscript. According to reviewers' suggestions, we have made extensive modifications to our manuscript and provided additional data to make our results convincing. Particularly, we have detected the IPK1-flag protein abundance on the plasma membrane (PM) in both WT and *pick* quadruple mutant (*T-4m*) backgrounds, and found that the relative abundance of PM-associated IPK1 is obviously reduced in *T-4m* versus WT, indicating the key role of PICKs in the PM-associated complex with IPK, and to some extents, confirming the *in vivo* interaction of PICKs and IPK1 and strengthening the model that PICKs recruit IPK1/IPK2s/PI-PLC to be a complex to establish an efficient InsP6 biosynthesis pathway. Furthermore, we have clarified the biological context in which how this complex works. The manuscript has also been double-checked, and the typos and grammar errors we found have been corrected.

Major comments:

Comment1: The study contains a wealth of data regarding the interaction of several proteins and present some elegant evidence of how these interactions can regulate the function of IP kinases. However, one weak point is that these interactions are presented without a clear biological context. As a consequence, the model derived from the study (Fig. 8) shows the identified protein complex loosely connected to known biosynthesis or signaling pathways. For instance, while the manuscript starts with the discovery that the phosphorylation of PICK1 and 2 is inhibited in response to P deficiency, the P status-dependent context is not further worked out during the study. Is PICK1 phosphorylation relevant at all and required for its interaction with GRF4? Does it affect the ability of PICKs to recruit other proteins? If yes, then is the formation of the proposed complex altered according to the cell's P status?

Response: We thank the reviewer for these insightful concerns. Actually, as mentioned in the discussion, PICKs required for InsP₆ biosynthesis have at least two functions. On one hand, we propose that PICKs act as organizers to recruit IPK1, IPK2s and PLCs to form a complex, therefore establishing an efficient InsP₆ biosynthesis pathway. We do think this complex may function throughout the life cycle, but not just in a certain biological context, since the *pick* quintuple mutant shows a similar retarded growth with *ipk1* loss-of-function mutant (Fig. 3g), and the *pick* sextuple mutant is embryonic lethal, phenocopying the *ipk2α ipk2β* double mutant (Fig. S1i; Zhan et al, 2015). Consistently, PICKs display similar tissue expression patterns with *IPK1*, *IPK2* and *PLC* (Fig. S7). Therefore, the role of PICKs in InsP₆ biosynthesis could be important enzymatic components more than just regulators.

On the other hand, our preliminary data show that PICKs may play a role in regulation of cytosolic Pi homeostasis in response to extracellular Pi deficiency. We need to explain here the initial purpose of the quantitative phosphoproteomics that had not been clarified in the manuscript (please see the response to the 9th point of Reviewer#3). We found that the extracellular Pi deficiency rapidly repressed the phosphorylation of PICK1 and PICK2 at certain Ser residues (Fig. S1a), and that mutation of these Ser indeed dramatically inhibited the PICK1/2-dependent S242 phosphorylation of GRF4 *in vitro* (Fig. S8). We have also provided additional data that Pi deficiency truly represses the *in vivo* phosphorylation of GRF S242 (please see the below figure). Since the phosphorylation of S242 promotes the interaction of GRF4 with IPK1/2/PLC2 (Fig. 5a, 6f, 7d), we hence believe that the extracellular Pi status may influence the activity of PICKs on GRF phosphorylation, thereby modulating InsP₆ biosynthesis and Pi homeostasis in the cytosol by regulation of the activity of IPK1/2/PLCs.

Because this manuscript mainly focuses on the InsP₆ biosynthetic mechanism as stated in the title, how the complex contributed by PICKs is responsive to Pi deficiency in detail has been actually beyond the scope of this manuscript. While we appreciate the concerns raised by the reviewer, we would prefer to reserve full consideration of such topic for a future manuscript.

Fig. *in vivo* phosphorylation of GRF4 in response to Pi deficiency

Comment2: Another weak point is that the shown protein-protein interactions derived almost exclusively from transient expression (with constitutive promoters) or in vitro assays. This is insufficient. At least the key interactions should be also demonstrated in planta (e.g. by co-IP) using the native promoters to drive the expression of the candidate genes. A similar weakness concerns the existence of the proposed protein complex. To demonstrate that a PICK-recruited complex is indeed formed at the plasma membrane, it is necessary to show that e.g., IPK1 or IPK2s are associated to the plasma-membrane and that this localization is disturbed in pick quadruple or quintuple mutants.

Response: We thank the reviewer for these valuable suggestions. Actually, most of the interacting proteins (e.g. GRFs, IPK2 α , IPK2 β , PI-PLCs) of PICKs were identified from IP/MS from planta, which is already an *in vivo* protein interaction assay. We agree that it's more reasonable to verify the protein-protein interactions in planta using the native promoters to drive the expression of the candidate genes. Nevertheless, these experiments require the de novo generation of many additional genetic materials, and will take at least one year that may affect the timeliness of this study. Therefore, we would like to consult with the reviewer to omit these experiments. That will be greatly appreciated.

We also appreciate the very good suggestion to determine if the association of IPK1/2 to the plasma membrane is disturbed in *pick* quadruple or quintuple mutants. As shown in Fig 7h, we have detected the IPK1-flag protein abundance on the plasma membrane (PM) in both WT and *T-4m* backgrounds, finding that the relative abundance of PM-associated IPK1 is reduced (by > 30%) in *T-4m* versus WT (the number in the panel represents the relative proportion of abundance of PM-associated IPK1 proteins to IPK1 total proteins, which are corrected by internal references, respectively). Similar results have been obtained from at least three independent experiments. Because *IPK1-flag* is driven by *35S* promoter that has a much broader expression than *PICKs* in WT, the reduction of PM-associated IPK1 in *T-4m* is actually underestimated. Overall, this result demonstrates that PICKs indeed recruit IPK1 to the PM *in vivo*, favoring the hypothesis that PICKs recruit IPK1, IPK2s and PI-PLC to form a complex to establish an efficient lipid-dependent InsP₆ biosynthesis pathway. This result may also in some ways demonstrate the *in vivo* interaction between PICKs and IPK1 as mentioned above.

Fig 7h. The expression of IPK1 on the plasma membrane

h Detection of IPK1 protein abundance on the plasma membrane (PM) under WT and *T-4m* backgrounds. Actin and BRI1 serve as internal references for total protein and plasma membrane protein, respectively. WT was used as a negative control. The number in the panel represents the relative proportion of abundance of PM-associated IPK1 proteins to IPK1 total proteins, which are corrected by internal references, respectively.

Comment3: I am also not satisfied with the analysis of IPs. For the most part, the manuscript mainly report IP levels as relative changes based on quantification of band intensities from SDS-PAGE gels. A major shortcoming of this approach and even of the

HPLC-MS/MS method used in the study, is that only IP5 and IP6 but no other IP could be detected. It is not even clear whether the IP5 species that serves as substrate for IP6 synthesis (i.e., 2-OH IP5) was the IP5 species disturbed by the loss of PICKs. Furthermore, none of the IP concentrations presented, irrespective of the method used, are normalized to seed or plant mass nor compared with a statistical method. A more thorough characterization of the IP profile, at least of the pick mutants, should be additionally carried out by SAX-HPLC or CE-MS as e.g., Kuo et al., 2018 doi.org/10.1111/tpj.13974 and Riemer et al., 2021 doi.org/10.1016/j.molp.2021.07.011. This would significantly strengthen the manuscript by more directly determining the steps in IP6 synthesis that are modulated by PICKs, and by providing much needed direct support to clarify whether PICKs do indeed control the lipid-dependent route of IP6 synthesis, as proposed in the model and in the manuscript title.

Response: We thank the reviewer for this valuable suggestion. Actually, HPLC-MS/MS is still a reliable method to detect InsPs (Zajdel A et al. Phytic acid inhibits lipid peroxidation in vitro. Biomed Res Int. 2013. doi: 10.1155/2013/147307; Lam G et al. Detection of myo-inositol tris pyrophosphate (ITPP) in equine following an administration of ITPP. Drug Test Anal. 2014. doi: 10.1002/dta.1473; Paraskova JV et al. Speciation of inositol phosphates in lake sediments by ion-exchange chromatography coupled with mass spectrometry, inductively coupled plasma atomic emission spectroscopy, and ^{31}P NMR spectroscopy. Anal Chem. 2015. doi: 10.1021/ac5033484.). The reason why we failed to detect the $\text{InsP}_{3/4/7/8}$ is that the amount of samples used for HPLC-MS/MS was insufficient, since the content of these InsPs is much lower than $\text{InsP}_{5/6}$ in plant. Because we lack the conditions to do SAX-HPLC or CE-ESI-MS assay as suggested by the reviewer, we have therefore optimized the condition of HPLC-MS/MS and used enough samples for the analysis, and have now successfully detected $\text{InsP}_{3/4/7/8}$ in seedlings. We show that the accumulation of $\text{InsP}_{3/5/6/8}$ is obviously reduced in *pick* quintuple mutant (Fig. 2f), indicating that PICKs regulate multiple steps of InsP_6 biosynthesis and favoring the conclusion that they control the lipid-dependent route of InsP_6 synthesis. In addition, we have clarified in the manuscript that the InsP_5 species used for InsP_6 synthesis is 2-OH IP5.

Supplementary Fig. 2 HPLC-MS/MS detection for InsPs

b The concentration of InsP_3 / InsP_4 / InsP_5 / InsP_7 / InsP_8 in 2.4 g of dry seeds and 10 g of 12-day-old seedlings detected by HPLC-MS/MS (n = 3). The concentration of InsP_6 belongs to the same batch of samples as shown in Fig. 2f.

Comment4: For the most part the manuscript is poorly written and provide insufficient methodological information to understand the results or to follow the rational behind the selection of candidate proteins. i) To improve the grammar and logic of the text, the authors should seek assistance of professional language editing or from a colleague with good command of English. ii) To facilitate understanding and interpretation of the results, more details are necessary in Material and Methods, including the name of all plasmids used in the different experiments/assays, gRNA sequences and types of mutations obtained, detailed description of protocols instead of referring to published studies, and instrumentation (e.g., brand and model of ICP-MS). iii) To make the Figs self-explanatory, more experimental information must be included in the legends, as is done in articles published in Nat Commun.

Response: We thank the reviewer for these suggestions. The English of the text has been polished by colleague with good command of English. At the same time, we have included more experimental details in Material and Methods, and provided more detailed explanations in figure legends according to reviewer's comments.

Comment5: When attempting to investigate IPK1 activity when supplying IP5 as a substrate, the authors only detected IP7 but not the expected direct product IP6. They simple concluded that IP6 was immediately converted to IP7 (lines 242-243). This is not acceptable. Does it mean that IPK1 has also IPK6 kinase activity, and that this activity is so high that all generated IP6 is consumed to generate IP7? That would go against published work. Without a clear characterization of the reason behind this unexpected result, the results shown in Fig. 5b-d are meaningless and definitely cannot be use to conclude about IPK1 activity. Furthermore, what IP5 species was provided as substrate in the in vitro assay? IPK1 can phosphorylate 2OH-IP5 but not other IP5 forms.

Response: We thank the reviewer for pointing out this. We agree that IPK1 is unlikely to convert InsP₆ to InsP₇, and the detection of InsP₇ is indeed an unexpected result due to unknown reason in the *in vitro* IPK1 activity assay (please see the responses to reviewer 1's concern 1/2/4). We have optimized the reaction and successfully detected the generation of InsP₆, but not InsP₇ *in vitro* (Fig. 5b-c). The related description has been corrected in the revised manuscript. Furthermore, 2OH-IP5 (D-myo-Inositol-1,3,4,5,6-pentaphosphate) was used as substrate for IPK1 kinase assay.

Fig. 5 IPCK1 facilitates IPK1 activity by phosphorylating GRF4

b Chromatogram absorption diagram. The abscissa represents the sample retention time, and the ordinate represents the peak area of the sample chromatogram. Different color absorption peaks and areas correspond to various protein combinations. ddH₂O and 2.5 µg of PSKR1 were used as the blank control and negative control, respectively. **c** The concentration of InsP₅ / InsP₆ converted from absorption peak area in **b**. Values are mean ± SD, n = 3.

Comment6: IP6 is a major P-storage compound in seeds and the lipid-independent pathway has been proposed to dominate IP6 biosynthesis in seeds. However, the results from the present manuscript suggest that a lipid-dependent route also makes a large contribution. How can membrane-derived phosphatidylinositol supply enough substrate to support the synthesis of large amounts of IP6 stored in seeds? Is there a possibility that PICKs also (or even mainly) affect the lipid-independent pathway?

Response: We thank the reviewer for this insightful comment. Because PICKs are localized to the plasma membrane and recruit IPK1/IPK2s/PLCs instead of ITPK1 (Supplementary Fig. 6), we therefore speculate that they are more likely to affect the lipid-dependent pathway than the lipid-independent one. Nevertheless, we cannot rule out the possibility that PICKs may also affect lipid-independent InsP₆ biosynthetic pathway. We have included this point in the discussion.

Comment7: Following the last comment and considering the importance of these questions to contextualize the identified mechanism, it is necessary to determine whether the lipid-independent pathway is also modulated by PICKs. The absence of physical interaction with proteins involved with the lipid-independent pathway does not automatically preclude that a direct or even indirect control can occur. At least the expression of MIPs and the analyses of intermediates formed by the lipid-independent route (see also comment 2) should be compared in WT and pick multiple mutants.

Response: We thank the reviewer for these valuable suggestions. Since the lipid-dependent and lipid-independent pathways have similar intermediates that have identical molecular weights, it's indeed difficult to discriminate the two pathways by measuring the levels of those intermediates. We therefore have detected the expression of *MIK* and *ITPK1*, two key genes involved in lipid-independent pathway, and found that they are comparable in WT and *pick* multiple mutants (please see the figure below). This may in some ways favor the conclusion that PICKs regulate the lipid-dependent InsP₆ biosynthesis pathway. Nevertheless, we still cannot rule out the possibility that PICKs may also affect the lipid-independent InsP₆ biosynthetic pathway. We have included this in the revised manuscript as mentioned in the last point.

Fig. Expression analysis of *MIK* and *ITPK1*

Comment8: Protein phosphorylation usually means control of activity. The authors show that PICK1 can phosphorylate GRF4 and that the phosphorylation level of GRF4's S242 is decreased in the pick quadruple mutant. However, is GRF4 phosphorylation at all modulated? If yes, and if it is important to modulate IP6 synthesis, suggested by the authors (e.g., model in Fig. 8), then how does it respond to the P nutritional status? Or is it developmentally controlled being e.g., activated during seed development when P reserves in the form of IP6 are generated?

Response: We appreciate these concerns. We have demonstrated that the phosphorylation of GRF4 S242 is responsive to the Pi status *in vivo* (please see the response to concern 1). Because the S242 mutated *GRF4* cannot rescue the InsP₆ accumulation in *grf3 grf4* double mutant in both seeds and seedlings (Fig. 4g), we would speculate that the phosphorylation of GRF4 S242 may also be developmentally controlled. In line with this, the mRNA levels of *PICKs* are abundant during seed development (Fig 1d; Supplementary Fig. 7).

Comment9: The altered expression of some Pi deficiency-induced genes cannot be used to imply that IP8 accumulation is reduced in the pick quadruple and quintuple mutants, as written in lines 152-153. To conclude about the level of any inositol pyrophosphate, such as IP8, their concentrations must be determined. Without these data, even the connection of the proposed protein complex with IP8-mediated Pi signaling, as shown in the model (Fig. 8), is too speculative.

Response: We thank this suggestion. As mentioned in the response to concern 3, we have detected the content of InsP_{3/4/5/6/7/8}, and found that the accumulation of InsP₈ is indeed reduced in *pick* quintuple mutant versus WT. We have included these data in the revised manuscript (Fig. 2f).

Comment10: The subcellular localization data shown in Fig. 1c are insufficient to prove that PICKs are present in the plasma-membrane (PM). A PM-specific dye or fusion protein should be used for the co-localization. Some green signal is also overlapping with SPX3:mCherry, even in what seems to be the nucleus (e.g., PICK5). Please check if the selected GFP filter does completely block mCherry-derived signal from passing through.

Response: We thank the reviewer for this good suggestion, and have included the new subcellular localization data with PM-specific marker (AtALMT1-mCherry) in the revised manuscript (Fig 1c).

Fig. 1 Expression patterns of IPCKs.

c Expression and localization of IPCK(1-6)-GFP fusions in wild-type (WT) mesophyll protoplasts. AtALMT1-mCherry RFP fluorescence was used as a plasma membrane marker. Columns 1-4 indicate GFP signals, bright-field differential interference contrast (DIC), and merged images of GFP and RFP, respectively. Bar = 10 μ m.

Comment11: Although the authors provide some evidence that the many interactions that they identified may indeed occur in planta, apart from PICK1-6, no tissue-specific localization is shown for GRF4, IPK1, IPK2s, and PI-PLC2,7 nor reference to other studies is provided. Thus, it remains unclear whether the different proteins that can be recruited by PICKs are even present in the same cells at the same time. This is key to demonstrate not only whether the interactions occur, but also where and under what conditions.

Response: We appreciate this concern. Based on the known tissue localization data, *IPK1* indeed has very similar tissue expression pattern with *PICKs*, as was revealed previously and in this study (Kuo et al., 2018, Plant J; Fig. 1d). Furthermore, using the public transcriptomic data in BAR website (<https://bar.utoronto.ca/eplant/>), we have shown that *PICK1/2*, *GRF4*, *IPK1*, *IPK2* and *PLC2* have spatial co-expression in many tissues of Arabidopsis, such as in developing seeds, supporting the *in vivo* interactions. We have included this result in the revised manuscript (shown as Supplementary Fig. 7).

Supplementary Fig. 7 Display of gene expression in different organs from online databases (<https://bar.utoronto.ca/eplant/>).

Comment12: According to the proposed model, many IP kinases would assemble as a large complex at the cytosol-facing side of the plasma-membrane to generate different IPs. If assemble in such a way, then how do the authors envisage that intermediates, such as IP4 and IP5 can still accumulate instead of the reactions always ending up in IP6? This aspect should at least be discussed.

Response: We appreciate this comment. Actually, not all of the IPK1 and IPK2s are recruited to the plasma membrane by PICKs, since they have high expression in the cytosol as shown in the subcellular localization (Supplementary Fig. 3a). Therefore, the free IPK2s possibly with lower activity may contribute to the biosynthesis of InsP₄ and InsP₅. We have included this point in the discussion.

Comment13: The criteria used for the selection of candidates in split-ubiquitin membrane Y2H screens or immunoprecipitation MS are not clearly described in the text. For instance, in lines 198-199, the authors write that "...GRF4 was found to be most abundant in PICK1 immunoprecipitants.". In the Excel sheet, GRF4 appears with a PepCount value of 20, below many other proteins. Were any of the other top proteins tested as well? Furthermore, it is not even clear why PICK1 and PICK2 were selected in the first place, as they were not the only proteins, whose phosphorylation was inhibited in response to P deficiency. Please provide more clear explanations for the

selection of these candidates.

Response: We thank the reviewer for these concerns. For the reason why PICK1 and PICK2 were selected in the first place, we would like first to clarify the original purpose of the quantitative phosphoproteomics that was not specified in the manuscript. Indeed, it is now well-known that plants can sense InsP_8 to modulate Pi starvation responses and signaling, but whether and how they directly sense Pi are still unknown. We speculate that there could be a plasma membrane-localized machinery that can sense the extracellular Pi levels. Aiming to identify the potential proteins involved in sensing extracellular Pi deficiency, we performed the phosphoproteomics by using hydroponically cultured 6-week-old plants treated with +Pi or -Pi condition for 1 hr. We do think that 1 hr of -Pi treatment may not induce the *in vivo* Pi starvation in plants, which can help us identify the potential proteins that we are interested in. Since we focused mainly on the plasma membrane proteins, we therefore identified the RLCKs in this study, which are localized to the PM and potentially mediate the transduction of extracellular signal from cell surface to the cytosol.

For the reason why GRF4 was selected from the IP/MS data, we would like to explain what has been misunderstood in the manuscript. The logic is as follows. Because PICK1 could not phosphorylate IPK1 *in vitro* (Supplementary Fig. 4a), we next wondered if there are intermediate molecules or chaperones that help PICK1 with the regulation of IPK1 activity. Since GRFs have been shown to be able to directly bind and modulate the activity of target proteins after activation by upstream regulators (Zhao et al, 2021, doi: 10.1111/plb.13268; Rashaun et al, 2016, doi: 10.3389/fpls.2016.0061; Huang et al, 2022, doi: 10.1007/s00299-021-02803-4), they appear to be the ideal candidate intermediating PICK1-IPK1 regulation. Among GRF proteins identified in PICK1 immunoprecipitants, GRF4 was found to be most abundant, and was therefore selected as a representative for further analysis. We have revised the related description in the manuscript.

Comment14: Considering that PICKs are required to recruit IPK1 via GRF4 phosphorylation, then how to interpret that overexpression of IPK1 or GRF4 can restore IP6 concentrations and Pi accumulation in the pick quadruple mutant?

Response: We thank the reviewer for pointing out this. As mentioned in the response to point 12, not all IPK1 are recruited by PICKs, and the free IPK1 in the cytosol possibly with low activity may also contribute to InsP_6 synthesis. Furthermore, the *pick* quadruple mutant is a mild mutant in InsP_6 reduction, in which PICK5 and PICK6 are still active and the expression of *PICK5* is even promoted (Supplementary Fig. 1h). Therefore, overexpression of IPK1 or GRF4 may bypass overcome the reduction of PICKs and restore InsP_6 and Pi accumulation in this quadruple mutant.

Minor points:

Comment15: The files containing the reported “Supplementary Tables” should have been named as such. It was difficult to find out to what specific Table each Excel sheet was referring to.

Response: Thanks for pointing out it. We have corrected the file name accordingly.

Comment16: Apart from showing that a PICK-GRF4-IPK1-IPK2-PLC multi protein complex can be formed, its connection to known pathways shown in the model of Fig. 8 are less clear. For example, the authors represent IP6 mainly in the context of Pi

signaling. However, their results indicate that IP6 levels are decreased also in seeds, where IP6 may have mainly a role as a P-storage form.

Response: As stated in the response to this reviewer's first point, this manuscript mainly focuses on the InsP₆ biosynthetic mechanism as described in the title, how the PICKs-dependent complex is responsive to Pi deficiency in detail has been actually beyond the scope of this manuscript. While we appreciate the concerns raised by the reviewer, we would prefer to reserve full consideration of such topic for a future manuscript.

Comment17: Lines 41-44: it is not the presence of phytate per se but rather the overall high amount of P in seeds that is a main source of P pollution. Even ruminants release a lot of P in their waste. Please rephrase accordingly.

Response: We have rephrased it according to reviewer's comment.

Comment18: Line 52: the term "excitation of Ca²⁺ signals" is inappropriate. Replace with "triggering".

Response: We have corrected it.

Comment19: Line 58: something is missing in the phrase "...undergoes myo-inositol-1-phosphate-synthase". Please rephrase accordingly.

Response: We have rephrased it accordingly.

Comment20: Lines 98 and 110: writing "preliminary study" is incorrect if referring to data generated in the present study.

Response: We have rephrased it accordingly.

Comment21: Fig. 1d: embryos are labeled as "endosperm". Please correct. Furthermore, all embryos shown in the Fig are completely damaged. GUS staining should be shown with intact embryos.

Response: Thanks for pointing out this. We have corrected the label to "embryos", and have provided the GUS staining data with intact embryos in the revised manuscript (Fig. 1d).

Comment22: Line 129 and throughout the manuscript: instead of "pentamutant" please write "quintuple mutant".

Response: We have corrected all "pentamutant" to "quintuple mutant".

Comment23: Line 135 and throughout the manuscript: in plant nutrition, "content" is used to express amount per tissue, organ or whole plant, while amount per unit mass (as $\mu\text{mol g}^{-1}$ DW) refers to "concentration". Please correct in the text and Figs.

Response: We appreciate this suggestion, and have revised them throughout the manuscript.

Comment24: Line 162: previously IPK1 was shown to localize in the nucleus and cytosol (Kuo et al., 2018 doi.org/10.1111/tpj.13974). What could be the reason that IPK1 was not localized in the nucleus in the present study?

Response: We thank the reviewer for this interesting question. Although we do not know the exact reason at present, the difference of IPK1 subcellular localization could be attributed to the distinct tissues or cells used in the two studies. For example, Kuo et al (2018) used a *35Sp:IPK1-YFP* stable transgenic line to monitor the expression of IPK1-YFP in root cells, while we utilized the mesophyll protoplasts transiently

expressing IPK1-GFP. We cannot exclude the possibility that IPK1 may perform distinctly in the intact tissue and free single cells, or in root and shoot.

Comment25: In Fig. 2f, IP concentrations in seeds are expressed as ng μL^{-1} . This suggest that the values are expressed per unit extract. How are these values normalized to the mass of seeds used for extraction?

Response: We have corrected the units of IP concentrations according to reviewer's comments.

Comment26: Please provide a full picture of all SDS/PAGE gels shown in the Figs.

Response: The uncropped pictures of PAGE gels have been provided as the attachment with the revised manuscript.

Comment27: Fig. 1d and elsewhere: two gels are show but without any explanation. Do they show two independent preparations? Furthermore, I cannot find the " \pm SD" from the grey values. Are these values expressed relative to WT? This seems to be the case but the procedure is not clearly described neither in the Material and Methods nor in the legends.

Response: We indeed had provided two gels in each figure. The upper and bottom gels indicate the IP6 accumulation in seeds and seedlings, respectively. We have made the labels more evident in each figure, and provided more detailed explanation in the figure legends. In addition, " \pm SD" has been removed from the according figure legends.

Comment28: Fig. 2: clarify in the Fig that "COM" refers to complementation of the T-4m quadruple mutant.

Response: We have added the corresponding explanations to Fig 2 legends.

Comment29: Lines 173-174 and Fig. 3e: Please clarify whether total P concentration was significantly altered or only slightly changed as it was the case for the pick quadruple and quintuple mutants.

Response: As suggested, we have detected the total P concentration of the genotypes mentioned in the Fig 3e, and found that it was increased in both seeds and seedlings of the *ipk1* mutant compared with that in WT and *pick* mutants (Fig. 3f). This is actually expected, as the *ipk1* mutant shows a more severe reduction of InsP₆ than the *pick* quadruple and quintuple mutants, which may cause a lower InsP₈ accumulation and in turn a higher induction of PSR gene expression, leading to enhanced Pi uptake and translocation in the *ipk1* mutant. Furthermore, we showed that overexpression of *IPK1* was sufficient to restore total P accumulation of *T-4m* to that of WT, favoring that IPK1 is epistatic to PICKs in Pi and InsPs homeostasis.

Fig. 3f. Total P concentration in seeds and seedlings

f Total P concentration in seeds or 7-day-old seedlings. Values are mean \pm SD, n = 3.

Comment30: Lines 188-191: The authors claim that PICK1 cannot phosphorylate IPK1 in vitro. However, can IPK1 be phosphorylated at all? If yes, please include a reference.

Response: To our knowledge, there is currently no evidence to suggest that IPK1 can be phosphorylated.

Comment31: Fig.3f: What does “10P-WT” mean? Was the transcriptome of C-5m and ipk1 compared to wild-type plants grown on low-P conditions? If yes, clarify in the text and add details about this specific growth condition in the legend and Material and Methods.

Response: “10P-WT” indicates the differential expressed genes (DEGs) in WT under low-Pi (10 μ M Pi) condition vs control condition, and “C-5m” or “ipk1” indicates the DEGs in C-5m or ipk1 vs WT under control condition, respectively. We have clarified it in the figure legend and Material and Methods.

Reviewer #1 (Remarks to the Author):

The authors have returned a manuscript bearing a wide range of novel data and considerable speculation.

The title, 'A clade of receptor-like cytoplasmic kinases and 14-3-3 proteins coordinate the lipid-originated inositol hexaphosphate biosynthesis' is not, in this reviewer's opinion, justified. But, this reviewer does hold the view that the manuscript is a substantial contribution.

Addressing my review of this resubmission, I will concentrate on the points that I raised on the original submission - those pertaining to enzymology and analytics of inositol phosphate measurements (and the additional comments of other reviewers on these topics).

Firstly, the revised analysis of the activity of recombinant IPK1 in Suppl Fig 5 is very convincing.

However, the authors have not provided the methodological details : the particular MS/MS transitions by which InsP5 and InsP6 are separately measured. The information is lacking from the Methods and the figure legends, nor is any reference provided.

The same (undefined) method is used to test the effect of inclusion of multiple recombinant proteins on in vitro activity (shown in Figure 2b/c). The legend states mean and sd of $n = 3$ is given in Figure 2c. The HPLC traces in b are single determinations, the source file provides the raw HPLC data for the single determinations of Figure 2b (but not the replicates whose data for InsP5 and InsP6 are extracted and reported in Figure 2c).

No description of the statistical test of differences (or not) is provided. No statement is made whether the data of Figure 2c are biological or technical replicates. The latter point is important - on which much of the narrative manuscript hangs.

The HPLC-MS/MS? method is used again in Supplementary Figure 2. The authors should be congratulated on providing the calibration curves. But, again, no detail is provided of HPLC separations and the critical MS/MS transitions that are used to quantify unknown species of InsP3, InsP4, InsP5, InsP7 and InsP8. It is very likely that these different classes (InsP3 vs InsP4, nor their isomers) are not resolved by HPLC. They likely coelute (we are not shown the data). Consequently the MS/MS transitions used to assign identity (InsP3, InsP4 etc) are of paramount importance. The source data provided is that of the extracted values only. The figure legend states $n=3$. Are these biological or technical replicates, this should be stated?

No details of statistical test are provided, despite the claims of difference associated with the measurement of InsP6.

Beyond this, the difficulty of assigning identity to any of the inositol phosphate species measured (InsP3, isomer undefined, InsP4, isomer undefined, InsP5 isomer undefined) makes it impossible for the authors to make any definitive statement on which pathway of InsP6 synthesis is controlled by PICKs. A pathway as metabolic concept is a statement of which intermediate is converted to which product. The experiments in this manuscript do not answer that question because they have not distinguished between isomers of any class of inositol phosphate, nor is there any analysis of flux.

The PI-PLC ELISA assay whose results are shown in Figure 7d are a 'black box' from which no information about the conduct of the assay or its controls escapes. This reviewer cannot conclude that the PI-PLC ELISA is measuring IPs (PLC activity) in this context ie. against a background of interfering cellular components (including other InsPs).

Reviewer #2 (Remarks to the Author):

In this second revision, the authors have addressed all my previous comments and I can now recommend editors to consider this paper for publication in Nature Communications.

Reviewer #3 (Remarks to the Author):

Most of my concerns (reviewer 3) have been addressed and the authors did overall a good job in addressing the issues.

There are a few points that still need clarification in my opinion:

1. Line 66-70. The wording is still strange: how is it possible to "... transfer(ring) phosphorylation of the phosphate group on ATP"? Do the authors suggest that ATP accepts another phosphate? Maybe it is just about English grammar here.

2. Regarding my concern 10 ... to avoid red and green color because some people are colorblind. I actually referred to the confocal images, when suggesting not to use green and red. Since red and green channels are shown separately also, I want to leave this to the editor.

3. This is my biggest issue: Answer to my comment 19 (and also to concerns raised by other reviewers, e.g. Concern 1 by reviewer 1) where we all emphasized that IPK1 has so far not been implicated in direct InsP7 synthesis. The authors state that they optimized the in vitro kinase system and now find that InsP6 and not InsP7 is generated when using InsP5 as a substrate. I find this strange. In fact, I was excited when I learned about a possible role of IPK1 in InsP7 synthesis (remember that for some InsP7 isomers their biosynthesis remains enigmatic). Why would it be an optimized system if they now lose this exciting activity of InsP7 synthesis? My original point was only to emphasize that this is a new function that has not been described before and that it was not correct to state it is an established function. If I understood correctly, the authors detected in their original protocol InsP7 by MS? If so, how can this be not exciting?

4. There are still some minor language issues but I guess this can be solved at a later stage

Reviewer #4 (Remarks to the Author):

In overall, I am satisfied with the revisions made by the authors. They have addressed all points raised in my initial review, providing new experimental data and further clarifications in the text. The new data and the amendments in the text have significantly improved the manuscript. I only have a few minor comments that require further attention by the authors:

1) According to the new HPLC-MS/MS data shown in Suppl. Fig. 2b, besides InsP5 and InsP6, also the concentrations of InsP3 and InsP8 are decreased in dry seeds of IPCK quintuple mutant. Since InsP7 is not significantly changed, how do the authors explain that InsP8 levels are decreased? This must be discussed clearly in the text.

2) Considering that the authors (page 23 in response letter) "cannot rule out the possibility that PICKs may also affect lipid-independent InsP6 biosynthetic pathway", the lipid-independent pathway should also be more clearly depicted in the model shown in Fig. 8.

3) Furthermore, it should be at least mentioned in the legend of Fig. 8 that not all IPK1 and IPK2 are recruited to the plasma membrane.

4) The Authors' response to comment 14 (page 27) should be mentioned more clearly in the Discussion. Furthermore, would it be possible to estimate the relative distribution of IPK1 and IPK2 by quantifying IPK1::GFP- and IPK2::GFP-derived signal intensities in the plasma membrane and in the cytosol?

5) I understand that a full investigation of all interactions identified in the study in the context of P-sufficient vs P-limiting conditions is beyond the scope of this story. However, this point must be openly mentioned in the Discussion.

6) Please indicate if any signal threshold was applied to the tissue-specific expression patterns

shown in the new Suppl. Fig. 7. If yes, please indicate the signal threshold for each gene and describe how it was made sure that the expression intensities of the indicated genes across different tissues are comparable.

The following outlines our point-by-point responses (in blue) to reviewers' comments:

Reviewer #1 (Remarks to the Author):

The authors have returned a manuscript bearing a wide range of novel data and considerable speculation. The title, 'A clade of receptor-like cytoplasmic kinases and 14-3-3 proteins coordinate the lipid-originated inositol hexaphosphate biosynthesis' is not, in this reviewer's opinion, justified. But, this reviewer does hold the view that the manuscript is a substantial contribution. Addressing my review of this resubmission, I will concentrate on the points that I raised on the original submission - those pertaining to enzymology and analytics of inositol phosphate measurements (and the additional comments of other reviewers on these topics).

Response: We thank the reviewer for these insightful comments. We agree that the evidence for the conclusion that IPCKs regulate the lipid-originated InsP₆ biosynthesis is circumstantial. Therefore, we have removed it from the title and weakened this point in the text as well. In addition, we have provided more detailed information regarding the InsPs detection in the figure legends and methods.

Firstly, the revised analysis of the activity of recombinant IPK1 in Suppl Fig 5 is very convincing. However, the authors have not provided the methodological details: the particular MS/MS transitions by which InsP₅ and InsP₆ are separately measured. The information is lacking from the Methods and the figure legends, nor is any reference provided.

Response: We thank the reviewer for this concern. The InsP₅ and InsP₆ were identified based on their molecular mass size displayed by the mass spectrometry results combined with the reference function of the standard sample. We have provided more methodological details regarding the HPLC-MS/MS analysis in the revised manuscript, which are as follows:

‘InsPs were detected using Hydrophilic Interaction High Performance Liquid Chromatography-Tandem Mass Spectrometry on an Agilent 1290 infinity HPLC system coupled to an Agilent 6460 triple Quad LC-MS/MS using InfinityLab Poroshell 120 HILIC-Z (2.1 × 100) column (Agilent Technologies, USA). Nitrogen was used as the sheath gas and drying gas. The nebulizer pressure was set to 45 psi and the flow rate of drying gas was 5 L / min. The flow rate and temperature of the sheath gas were 11 L / min and 350°C, respectively. Chromatographic separation was carried out on a HPLC column (100 × 2.1 mm, 2.7 μm). The column temperature was 35°C. The mobile phases consisted of (A) ammonium acetate in distilled water (pH 10.0) and (B) Acetonitrile. The gradient program was as follows: 0-10 min, 90 % →

55 % of B; 10-12 min, 55 % → 90 % of B; 12-20 min, 90 % of B. The flow rate was set at 0.3 ml / min. Mass spectrometric detection was completed by use of an electrospray ionization (ESI) source in negative ion multiple-reaction monitoring (MRM) mode. InsPs were identified based on comparison to known InsPs species. The mass spectrometry parameters corresponding to different InsPs show as below: InsP₃ (MRM: 419 → 321, 419 → 337, Acquisition time is 6-7 min); InsP₄ (MRM: 499 → 401, 499 → 417, Acquisition time is 6-7 min); InsP₅ (MRM: 579 → 480.9, 579 → 382.8, Acquisition time is 6-7 min); InsP₆ (MRM: 659 → 560.8, 659 → 577, Acquisition time is 6-7 min); InsP₇ (MRM: 739 → 575, 739 → 657, Acquisition time is 5-6 min); InsP₈ (MRM: 819 → 737, 819 → 655, Acquisition time is 4-5 min). According to the regression equation calculated from the standard sample, substitute the response value of the sample into the equation to convert the corresponding concentration.'

The same (undefined) method is used to test the effect of inclusion of multiple recombinant proteins on in vitro activity (shown in Figure 2b/c). The legend states mean and sd of n =3 is given in Figure 2c. The HPLC traces in bare single determinations, the source file provides the raw HPLC data for the single determinations of Figure 2b (but not the replicates whose data for InsP₅ and InsP₆ are extracted and reported in Figure 2c). No description of the statistical test of differences (or not) is provided. No statement is made whether the data of Figure 2c are biological or technical replicates. The latter point is important - on which much of the narrative manuscript hangs.

Response: We guess the reviewer referred to the Figure 5b/c instead of Figure 2b/c. The same method of HPLC-MS/MS was used in Figure 5b/c. The data in Figure 5c represent three biological repeats, and were analyzed by unpaired t-test. We have provided these information in the figure legend. Moreover, we only showed the HPLC traces in single determinations in Figure 5b, because three biological repeats would make the diagram presentation so complex. Nevertheless, we have provided the raw data of three biological repeats in the source file.

The HPLC-MS/MS? method is used again in Supplementary Figure 2. The authors should be congratulated on providing the calibration curves. But, again, no detail is provided of HPLC separations and the critical MS/MS transitions that are used to quantify unknown species of InsP₃, InsP₄, InsP₅, InsP₇ and InsP₈. It is very likely that these different classes (InsP₃ vs InsP₄, nor their isomers) are not resolved by HPLC. They likely coelute (we are not shown the data). Consequently the MS/MS transitions used to assign identity (InsP₃, InsP₄ etc) are of paramount importance. The source data provided is that of the extracted values only. The figure legend states n=3. Are these biological or technical replicates, this should be stated?

Response: We thank the reviewer for pointing out this. As mentioned above, we have

provided more detailed information regarding HPLC-MS/MS detection in the revised manuscript. The HPLC-MS/MS methods used in this study are same, the only difference are the sample solvents for elution of InsPs from TiO₂ beads. The solvent used in Supplementary Figure 2 (InsP detection in plants and seeds enriched with TiO₂ beads) was 10% ammonia solution, while the ones used in Figure 5 b/c and Supplementary Figure 5 (*in vitro* InsP detection) were acetonitrile containing reaction substances. We agree that HPLC cannot separate the different InsPs. Nevertheless, the enrichment of InsPs on TiO₂ beads helps to remove most of the impurities before the process of HPLC, and the different InsPs can be identified based on their molecular mass size displayed by the mass spectrometry results combined with the reference function of the standard sample. Indeed, HPLC-MS/MS has been widely used to detect InsPs (Zajdel A et al. Biomed Res Int. 2013, doi: 10.1155/2013/147307; Lam G et al. Drug Test Anal. 2014, doi: 10.1002/dta.1473; Paraskova JV et al. Anal Chem. 2015. doi: 10.1021/ac5033484). Additionally, we have provided more details of the raw data in the source file. 'n=3' indicates three biological repeats, which has been clarified in the figure legend.

No details of statistical test are provided, despite the claims of difference associated with the measurement of InsP6.

Response: We thank the reviewer for pointing out it, and have provided the information of statistical analysis in the figure legends.

Beyond this, the difficulty of assigning identity to any of the inositol phosphate species measured (InsP3, isomer undefined, InsP4, isomer undefined, InsP5 isomer undefined) makes it impossible for the authors to make any definitive statement on which pathway of InsP6 synthesis is controlled by PICKs. A pathway as metabolic concept is a statement of which intermediate is converted to which product. The experiments in this manuscript do not answer that question because they have not distinguished between isomers of any class of inositol phosphate, nor is there any analysis of flux.

Response: We appreciate this comment, and agree that it's currently difficult to distinguish the isomers of any class of InsPs. Therefore, as we mentioned in the first point, we have weakened the conclusion in the text that the lipid-dependent InsP₆ synthetic pathway is controlled by IPCKs. We have removed 'lipid-originated' from the title as well.

The PI-PLC ELISA assay whose results are shown in Figure 7d are a 'black box' from which no information about the conduct of the assay or its controls escapes. This reviewer cannot conclude that the PI-PLC ELISA is measuring IPs (PLC activity) in this context ie. against a background of interfering cellular components (including

other InsPs).

Response: We thank the reviewer for pointing out this. The PI-PLC activity is generally detected by the isotope labeling method in plants (Komis, et al., 2008, *New Phytologist*; Huang, et al., 2009, *Planta*; Zhang, et al., 2015, *J. Plant Biochem. Biotechnol.*), but we do not have the conditions to do so. Because of the time crunch, we chose a commercial ELISA kit to detect the PI-PLC activity. This kit measuring PI-PLC activity is achieved by the antibodies that specifically recognize the activated PI-PLCs with increased phosphorylation or acetylation (Coursol et al., 2000, *Plant J*; Martelli et al., 2000, *FEBS Letters*; Balasubramanian et al., 2008, *Leukemia*), but not by ones recognizing InsPs. We apologize for the oversight of its reasonableness, as the properties of the antibodies and the composition of the solution used in the ELISA kit are largely not available because of the commercial secret. We thus agree that the results from this ELISA assay is not so convincing. Since we have adopted the reviewer's suggestion that the lipid-dependent pathway should not be concluded to the way in which IPCKs regulate InsP₆ biosynthesis, then PI-PLC is no longer the subject of this study, and the data shown in Figure 7d is indeed not an essential evidence for the main conclusion. Therefore, to ensure the accuracy of the data, we have removed the Figure 7d data from the manuscript. Nevertheless, we have shown that the phosphorylation of GRF4 by IPCKs promotes its interaction with PI-PLC2 (Figure 7c), and that the concentration of InsP₃ is reduced in the *C-5m* mutant versus WT (Supplementary Fig.2b), which in some ways support the potential regulation of PI-PLC by IPCKs.

Reviewer #2 (Remarks to the Author):

In this second revision, the authors have addressed all my previous comments and I can now recommend editors to consider this paper for publication in *Nature Communications*.

Response: We thank the reviewer for the time.

Reviewer #3 (Remarks to the Author):

Most of my concerns (reviewer 3) have been addressed and the authors did overall a good job in addressing the issues.

There are a few points that still need clarification in my opinion:

1. Line 66-70. The wording is still strange: how is it possible to "... transfer(ring) phosphorylation of the phosphate group on ATP"? Do the authors suggest that ATP accepts another phosphate? Maybe it is just about English grammar here.

Response: We thank the reviewer for pointing out this. We have corrected it accordingly.

2. Regarding my concern 10 ... to avoid red and green color because some people are colorblind. I actually referred to the confocal images, when suggesting not to use green and red. Since red and green channels are shown separately also, I want to leave this to the editor.

Response: We thank the reviewer for this suggestion. We feel very sorry that the combination of red and green color will cause trouble to red-green color blindness group. However, we think this may be not the case for the confocal images, because the red and green channels are shown separately, and the merge of red and green displays a yellow color. We prefer to use the mCherry and GFP over YFP, because the absorption peak difference between mCherry (770 to 622 nm) and GFP (577 to 492 nm) is larger than that between mCherry and YFP (597 to 577 nm), resulting in less interference between channels.

3. This is my biggest issue: Answer to my comment 19 (and also to concerns raised by other reviewers, e.g. Concern 1 by reviewer 1) where we all emphasized that IPK1 has so far not been implicated in direct InsP7 synthesis. The authors state that they optimized the *in vitro* kinase system and now find that InsP6 and not InsP7 is generated when using InsP5 as a substrate. I find this strange. In fact, I was excited when I learned about a possible role of IPK1 in InsP7 synthesis (remember that for some InsP7 isomers their biosynthesis remains enigmatic). Why would it be an optimized system if they now lose this exciting activity of InsP7 synthesis? My original point was only to emphasize that this is a new function that has not been described before and that it was not correct to state it is an established function. If I understood correctly, the authors detected in their original protocol InsP7 by MS? If so, how can this be not exciting?

Response: We thank the reviewer for this comment. As suggested by Reviewer #1, we indeed found that there were a series of problems in the original reaction system of the *in vitro* IPK1 activity assay. For instance, the recovery rate of substrates and products were very low (less than 10%), the decrease of the substrates was not correlated well with the increase of the products, and only InsP₇ but no InsP₆ was detected in the products. Although we really have no idea about how InsP₇ was produced in this system, these problems have made the results not convincing. After optimization of the protocol (as mentioned in the last response letter), the assay becomes more reliable, with more stable and appropriate reaction parameters (e.g. the recovery rate, the correlation of substrates and products, the product synthesis rate, *K_m* value etc.), and only InsP₆ rather than InsP₇ is detected in the products. Therefore, we prefer to believe the latter results. We appreciate the reviewer's interest in the possible role of IPK1 on InsP₇ synthesis, which is indeed an interesting topic but needs more future work to prove.

4. There are still some minor language issues but I guess this can be solved at a later

stage

Response: We appreciate these feedbacks.

Reviewer #4 (Remarks to the Author):

In overall, I am satisfied with the revisions made by the authors. They have addressed all points raised in my initial review, providing new experimental data and further clarifications in the text. The new data and the amendments in the text have significantly improved the manuscript. I only have a few minor comments that require further attention by the authors:

1) According to the new HPLC-MS/MS data shown in Suppl. Fig. 2b, besides InsP5 and InsP6, also the concentrations of InsP3 and InsP8 are decreased in dry seeds of IPCK quintuple mutant. Since InsP7 is not significantly changed, how do the authors explain that InsP8 levels are decreased? This must be discussed clearly in the text.

Response: We thank the reviewer for pointing out this. We were also surprised by the result that lack of IPCKs reduces the concentration of InsP₈ without affecting that of InsP₇. We speculate that InsP₇ may function as a transitional intermediate in InsP₈ synthesis in plants. Once synthesized, the InsP₇ might be rapidly converted to InsP₈ to maintain the phosphorus homeostasis. In line with this, the concentration of InsP₇ is much lower than that of InsP₈ in WT plants (Supplementary Figure 2b). Additionally, although we have found that IPCKs do not interact with VIH1 (the rate-limiting enzyme responsible for InsP₈ synthesis), we cannot completely rule out the possibility that IPCKs may indirectly affect VIH1 expression or activity. We have included these points in the Discussion.

2) Considering that the authors (page 23 in response letter) “cannot rule out the possibility that PICKs may also affect lipid-independent InsP6 biosynthetic pathway”, the lipid-independent pathway should also be more clearly depicted in the model shown in Fig. 8.

Response: We thank the reviewer for this suggestion, and have added the lipid-independent pathway into the model.

3) Furthermore, it should be at least mentioned in the legend of Fig. 8 that not all IPK1 and IPK2 are recruited to the plasma membrane.

Response: We have revised the legend of Figure 8 accordingly.

4) The Authors' response to comment 14 (page 27) should be mentioned more clearly in the Discussion. Furthermore, would it be possible to estimate the relative distribution of IPK1 and IPK2 by quantifying IPK1::GFP- and IPK2::GFP-derived

signal intensities in the plasma membrane and in the cytosol?

Response: As suggested, we have clarified the comment 14 in the Discussion. Moreover, we appreciate the valuable suggestion to estimate the relative distribution of IPK1 and IPK2 by quantifying IPK1::GFP- and IPK2::GFP-derived signal intensities in the plasma membrane and in the cytosol, but unfortunately we have no *IPK1::GFP* or *IPK2::GFP* transgenic lines at present. We therefore would like to reserve it as future work.

5) I understand that a full investigation of all interactions identified in the study in the context of P-sufficient vs P-limiting conditions is beyond the scope of this story. However, this point must be openly mentioned in the Discussion.

Response: We agree with this suggestion, and have included this point in the Discussion.

6) Please indicate if any signal threshold was applied to the tissue-specific expression patterns shown in the new Suppl. Fig. 7. If yes, please indicate the signal threshold for each gene and describe how it was made sure that the expression intensities of the indicated genes across different tissues are comparable.

Response: We thank the reviewer for pointing out it. The certain signal threshold was applied to *GRF4* and *PI-PLC2* in Supplementary Figure 7, because they have much higher expression levels than other genes. To make it less confusing, we have removed the threshold in the revised version.

Reviewer #1 (Remarks to the Author):

The authors have returned a manuscript which advances the same narrative that IPCKs control a pathway of InsP6 synthesis.

I will comment on the quality of the inositol phosphate and phosphate analysis by reference to the published literature.

Without identification of isomers, it is impossible to discriminate between different pathways. Yet, the methods that allow thorough identification of inositol phosphate species have existed for 30y and have been used by others. More recently, CE-MS has been used to great effect.

In light of this, the authors should have compared their results with what is known about inositol phosphate species in plants and in Arabidopsis. Had the authors compared the levels of inositol phosphates that they report with published values obtained by others (Bentsink et al 2003) or Riemer et al [ref 34], they would see massive disparities in what they report with the published literature.

Had the authors distinguished between stereoisomers, they would be able to make more incisive arguments. They are left with the observation that InsP6 is altered in IPCK mutants, but the values they obtain bear no relation to the published literature.

Consequently, this reviewer has no confidence in the inositol phosphate analysis of tissue of plants reported this manuscript.

The authors should also have considered how their measured values for P compare with other studies. There are substantive disparities here. Again, Riemer et al [ref 34] or Nagarajan et al Plant Physiol 2011 Jul;156(3):1149-63. doi: 10.1104/pp.111.174805. Epub 2011 May 31. Arabidopsis Pht1;5 mobilizes phosphate between source and sink organs and influences the interaction between phosphate homeostasis and ethylene signaling. provide comparators.

Reviewer #4 (Remarks to the Author):

The authors have addressed to all my further comments and, in overall, responded satisfactorily. The only exception is the rephrasing of the legend of Fig. 8 (lines 998-1001), were the authors were asked to indicate that not all IPK1, IPK2 and PLCs interact with and are recruited by IPCKs. The way the revised sentences are written is confusing. My suggestion is to rephrase as follows: "IPCKs interact with part of IPK1, IPK2s and PI-PLCs pools, facilitating the activity of these enzymes, via phosphorylation of GRF 14-3-3 proteins. The recruited enzymes form a potential plasma membrane-located complex, therefore establishing a spatially efficient InsP6 biosynthetic pathway."

Other than that, I want to congratulate the authors for this impressive work with several of novel and relevant messages.

Reviewer #1 (Remarks to the Author):

The authors have returned a manuscript which advances the same narrative that IPCKs control a pathway of InsP6 synthesis. I will comment on the quality of the inositol phosphate and phosphate analysis by reference to the published literature. Without identification of isomers, it is impossible to discriminate between different pathways. Yet, the methods that allow thorough identification of inositol phosphate species have existed for 30y and have been used by others. More recently, CE-MS has been used to great effect. In light of this, the authors should have compared their results with what is known about inositol phosphate species in plants and in Arabidopsis. Had the authors compared the levels of inositol phosphates that they report with published values obtained by others (Bentsink et al 2003) or Riemer et al [ref 34], they would see massive disparities in what they report with the published literature. Had the authors distinguished between stereoisomers, they would be able to make more incisive arguments. They are left with the observation that InsP6 is altered in IPCK mutants, but the values they obtain bear no relation to the published literature. Consequently, this reviewer has no confidence in the inositol phosphate analysis of tissue of plants reported this manuscript.

The authors should also have considered how their measured values for P compare with other studies. There are substantive disparities here. Again, Riemer et al [ref 34] or Nagarajan et al Plant Physiol 2011 Jul;156(3):1149-63. doi: 10.1104/pp.111.174805. Epub 2011 May 31. Arabidopsis Pht1;5 mobilizes phosphate between source and sink organs and influences the interaction between phosphate homeostasis and ethylene signaling. provide comparators.

Response: We thank the reviewer for the suggestion of the CE-MS for identification of the InsP isomers. Unfortunately, we have indeed no condition to do so. Therefore, in the revised version we do not mention or make any conclusion that IPCKs control a particular pathway of InsP6 synthesis, but only state that IPCKs regulate InsP6 synthesis or accumulation. We only used biosynthesis or synthesis in the cases when enzymes are involved or discussed.

We also thank the reviewer for pointing out the unusual results of the InsPs content that we did not notice before, as we focused particularly on the difference of the InsPs content between WT and *ipck* mutant, and neglected to compare the data with other published literatures. We realized that the levels of InsPs in seedlings shown in Fig 2 and Fig S2 were indeed much higher than that in other published literatures, which promoted us to check the experimental procedures and data processing of the HPLC-MS/MS-dependent InsPs detection, and we have found critical mistakes in the data processing that caused the large disparities. The technician who is responsible for HPLC-MS/MS mistook the unit of the standard curve ($\mu\text{g}/\text{ml}$) for $\mu\text{g}/\text{mg}$. As a result, the concentration of InsPs ($\mu\text{g}/\text{mg}$) were calculated directly based on the standard curve without multiplying by sample volume (ml) and in turn being divided by sample weight (g) (please see the detailed data processing and correction processes listed in the **Data**

Correction and Comparison sheet), which made the unit of final InsP concentration from $\mu\text{g/g}$ to $\mu\text{g/mg}$. In addition, we have also found that the calculation of InsP6 levels in seeds did not take into account its dilution multiple (Fig 2). Since 2.4 g seeds were used to detect InsP6 concentration, the samples were actually diluted 100 times by the technician before HPLC-MS/MS assay. In the **Data Correction and Comparison** sheet, we have corrected this calculation, and also provided another repeat of InsP6 detection data using 0.01 g seeds as a sample, showing that these results are comparable. The raw data of HPLC-MS/MS have been uploaded to Metabolights database but has not yet been curated for access before the completion of review process by the administrator, instead we provided all the corresponding chromatographies in the excel files **Fig. 2.** and **Supplementary Fig. 2 included in Source data.rar** for your reference. We therefore feel very sorry for these mistakes in the data processing, which were mainly caused by the insufficient communication between the authors and the technician. We are also really grateful to the reviewer for pointing out the problems with the data, and letting us have the chance to correct the careless mistakes in data processing.

After correction, the concentration of InsP6 in WT seedlings in this study is $\sim 3 \mu\text{g/g}$, which is within the range of $0.36\text{--}28 \mu\text{g/g}$ in Arabidopsis leaves in the published literatures (Bentsink et al., 2003, Theor Appl Genet, 106: 1234; Riemer et al., 2021, Mol Plant, 14: 1864). Moreover, the InsP6 concentration in WT seeds in this study is now $\sim 3 \text{ mg/g}$, which is also in the same order of magnitude ($2\text{--}20 \text{ mg/g}$) reported in other literature (Bentsink et al., 2003). The existing difference of InsP6 content between our study and other published research can be caused by factors such as the different genetic backgrounds of the plants used, the distinct plant growth conditions, different tissue sampling and/or methods for InsP6 detection and etc. With these variations, the leaf InsP6 concentration (550 pmol/g , $\sim 0.36 \mu\text{g/g}$) obtained by Riemer et al (2021) was much lower than that ($2\text{--}28 \mu\text{g/g}$) by Bentsink et al (2003), and Vogiatzaki et al (2017) detected $\sim 20 \mu\text{mol/g}$ ($\sim 0.63 \text{ mg/g}$) of total P in Arabidopsis seeds (Vogiatzaki et al., 2017, Curr Biol, 27: 2893), which indicated that the InsP6 concentration was less than 0.63 mg/g in these seeds. Overall, despite the difference, the corrected InsP6 content in this study is to a large extent comparable to that in other published literatures.

In addition, as suggested by the reviewer, we have compared the P content with that in other studies, particularly reported by Riemer et al (2021) and Nagarajan et al (2011, Plant Physiol, 156:1149). Riemer et al (2021) detected $\sim 8 \text{ mg/g DW}$ (dry weight) of shoot P concentration in 30-day-old WT Arabidopsis plants, and Nagarajan et al (2011) detected $\sim 90 \mu\text{mol/g}$ ($\sim 2.8 \text{ mg/g}$) DW of P concentration in 12-day-old WT seedlings. Both of the data were based on dry weight. By comparison, we detected $\sim 0.7 \text{ mg/g FW}$ (fresh weight) of P concentration in 7-day-old WT seedlings in this study. Since the ratio of dry weight to fresh weight is generally around 10%, the P concentration in our study is then estimated to be $\sim 7 \text{ mg/g DW}$, and is therefore comparable to the data shown in the above references.

Reviewer #4 (Remarks to the Author):

The authors have addressed to all my further comments and, in overall, responded satisfactorily. The only exception is the rephrasing of the legend of Fig. 8 (lines 998-1001), where the authors were asked to indicate that not all IPK1, IPK2 and PLCs interact with and are recruited by IPCKs. The way the revised sentences are written is confusing. My suggestion is to rephrase as follows:

“IPCKs interact with part of IPK1, IPK2s and PI-PLCs pools, facilitating the activity of these enzymes, via phosphorylation of GRF 14-3-3 proteins. The recruited enzymes form a potential plasma membrane-located complex, therefore establishing a spatially efficient InsP6 biosynthetic pathway.”

Other than that, I want to congratulate the authors for this impressive work with several of novel and relevant messages.

Response: We thank the reviewer for the suggestion, and have made modifications accordingly (line 983-986) in the revised manuscript.

Reviewer #1 (Remarks to the Author):

This reviewer thanks the authors for returning a manuscript with the raw data that allows a more careful assessment of the validity of the claims of the manuscript. The manuscript is improved by the inclusion of the data. The reviewer has a few minor points/suggestions for final corrections.

They are:

Lines 69-73. The sentence 'In addition IPK1 presumably cooperates with IPK2..' ignores a body of literature that identified significant reductions in InsP6 content of ITPK mutants. The seminal reference of Kim and Tai <https://link.springer.com/article/10.1007/s00438-011-0631-2> [14] could be cited.

The sentence could, without bias, be written, 'In addition IPK1 presumably cooperates with IPK2 and or ITPKs ...'.

Presented this way, ie. before presentation of results, is a less biased reading of the literature.

Line 357 Mammals have IPMKs (IPK2 orthologs) and IP3 3-Kinase, this sentence and the manuscript potentially confuses the two different enzyme classes.

Line 548 The use of 'D' for D-myo-Ins(1,3,4,5,6) is incorrect: use myo-Ins(1,3,4,5,6)P5.

Line 552 should state the volume injected onto HPLC.

Nomenclature: please refer to InsPs as InsP3, InsP4 etc and Ins(1,3,4,5)P4, Ins(1,3,4,5,6)P5 etc. Do not use other forms; myo-should be italicized wherever and appropriate subscripts used.

While the authors have explained how the values for InsP levels have been corrected in their rebuttal, the discussion makes no comment on the levels of different inositol phosphates or PP-InsPs measured. It should do so, and it should do so with explicit reference to the very thorough work in the literature (work of Laha, Schaaf, Jessen, there are several papers to be cited - they are already in the list of citations).

The discussion should be candid enough to say the inability to describe stereoisomers or enantiomers does not allow the authors to discount effect of IPCKs on InsP6 synthesis arising from ITPKs (citing [14] :10.1007/s00438-011-0631-2).

The claim in lines 331-33, 'In organisms from yeast to humans, 'IPMK (Inositol polyphosphate multikinase) / IPK2 is often the only enzyme responsible for converting InsP4 into InsP5, is not correct: ITPK1, human and plant, has been shown to perform the same function(10.1126/stke.14pe5; 10.1042/BCJ20200423; 10.1073/pnas.1911431116). These or similar references should be cited.

There are a few instances of poor phrasing that should be corrected in the journal's editing process.

This reviewer thanks the authors for their substantial endeavour.

REVIEWERS' COMMENTS

Reviewer #1 (Remarks to the Author):

This reviewer thanks the authors for returning a manuscript with the raw data that allows a more careful assessment of the validity of the claims of the manuscript. The manuscript is improved by the inclusion of the data. The reviewer has a few minor points/suggestions for final corrections.

They are:

Lines 69-73. The sentence 'In addition IPK1 presumably cooperates with IPK2..' ignores a body of literature that identified significant reductions in InsP6 content of ITPK mutants. The seminal reference of Kim and Tai <https://link.springer.com/article/10.1007/s00438-011-0631-2> [14] could be cited. The sentence could, without bias, be written, 'In addition IPK1 presumably cooperates with IPK2 and or ITPKs ...'. Presented this way, ie. before presentation of results, is a less biased reading of the literature.

Response: Thanks for this suggestion, we have corrected this sentence and added the corresponding citation in the revised manuscript (Line 71-73).

Line 357 Mammals have IPMKs (IPK2 orthologs) and IP3 3-Kinase, this sentence and the manuscript potentially confuses the two different enzyme classes.

Response: We appreciate this comment. Inositol polyphosphate multikinase (IPMKs) was identified as an enzyme that generates a series of water-soluble inositol phosphates (DOI: 10.1073/pnas.0506184102), and Inositol 1,4,5-trisphosphate 3-kinase (IP3 3-kinase/IP3K) plays an important role in signal transduction in animal cells by phosphorylating inositol 1,4,5-trisphosphate (IP3) to inositol 1,3,4,5-tetrakisphosphate (IP4) (<https://doi.org/10.1038/sj.cr.7290270>). Both enzymes were considered to function in generating InsPs. Therefore, we have revised the related description (Line 331-333 and 357-358).

Line 548 The use of 'D' for D-myo-Ins(1,3,4,5,6) is incorrect: use myo-Ins(1,3,4,5,6)P5.

Response: We thank the reviewer for pointing out it, and have revised them accordingly in the text (Line 569 and 625).

Line 552 should state the volume injected onto HPLC.

Response: The volume injected onto HPLC is 10 μ l. We have added it in Line 584.

Nomenclature: please refer to InsPs as InsP3, InsP4 etc and Ins(1,3,4,5)P4, Ins(1,3,4,5,6)P5 etc. Do not use other forms; myo-should be italicized wherever and appropriate subscripts used.

Response: We have checked thoroughly about the forms of InsPs, and we also changed all myo- into italicized.

While the authors have explained how the values for InsP levels have been corrected in their rebuttal, the discussion makes no comment on the levels of different inositol phosphates or PP-InsPs measured. It should do so, and it should do so with explicit reference to the very thorough work in the literature (work of Laha, Schaaf, Jessen, there are several papers to be cited - they are already in the list of

citations). The discussion should be candid enough to say the inability to describe stereoisomers or enantiomers does not allow the authors to discount effect of IPCKs on InsP6 synthesis arising from ITPKs (citing [14] :10.1007/s00438-011-0631-2).

Response: We have added this point in the discussion (Line 383-385).

The claim in lines 331-33, 'In organisms from yeast to humans, 'IPMK (Inositol phlyphosphate multikinase) / IPK2 is often the only enzyme responsible for converting InsP4 into InsP5, is not correct: ITPK1, human and plant, has been shown to perform the same functon(10.1126/stke.14pe5; 10.1042/BCJ20200423; 10.1073/pnas.1911431116). These or similar references should be cited.

Response: We appreciate this comment, and have revised this description (Line 331-333) accordingly. To avoid the disruption of the logic flow, ITPK1 is not mentioned here. Therefore, the ITPK1-related references mentioned by the reviewer are not cited.

There are a few instances of poor phrasing that should be corrected in the journal's editing process. This reviewer thanks the authors for their substantial endeavour.

Response: Thank you for your comments.